# Theoretical and Practical Analysis of Fréchet Regression via Comparison Geometry

## Abstract

Fréchet regression extends classical regression methods to non-Euclidean metric spaces, enabling the analysis of data relationships on complex structures such as manifolds and graphs. This work establishes a rigorous theoretical analysis for Fréchet regression through the lens of comparison geometry which leads to important considerations for its use in practice. The analysis provides key results on the existence, uniqueness, and stability of the Fréchet mean, along with statistical guarantees for nonparametric regression, including exponential concentration bounds and convergence rates. Additionally, insights into angle stability reveal the interplay between curvature of the manifold and the behavior of the regression estimator in these non-Euclidean contexts. Empirical experiments validate the theoretical findings, demonstrating the effectiveness of proposed hyperbolic mappings, particularly for data with heteroscedasticity, and highlighting the practical usefulness of these results.

## 1 Introduction

Fréchet regression [35] is a powerful statistical tool for analyzing relationships between variables when the response or predictor lies in a non-Euclidean space. It generalizes classical regression to settings where the response variable $Y$ resides in a metric space $\mathcal{M}$. Given predictors $X$, Fréchet regression seeks to estimate the conditional Fréchet mean.

$$\mu(x) = \arg\min_{m \in \mathcal{M}} \mathbb{E}\left[d^2(Y, m) \mid X = x\right], \tag{1}$$

where $d$ is the metric on $\mathcal{M}$. This approach accommodates data in various non-Euclidean spaces, such as manifolds, trees, and graphs [29, 17, 18, 36, 13]. In recent years, several variants of Fréchet regression have been proposed [39, 7, 37, 19, 44, 42], each addressing different aspects such as variable selection, error modeling, and high-dimensional data handling. However, most existing studies primarily focus on specific geometric settings or lack a comprehensive theoretical framework that accounts for varying curvature bounds. This study fills this gap by leveraging comparison geometry to provide a unified theoretical analysis of Fréchet regression across $\mathrm{CAT}(K)$ spaces with diverse curvature properties.

Fréchet regression allows the assumption of a non-Euclidean space in the space of the data, so one can expect that its behavior can be described depending on the geometrical properties of the space. To investigate this, this study utilizes comparison geometry, which is a fundamental branch of differential geometry that investigates the geometric properties of a given space by comparing it to model spaces of constant curvature [12, 20, 11, 41]. Unlike information geometry [3, 5, 33, 4, 27, 28], which focuses on general statistical manifolds, this framework leverages classical comparison theorems to derive insights about the structure and behavior of more complex or less regular spaces. By establishing inequalities and structural similarities between a target space and well-understood model

spaces (e.g., Euclidean, spherical, or hyperbolic geometries), comparison geometry enables the extension of geometric and topological results to broader contexts, including spaces that may lack smoothness or traditional manifold structures. In this framework, $\mathrm{CAT}(K)$ spaces are pivotal objects of study, which are the generalization of constant curvature space [6, 22, 9]. $\mathrm{CAT}(K)$ spaces are geodesic metric spaces, where geodesic triangles are thinner than their comparison triangles in the model space of constant curvature $K$. Consider several known examples of $\mathrm{CAT}(K)$ spaces. Euclidean spaces $\mathbb{R}^n$ are classic examples with $K = 0$, exhibiting flat geometry. Hyperbolic spaces, which have constant negative curvature ($K < 0$), serve as models for spaces exhibiting exponential growth and are useful in areas like network analysis and evolutionary biology. On the other hand, trees can be viewed as $\mathrm{CAT}(0)$ spaces, providing a discrete analog with unique geodesics between points. Additionally, certain types of manifold structures used in shape analysis and computer graphics also qualify as $\mathrm{CAT}(K)$ spaces under specific curvature conditions. These examples demonstrate the broad applicability of $\mathrm{CAT}(K)$ spaces in modeling diverse geometric contexts encountered in statistical analysis. By considering such spaces, this study aims to describe the behavior of the Fréchet regression in terms of curvature $K$ in particular.

## 2 Notation

In this section, the notations and definitions required for the following analysis are organized. Let $\mathcal{M}$ be a metric space and $d$ be the metric on $\mathcal{M}$. Here, the metric space $(\mathcal{M}, d)$ is geodesic space if every pair of points in $\mathcal{M}$ can be connected by a geodesic, a curve whose length equals the distance between the points.

**Definition 1** ($\mathrm{CAT}(K)$ space). *Let $(\mathcal{M}, d)$ be a geodesic metric space and let $K \in \mathbb{R}$. The space $\mathcal{M}$ is said to be a $\mathrm{CAT}(K)$ space if it satisfies the following curvature condition: for any geodesic triangle $\triangle pqr$ in $\mathcal{M}$ with perimeter less than $2D_K$ (where $D_K = \pi/\sqrt{K}$ if $K > 0$, and $D_K = \infty$ otherwise), and for any points $x, y$ on the edges $[pq]$ and $[qr]$ respectively, the distance between $x$ and $y$ in $\mathcal{M}$ does not exceed the distance between the corresponding points $\bar{x}$ and $\bar{y}$ on the comparison triangle $\triangle \bar{p}\bar{q}r$ in the model space of constant curvature $K$: $d(x, y) \leq d_{\mathbb{M}_K^2}(\bar{x}, \bar{y})$, where the comparison triangle $\triangle \bar{p}\bar{q}r$ is a triangle in the simply connected, complete 2-dimensional Riemannian manifold $\mathbb{M}_K^2$ of constant curvature $K$ that preserves the side lengths as $d_{\mathbb{M}_K^2}(\bar{p}, \bar{q}) = d(p, q)$, $d_{\mathbb{M}_K^2}(\bar{q}, \bar{r}) = d(q, r)$, and $d_{\mathbb{M}_K^2}(\bar{r}, \bar{p}) = d(r, p)$.*

**Definition 2** (Geodesic convexity). *A function $f \colon \mathcal{M} \to \mathbb{R}$ is geodesically convex if for every geodesic $\gamma \colon [0, 1] \to \mathcal{M}$, $f(\gamma(t)) \leq (1 - t)f(\gamma(0)) + tf(\gamma(1))$, for all $t \in [0, 1]$.*

**Definition 3** ($\lambda$-strong geodesic convexity). *A function $f \colon \mathcal{M} \to \mathbb{R}$ is $\lambda$-strongly geodesically convex around $p \in \mathcal{M}$ if there exists a constant $\lambda > 0$ depending only on $K$ and $\mathrm{diam}(\mathcal{M})$ such that*

$$f(x) - f(p) \geq \lambda d^2(x, p), \tag{2}$$

*for every $x \in \mathcal{M}$.*

**Definition 4** (Lower semicontinuity). *A functional $F \colon \mathcal{M} \to \mathbb{R} \cup \{+\infty\}$ is lower semicontinuous at a point $x \in \mathcal{M}$ if for every sequence $\{x_n\}$ converging to $x$, it satisfies*

$$F(x) \leq \liminf_{n \to +\infty} F(x_n). \tag{3}$$

**Definition 5** (Weak convergence in metric space). *A sequence of probability measures $\{\nu_n\}$ on $\mathcal{M}$ is said to converge weakly to a probability measure $\nu$ (denoted by $\nu_n \Rightarrow \nu$) if for every bounded continuous function $f \colon \mathcal{M} \to \mathbb{R}$,*

$$\lim_{n \to +\infty} \int_{\mathcal{M}} f(y)d\nu_n(y) = \int_{\mathcal{M}} f(y)d\nu(y).$$

**Definition 6** (Alexandrov angle). *The Alexandrov angle $\angle_x(y, z)$ is defined as the limit of secular angles between short sub-segments. Concretely, if $y'$ is a point on $[xy]$ with $d(x, y') \to 0$ and $z'$ is a point on $[xz]$ with $d(x, z') \to 0$. Then,*

$$\angle_x(y, z) := \lim_{y' \to x, z' \to x} \angle_x^{(\mathrm{sec})}(y'z'),$$

*where $\angle_x^{(\mathrm{sec})}(y'z')$ is the ordinary angle in the comparison triangle for $\triangle xy'z'$ in the model space.*

**Definition 7** (Riemannian exponential map)**.** *Let $T_z\mathcal{M}$ be the tangent space of $\mathcal{M}$ at a point $z \in \mathcal{M}$. For a fixed point $z$, the Riemannian exponential map at $z$, denoted by $\exp_z$ is a map from the tangent space at $z$ to the manifold $\mathcal{M}$: $\exp_z : T_z\mathcal{M} \to \mathcal{M}$. Here, the Riemannian exponential map is constructed as i) Choose a tangent vector $v \in T_z\mathcal{M}$. ii) Consider the unique geodesic $\gamma_v(t)$ emanating from $z$ with initial velocity $v$. Formally, $\gamma_v(t)$ satisfies $\gamma_v(0) = z$ and $\gamma_v'(0) = v$. iii) The exponential map sends the tangent vector $v$ to the point on the manifold reached by traveling along the geodesic $\gamma_v$ for unit time, $\exp_z(v) = \gamma_v(1)$.*

# 3 Theory

See Appendix B for complete proofs of all statements.

## 3.1 Key Lemmas

Here, we summarize key lemmas required for our study. These results follow those of previous studies [43, 23, 24], but are presented below for the sake of uniformity of notation and to keep the manuscript self-contained. First, it can be shown that in $\mathrm{CAT}(K)$ spaces with $K \leq 0$, the convexity properties ensure the existence and uniqueness of the Fréchet mean under mild conditions. For $\mathrm{CAT}(K)$ spaces with $K > 0$, additional constraints on the diameter of the space may be necessary to ensure uniqueness due to potential multiple minima arising from positive curvature.

**Lemma 1.** *Let $(\mathcal{M}, d)$ be a $\mathrm{CAT}(K)$ space for $K \leq 0$. For any fixed point $p \in \mathcal{M}$, the function $f : \mathcal{M} \to \mathbb{R}$ defined by $f(x) = d^2(p, x)$ is geodesically convex.*

Lemma 1 establishes that the squared distance function retains geodesic convexity in $\mathrm{CAT}(K)$ spaces with non-positive curvature. This property is fundamental because it ensures that the Fréchet functional, which aggregates squared distances, inherits convexity. Consequently, optimization procedures to find the Fréchet mean are well-behaved, avoiding local minima and guaranteeing global optimality under the given conditions.

**Lemma 2.** *Let $(\mathcal{M}, d)$ be a complete $\mathrm{CAT}(K)$ space. For any probability measure $\nu$ on $\mathcal{M}$ with compact support, there exists at least one minimizer $m \in \mathcal{M}$ of the Fréchet functional:*

$$m = \underset{x \in \mathcal{M}}{\arg\min} \int_{\mathcal{M}} d^2(y, x) d\nu(y).$$

**Lemma 3.** *Let $(\mathcal{M}, d)$ be a $\mathrm{CAT}(K)$ space with $K \leq 0$ that is strictly geodesically convex, meaning that the squared distance function $f(x) = d^2(p, x)$ is strictly geodesically convex for any fixed point $p \in \mathcal{M}$. Then, for any probability measure $\nu$ on $\mathcal{M}$ with compact support, the Fréchet mean $m$ is unique.*

Based on Lemma 1, which ensures geodesic convexity of the squared distance function in non-positively curved $\mathrm{CAT}(K)$ spaces, and Lemma 2, which guarantees the existence of a Fréchet mean under compact support, one can establish the stability of the Fréchet mean under measure perturbations. Furthermore, Lemma 3 ensures uniqueness under strict geodesic convexity, thereby enabling Proposition 1 to assert the convergence of Fréchet means in non-positively curved spaces.

**Proposition 1.** *Let $(\mathcal{M}, d)$ be a $\mathrm{CAT}(K)$ space with $K \leq 0$. Suppose $\{\nu_n\}$ is a sequence of probability measures on $\mathcal{M}$ that converges weakly to a probability measure $\nu$. Assume that for each $n$, the measure $\nu_n$ has a unique Fréchet mean $m_n$, and $\nu$ also has a unique Fréchet mean $m$. Then, the sequence of Fréchet means $\{m_n\}$ converges to $m \in \mathcal{M}$.*

Proposition 1 claims that the $\mathrm{CAT}(K)$ condition with $K \leq 0$ ensures that the space is non-positively curved, which imbues the space with strict convexity properties crucial for the uniqueness and stability of minimizers. This geometric structure prevents the existence of multiple local minima, thereby facilitating the continuity of minimizers under perturbations of the measure. Here, the stability of the Fréchet mean under measure perturbations is foundational for Fréchet regression. It ensures that as predictors vary and induce changes in the conditional distributions of responses, the conditional Fréchet means (regression estimates) behave predictably and converge appropriately as sample size increases.

**Lemma 4.** *Let $(\mathcal{M}, d)$ be a $\mathrm{CAT}(K)$ space with positive curvature bound $K > 0$. If the diameter of the support of the probability measure $\nu$, denoted by $\mathrm{diam}(\mathrm{supp}(\nu))$, satisfies $\mathrm{diam}(\mathrm{supp}(\nu)) < \frac{\pi}{2\sqrt{K}}$, then the Fréchet mean $m$ of $\nu$ is unique.*

127 In Lemma 4, the diameter constraint ensures that all points in the support of $\nu$ lie within a geodesic
128 ball of radius $R = \pi/2\sqrt{K}$. In $\mathrm{CAT}(K)$ spaces with $K > 0$, such balls are geodesically convex,
129 meaning any geodesic between two points within the ball lies entirely inside the ball. This local
130 convexity is crucial for preserving strict convexity properties of the Fréchet functional.

131 In addition, applying Lemmas 2 and 3, the following statement can be obtained.

132 **Lemma 5.** *Let $(\mathcal{M}, d)$ be a complete $\mathrm{CAT}(K)$ space and consider a conditional distribution $\nu_x$ of*
133 $Y$ *given* $X = x$. *If for each* $x$, *the support of* $\nu_x$ *satisfies*

$$\mathrm{diam}(\mathrm{supp}(\nu_x)) < D_K = \begin{cases} +\infty & \text{if } K \leq 0, \\ \frac{\pi}{\sqrt{K}} & \text{if } K > 0, \end{cases}$$

134 *then then the conditional Fréchet mean in Eq.* (1) *exists and is unique for each* $x$.

## 3.2 Convergence Rates and Concentration

136 Let $\hat{\mu}_n^*$ denote a nonparametric Fréchet regression estimator (e.g., Nadaraya–Watson–type kernel
137 smoothing [32, 40, 8] on the predictor space). Then, the following statements for the concentration
138 results, the pointwise consistency, and rates of convergence can be obtained. The important point is
139 that one has to rely on exponential concentration inequalities valid in $\mathrm{CAT}(K)$ spaces (e.g., specific
140 versions of concentration of measure or deviation bounds for Fréchet means).

141 **Theorem 1** (Concentration for the sample Fréchet mean). *Let $(\mathcal{M}, d)$ be a complete $\mathrm{CAT}(K)$ space*
142 *of diameter at most $D$. Suppose that $Y_1, Y_2, \ldots, Y_n$ are independent and identically distributed*
143 *random points in $\mathcal{M}$, and let $\mu$ and $\hat{\mu}_n$ be the population and sample Fréchet mean.*

$$\mu := \arg\min_{z \in \mathcal{M}} \mathbb{E}[d^2(Y, z)],$$

$$\hat{\mu} := \arg\min_{z \in \mathcal{M}} \frac{1}{n} \sum_{i=1}^{n} d^2(Y_i, z).$$

144 *Assume further that each $d^2(Y_i, z)$ is essentially bounded by $D^2$, or more generally that $d^2(Y_i, z)$*
145 *has sub-Gaussian tails uniformly in $z$. Then there exists $\delta > 0$ such that for every $\epsilon > 0$,*

$$\mathbb{P}\left[d(\hat{\mu}, \mu) > \epsilon\right] \leq 2 \left(\frac{\alpha(K, D)D}{\delta}\right)^m e^{-\frac{n(\alpha(K,D)\epsilon^2)^2}{8D^2}}, \tag{4}$$

146 *where $m$ is the dimension of the manifold, and $\alpha(K, D)$ is the strong convexity constant.*

147 In addition to the concentration for the sample Fréchet mean in the standard sense, the following
148 proposition gives the concentration in $L_p$ sense.

149 **Proposition 2.** *Under the hypotheses of Theorem 1, there exist explicit constants $C_p(K, D)$ such*
150 *that for any integer $n \geq 1$ and $p \geq 1$,*

$$\mathbb{E}[d^p(\hat{\mu}_n, \mu)] \leq C_p(K, D)(n^{-p/2}). \tag{5}$$

151 *That is, $d(\hat{\mu}_n, \mu)$ converges to 0 in $L^p$ at a rate on the order of $n^{-p/2}$.*

152 Moreover, the following theorem gives the pointwise consistency of nonparametric Fréchet regression
153 in a $\mathrm{CAT}(K)$ space. The main idea parallels classical kernel-based regression arguments in $\mathbb{R}^d$, but
154 replaces ordinary arithmetic means by Fréchet means in the metric space $(\mathcal{M}, d)$.

155 **Assumption 1** (Kernel LLN condition). *For any bounded (or square-integrable) function $f: \mathcal{M} \to \mathbb{R}$,*
156 *nonnegative weights $\{w_{n,i}(x)\}_{i=1}^{n}$ satisfies*

$$\sum_{i=1}^{n} w_{n,i}(x)f(Y_i) \overset{a.s.}{\underset{n \to \infty}{\to}} \mathbb{E}[f(x) \mid X = x]. \tag{6}$$

157 **Theorem 2** (Pointwise consistency of nonparametric Fréchet regression). *Let $\{(X_i, Y_i)\}_{i=1}^{n}$ be i.i.d.*
158 *sample with $X_i \in \mathbb{R}^d$ and $Y_i \in \mathcal{M}$, where $(\mathcal{M}, d)$ is a complete $\mathrm{CAT}(K)$ space with diameter*
159 $\mathrm{diam}(\mathcal{M}) \leq D$. *Define the population Fréchet regression function:*

$$\mu^*(x) := \arg\min_{z \in \mathcal{M}} \mathbb{E}[d^2(Y, z) \mid X = x].$$

Assume that $\mu^*(x)$ is well-defined and unique for each $x$, provided as Theorem 5 Also, let $\{w_{n,i}(x)\}_{i=1}^n$ be nonnegative weights that sum to $1$ for each fixed $x$. For instance, in kernel regression, one sets

$$w_{n,i}(x) = \frac{W(\|x - X_i\|/h_n)}{\sum_{j=1}^n W(\|x - X_j\|/h_n)},$$

where $W(\cdot)$ is a usual kernel (with compact support or exponential decay), and $h_n \to 0$ is a bandwidth. Define the nonparametric Fréchet-regression estimator at $x$ by

$$\hat{\mu}_n^*(x) = \arg\min_{z \in \mathcal{M}} \sum_{i=1}^n w_{n,i}(x) d^2(Y_i, z). \tag{7}$$

Then, under mild regularity conditions on the weights in Assumption 1, $\hat{\mu}_n^*(x) \overset{a.s.}{\underset{n\to\infty}{\to}} \mu^*(x)$, for each fixed $x \in \mathbb{R}^d$.

Here, additional assumptions allow us to obtain the convergence rates in $\mathrm{CAT}(K)$ spaces.

**Theorem 3** (Convergence rates in $\mathrm{CAT}(K)$ spaces)**.** *Under the assumptions of Theorem 2, suppose additionally:*

- *$\mu^*: \mathbb{R}^d \to \mathcal{M}$ is $\beta$-Hölder (or Lipschitz) continuous, with respect to the usual Euclidean norm on $\mathbb{R}^d$ and the distance $d$ on $\mathrm{CAT}(K)$. That is, there exists $L > 0$ and $\beta > 0$ such that*

$$d(\mu^*(x), \mu^*(x')) \leq L \cdot \|x - x'\|^\beta, \tag{8}$$

  *for all $x, x' \in \mathbb{R}^d$.*

- *The kernel weights $w_{n,i}(x)$ satisfy standard nonparametric conditions:*

$$\sum_{i=1}^n w_{n,i}(x) = 1, \; w_{n,i}(x) \approx W\left(\frac{\|x - X_i\|}{h_n}\right), \qquad h_n \to 0, \quad nh_n^d \to +\infty. \tag{9}$$

- *Each conditional distribution $Y \mid X = x$ has finite second moments in the $\mathrm{CAT}(K)$ space and a unique Fréchet mean $\mu^*(x)$.*

- *The distribution of $Y \mid X = x$ varies smoothly in a local neighborhood of $x$. Formally, one assumes that for $x'$ near $x$, the conditional distributions $\mathbb{P}[Y \in \cdot \mid X = x']$ do not differ too much, ensuring small bias when $x' \approx x$.*

*Then for the nonparametric Fréchet regression estimator $\hat{\mu}_n^*$,*

$$\sup_{x \in \mathcal{X}_0} \mathbb{E}\left[d^2(\hat{\mu}_n^*(x), \mu^*(x))\right] = O\left(\frac{1}{nh_n^d} + h_n^{2\beta}\right), \tag{10}$$

*where $\mathcal{X}_0 \subseteq \mathbb{R}^d$ is any compact subset over which the kernel is applied.*

From the above theorem, one can see that the usual $\left(\frac{1}{nh_n^d} + h_n^\beta\right)$ trade-off from Euclidean nonparametric statistics carries over to the $\mathrm{CAT}(K)$ setting, once one accounts for i) geodesic convexity for controlling variance and ii) the Hölder continuity of $\mu^*(x)$ for controlling bias.

**Implications:** Section 3.2 provides the statistical properties of Fréchet regression estimators within $\mathrm{CAT}(K)$ spaces. Theorem 1 offers exponential concentration bounds for the sample Fréchet mean, indicating that the estimator converges to the true mean with high probability as the sample size increases. Proposition 2 further quantifies this convergence in an $L^p$ sense, demonstrating that the expected distance between the sample and population Fréchet means decreases at a rate proportional to $n^{-1/2}$. These results are pivotal for understanding the efficiency and reliability of Fréchet regression estimators. They assure that given sufficient data, the regression estimates will not only be consistent but also achieve convergence rates comparable to those observed in classical Euclidean nonparametric regression.

## 3.3 Angle Stability for Conditional Fréchet Means

Understanding not just the position but also the directional relationships around the Fréchet mean is crucial for capturing the local geometry of the data distribution. Angle stability ensures that small perturbations in the underlying probability measures or data configurations do not lead to significant distortions in the angular relationships among points relative to the Fréchet mean. This property is particularly valuable when analyzing directional data or when the regression function's local behavior depends on angular relationships, such as shape analysis or directional statistics.

First, the following lemma for the angle comparison in $\mathrm{CAT}(K)$ spaces is provided.

**Lemma 6.** *Let $(\mathcal{M}, d)$ be a $\mathrm{CAT}(K)$ space, and let $\triangle xyz \subset \mathcal{M}$ be a geodesic triangle of perimeter $\leq \pi/\sqrt{K}$ when $K > 0$. Let $\triangle \bar{x}\bar{y}\bar{z}$ be its comparison triangle in the simply connected model space of constant curvature $K$. Then for each vertex $x$ and the corresponding comparison vertex $\bar{x}$, $\angle_x(y, z) \leq \angle_{\bar{x}}(\bar{y}, \bar{z})$, where $\angle_x(y, z)$ is the Alexandrov angle (or geodesic angle) at $x$ formed by the geodesic segments $[xy]$ and $[xz]$.*

Note the assumption that the perimeter of $\triangle xyz$ is $\leq \pi/\sqrt{K}$ (when $K > 0$) is used to ensure i) The geodesics $[xy], [yz], [zx]$ are short enough so that the entire triangle $\triangle xyz$ (and sub-triangles $\triangle xy'z'$) can be compared in the standard simply connected model space (the sphere of radius $1/\sqrt{K}$ if $K > 0$). ii) One avoids the potential degeneracy where side lengths might exceed $\pi/\sqrt{K}$, which could cause the model triangle in spherical geometry to become ambiguous or wrap around the sphere. In the case $K \leq 0$, there is no maximum perimeter restriction because the simply connected model space (Euclidean or hyperbolic) is unbounded in diameter.

Next, the lemma for the angle continuity under small perturbation is provided.

**Lemma 7.** *Let $\triangle pqr$ and $\triangle p'q'r'$ be two geodesic triangles in a $\mathrm{CAT}(K)$ space $(\mathcal{M}, d)$. Suppose each has a perimeter $\pi/\sqrt{K}$ when $K > 0$ (no restriction is needed if $K \leq 0$). Also assume $d(p, p') + d(q, q') + d(r, r')$ is small. Then, for the angles at $p$ in $\triangle pqr$ and at $p'$ in $\triangle p'q'r'$,*

$$|\angle_p(q, r) - \angle_{p'}(q', r')| \leq C\delta_{pp'qq'rr'}, \tag{11}$$

*where $C > 0$ is a constant depending only on $K$ and the maximum side length (or perimeter) constraints, and*

$$\delta_{pp'qq'rr'} := d(p, p') + d(q, q') + d(r, r'). \tag{12}$$

Based on the above lemmas, the following statements are obtained.

**Proposition 3** (Angle perturbation via conditional measures)**.** *Let $\{\nu_x\}$ be a family of probability measures on a $\mathrm{CAT}(K)$ space $(\mathcal{M}, d)$, each supported in a geodesic ball of diameter $\leq D = \pi/2\sqrt{K}$ when $K > 0$. Let $\mu^*(x)$ be the unique Fréchet mean of $\nu_x$. Suppose $\nu_x$ and $\nu_{x'}$ are close in the Wasserstein metric on measures: $d_W(\nu_x, \nu_{x'}) \leq \epsilon$. Then, for any fixed $u, v \in \mathcal{M}$, one has*

$$|\angle_{\mu^*(x)}(u, v) - \angle_{\mu^*(x')}(u, v)| \leq C\epsilon,$$

*where the constant $C > 0$ depends on the strong-convexity modulus $\alpha(K, D)$. In particular, smaller $\epsilon$ implies the angles at $\mu^*(x)$ and $\mu^*(x')$ to points $u, v$ differ by at most $O(\epsilon)$.*

**Theorem 4** (Angle stability for conditional Fréchet means)**.** *Let $\{(X_i, Y_i)\} \subset \mathbb{R}^d \times \mathcal{M}$ with $\mathcal{M}$ a $\mathrm{CAT}(K)$ space of diameter $\leq D = \pi/2\sqrt{K}$ if $K > 0$. For each $x \in \mathbb{R}^d$, let $\nu_x(\cdot)$ be the conditional distribution of $Y$ given $X = x$. Assume each $\nu_x$ has the unique Fréchet mean $\mu^*(x)$. Moreover, suppose that for $x, x'$ sufficiently close, the measures $\mu^*(x)$ and $\mu^*(x')$ differ by at most $\epsilon(\|x - x'\|)$ in the Wasserstein distance. Then for any finite set of points $\{u_1, \ldots, u_m\} \subset \mathcal{M}$,*

$$\sup_{1 \leq i < j \leq m} |\angle_{\mu^*(x)}(u_i, u_j) - \angle_{\mu^*(x')}(u_i, u_j)| \leq C\epsilon_{xx'},$$

*where $C > 0$ is a constant depending on the strong-convexity modulus $\alpha(K, D)$ and $\epsilon_{xx'} = \epsilon(\|x - x'\|)$. Thus, all angles at $\mu^*(x)$ relative to a finite set of directions $u_1, \ldots, u_m$ vary continuously and Lipschitzly with $x$.*

**Implications:** The established angle stability results in Section 3.3 imply that the geometric structure surrounding the conditional Fréchet mean remains consistent under minor changes in the data distribution. This consistency is essential for applications where the relative orientation of data points carries meaningful information, ensuring that the regression estimates preserve intrinsic geometric relationships.

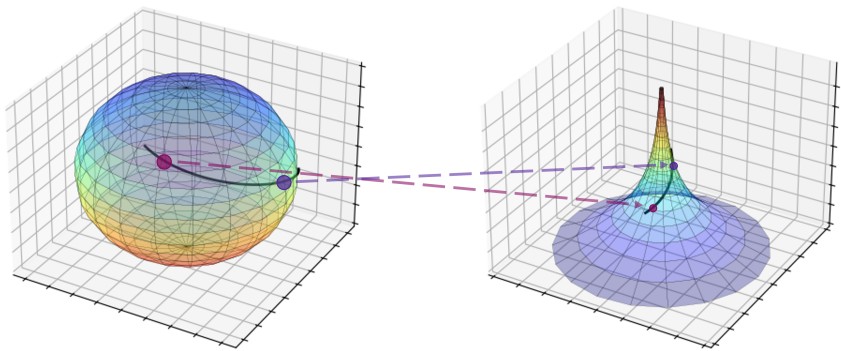

Figure 1: Mapping from spherical data into hyperbolic space.

### 3.4 Local Jet Expansion of Fréchet Functionals

**Lemma 8.** *Let $z \in \mathcal{M}$ and let $\exp_z \colon T_z\mathcal{M} \to \mathcal{M}$ be the Riemannian exponential map (in a local sense if $\mathcal{M}$ is a manifold, or a suitable geodesic parameterization if $\mathcal{M}$ is just a geodesic metric space). Then for points $u, v$ sufficiently close to $z$, define $U := \exp_z^{-1}(u)$ and $V := \exp_z^{-1}(v)$. Then,*

$$\angle_z(u,v) = \angle_0(U,V) + O(\|\exp_z^{-1}(u)\|^2 + \|\exp_z^{-1}(v)\|^2),$$

*where $\angle_0(U,V)$ is the standard Euclidean angle in $T_z\mathcal{M} \approx \mathbb{R}^m$, and the big-Oh term depends on curvature bounds near $z$.*

**Proposition 4** (Local Jet expansion of Fréchet functionals). *Let $\nu$ be a probability measure on a sufficiently regular $\mathrm{CAT}(K)$ space $(\mathcal{M}, d)$. Suppose that $\mu(x)$ is the Fréchet mean of $\nu_x$: $\mu(x) := \arg\min_{z \in \mathcal{M}} \int d^2(y, z) d\nu_x(y)$, and consider the Fréchet functional $F_x(z) = \int d^2(y, z) d\nu_x(y)$. Then, in a sufficiently small neighborhood of $\mu$, the functional $F$ can be expanded in the tangent space $T_\mu\mathcal{M}$ via the exponential map. Specifically, using local coordinates $\exp_\mu \colon T_\mu\mathcal{M} \supset B_r(0) \to \mathcal{M}$, for a vector $v$ with $\|v\|$ small, define $z = \exp_\mu(v)$. The expansion is given by*

$$F(\exp_\mu(v)) = F_x(\mu) + \langle \nabla F_x(\mu), v \rangle + \frac{1}{2}\langle H_x v, v \rangle + R(v),$$

*where $\nabla F_x(\mu)$ is the gradient (which is zero if $\mu$ is the unique minimizer), $H_x$ is the Hessian (a linear operator on $T_\mu\mathcal{M}$), and the remainder term $R(v)$ satisfies $|R(v)| = O(\|v\|^3)$.*

**Implications:** The analysis in Section 3.4 offers a nuanced understanding of the Fréchet functional's local behavior around its minimizer, the Fréchet mean. By expanding the Fréchet functional in the tangent space via the exponential map, one can gain insights into the functional's curvature and higher-order properties.

### 3.5 Auxiliary Statements

Here, a couple of auxiliary propositions that facilitate a deeper understanding of the structural properties of the Fréchet functional within $\mathrm{CAT}(K)$ spaces are introduced in this section. These propositions decompose the Fréchet functional into radial and angular components, enabling a more nuanced analysis of variance and stability around the Fréchet mean.

**Proposition 5** (Angle Splitting in Distance Sums). *Consider the Fréchet functional $F(z) = \int d^2(y, z) d\nu(y)$. For $z$ near $\mu^*$, decompose:*

$$d^2(y,z) = d^2(y, \mu^*) + \Pi_d(y, z, \mu^*) + \Pi_\angle(y, z, \mu^*),$$

*where $\Pi_d$ captures radial changes in distances $\Pi_\angle$ represents angular corrections around $\mu^*$. If $\angle_{\mu^*}(y, z)$ remains small near $\mu^*$, then $\Pi_\angle$ is of order $\langle \angle_{\mu^*}(y, z)\rangle d(\mu^*, z)$.*

**Proposition 6** (Angle–Distance Decomposition of Conditional Variance). *Let $\nu_x$ be the conditional distribution of $Y$ given $X = x$ on a sufficiently smooth $\mathrm{CAT}(K)$ space $(\mathcal{M}, d)$. Suppose $\mu^*(x)$ is*

| Data manifold | Mean squared error (MSE) |
|---|---|
| Sphere ($K = 1$) | $0.4915(\pm 0.0086)$ |
| Hyperbolic ($K = -1$) | $0.4228(\pm 0.0021)$ |

Table 1: Evaluation of Fréchet regression on different spaces.

the unique Fréchet mean of $\nu_x$. Around $\mu^*(x)$, let

$$R_x(y) := d(y, \mu^*(x)), \quad \phi_x(y) := \angle_{\mu^*(x)}(u_0, y), \tag{13}$$

for a fixed reference point $u_0 \in \mathcal{M}$. Then the conditional variance can be partially decomposed into a radial variance term, an angle–radial covariance term, and higher-order corrections:

$$\mathrm{Var}_{\nu_x}\left[d^2(Y, \mu^*(x))\right]$$
$$= \mathrm{Var}[A_x(Y)] + \mathrm{Cov}\left(\phi_x(Y), R_x(Y)^2\right) + \beta, \tag{14}$$

where $A_x$ is the radial part and $\beta$ is the higher-order term.

**Implications:** The auxiliary propositions presented in Subsection 3.5 play an important role in refining the theoretical underpinnings of Fréchet regression within $\mathrm{CAT}(K)$ spaces. By decomposing the Fréchet functional into radial and angular components, these propositions enable a more granular analysis of variance and stability around the Fréchet mean.

# 4 Experiments

From the discussion in Section 3, it can be seen that the negative curvature space has better properties in terms of estimation than the positive curvature space with broader support. To confirm these results, this section considers numerical experiments. See Appendix A for the intuitive understanding of the following hyperbolic mapping.

## 4.1 Illustrative Example

A point on the unit sphere is parameterized as $x = \sin(\phi)\cos(\theta)$, $y = \sin(\phi)\sin(\theta)$, $z = \cos(\phi)$, where $\phi \in [0, \pi]$ is the polar angle and $\theta \in [0, 2\pi]$ is the azimuthal angle. Let $R$ be the radius of the sphere. Here, consider the stereographic projection: The plane is tangent to the sphere at the south pole $(0, 0, -R)$ and is defined $z = -R$, and the north pole $N = (0, 0, R)$ serves as the projection point. For a point $p = (x, y, z)$, the stereographic projection $\pi(p) = (u, v)$ on the plane is given by $u = \frac{Rx}{R+z}$, $v = \frac{Ry}{R+z}$. This plane can be considered in the hyperbolic space, and one can visualize it as the pseudosphere (see Figure 1). Also, a point $(x, y, z)$ can be mapped back to the sphere as

$$x = \frac{2R^2 u}{R^2 + u^2 + v^2}, y = \frac{2R^2 v}{R^2 + u^2 + v^2}, z = R\frac{u^2 + v^2 - R^2}{R^2 + u^2 + v^2}.$$

See Appendix E (including Python code in Listing 2) for the detailed data-generating process.

Table 1 shows the evaluation results of Fréchet regression on the spherical and hyperbolic coordinates. It can be seen that the hyperbolic mapping yields better results. Note that, the previous studies [15, 16] reported the effectiveness of such mapping for statistical problems of spherical data, and the objective of experiments in this section is just to confirm the theoretical results.

## 4.2 Experiment on Real-world Dataset

In addition to the illustrative example, consider the experiments on the real-world datasets. This section uses the following: i) HYG Steller database [1], which is a comprehensive dataset containing information on stars brighter than magnitude 6.5. ii) USGS Earthquake catalogue [2], represented in spherical coordinates. iii) NOAA Climate data [3], from weather satellites. See Appendix 4.2 for the

---

[1] https://github.com/astronexus/HYG-Database?tab=readme-ov-file
[2] https://earthquake.usgs.gov/earthquakes/feed/v1.0/summary/2.5_week.csv
[3] http://celestrak.org/NORAD/elements/table.php?GROUP=weather&FORMAT=tle

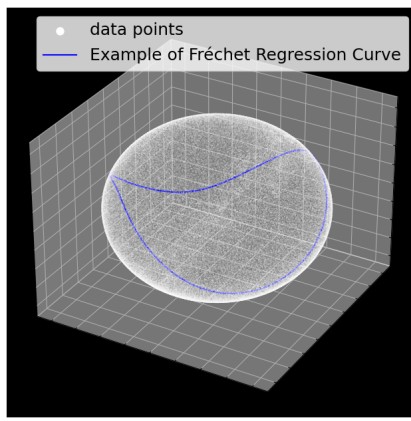
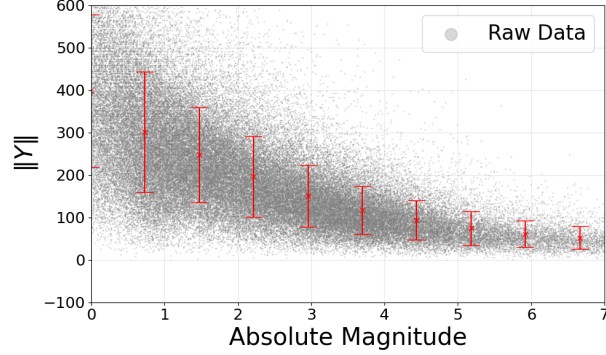

Figure 2: Visualization of the HYG Stellar database.

Figure 3: Heteroscedasticity in the HYG Stellar dataset.

| Dataset | MSE |
|---|---|
| HYG Stellar | $0.3765(\pm0.0036)$ |
| USGS Earthquake | $0.5832(\pm0.0831)$ |
| NOAA Climate | $0.4384(\pm0.0678)$ |
| HYG Stellar (hyperbolic) | $0.2660(\pm0.0032)$ |
| USGS Earthquake (hyperbolic) | $0.4743(\pm0.0541)$ |
| NOAA Climate (hyperbolic) | $0.3259(\pm0.0683)$ |

Table 2: Evaluation of Fréchet regression on different spaces.

details of this experiment (including Python code in Listing 3 for the visualization and data format check of the dataset). Table 2 shows the experimental results of Fréchet regression on different coordinates for the real datasets. The mapping procedure is the same as Section 4.1. As with the illustrative example, we can confirm that Fréchet regression on hyperbolic surfaces yields better results on the real datasets. As discussed in more detail in Appendix A, such a mapping of responses to hyperbolic space may be particularly useful when heteroscedasticity is assumed in the data. Indeed, heteroscedasticity can be observed in the HYG Stellar dataset (see Figure 3).

# 5 Conclusion

This study provides a comprehensive theoretical analysis of Fréchet regression within the framework of comparison geometry, focusing on $\mathrm{CAT}(K)$ spaces. It establishes foundational results on the existence, uniqueness, and stability of the Fréchet mean under varying curvature conditions. Notably, the analysis demonstrates how curvature properties influence statistical estimation, with non-positive curvature spaces offering advantageous stability and convergence properties. The paper also extends statistical guarantees to nonparametric Fréchet regression, including exponential concentration bounds and convergence rates, which align with classical Euclidean results. Angle stability and local jet expansion further highlight the behavior of Fréchet functionals, offering geometric insights of regression in non-Euclidean spaces. Experimental results support the theoretical findings, showing that hyperbolic mappings often improve performance under heteroscedasticity assumption.

**Limitations:** While this study provides a robust theoretical foundation for Fréchet regression in $\mathrm{CAT}(K)$ spaces, several limitations exist. Firstly, the analysis predominantly focuses on spaces with constant curvature bounds, which may not encompass all practical scenarios where data resides in more heterogeneous geometric contexts. Additionally, the reliance on strong convexity conditions and diameter constraints in positively curved spaces may restrict the applicability of the results. As has been done in the information geometry framework [1, 34, 10, 25, 26, 31, 2], future work could explore relaxing assumptions, extending the framework to broader classes of metric spaces, and developing efficient algorithms.

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

# A Intuitive Understanding for Hyperbolic Mapping

In regression analysis, transforming the response variable can often lead to improved model performance by stabilizing variance, normalizing distributions, or linearizing relationships. A classical example is the logarithmic transformation $Y \mapsto \log(Y)$ which can enhance the performance of a linear regression model under certain conditions. Similarly, mapping spherical responses into hyperbolic space can offer analogous benefits, particularly in scenarios where the data exhibits inherent geometric or hierarchical structures.

**Log Transformation in Linear Regression**  Consider the simple linear regression model:

$$Y = \beta X + \epsilon,$$

where $Y$ is the response variable, $X$ is the predictor, $\beta$ is the regression coefficient, and $\epsilon$ is the error term with $\mathbb{E}[\epsilon] = 0$ and $\mathrm{Var}(\epsilon) = \sigma^2$. Applying a logarithmic transformation to $Y$ yields

$$\log(Y) = \beta X + \epsilon,$$
$$Y = \exp(\beta X + \epsilon) = \exp(\beta X) \cdot \exp(\epsilon).$$

Assuming $\epsilon$ is small and approximately normally distributed, $\exp(\epsilon)$ introduces multiplicative noise to $Y$ effectively stabilizing variance across different levels of $X$. This transformation often reduces heteroscedasticity in the residuals, leading to improved regression performance. Here, the heteroscedasticity refers to the phenomenon where the variability of the errors (or residuals) in a regression model is not constant across the range of predictor variables.

**Definition 8** (Heteroscedasticity). *Consider a regression model:*

$$Y_i = \beta X_i + \epsilon_i,$$

*where $\epsilon_i \sim \mathcal{N}(0, \sigma^2(X_i))$. Here, the variance of the error term $\sigma^2(X)$ depends on $X$. In a heteroscedastic model, the variance of $\epsilon_i$ is a function of the predictors $X_i$:*

$$\mathrm{Var}(\epsilon_i \mid X_i) = \sigma^2(X_i).$$

*In contrast, for homoscedasticity, the variance of $\epsilon_i$ is constant.*

**Hyperbolic Mapping via Stereographic Projection**  Analogous to the log transformation, hyperbolic mapping transforms the response variable into a space where the geometric structure can lead to improved regression characteristics. The procedure involves mapping points from a spherical representation to a hyperbolic plane using stereographic projection. A point on the unit sphere of radius $R$ is parameterized using spherical coordinates:

$$x = R \sin(\phi) \cos(\theta),$$
$$y = R \sin(\phi) \sin(\theta),$$
$$z = R \cos(\phi),$$

where $\phi \in [0, \pi]$ is the polar angle and $\theta \in [0, 2\pi)$ is the azimuthal angle. The stereographic projection maps a point $p = (x, y, z)$ on the sphere to a point $p \mapsto \psi(p) = (u, v)$ on the plane tangent to the sphere at the south pole $(0, 0, -R)$ and defined by $z = -R$. The north pole $N = (0, 0, R)$ serves as the projection point. The projection formulas are

$$u = \frac{Rx}{R + z},$$
$$v = \frac{Ry}{R + z}.$$

This plane can be interpreted as a model of hyperbolic space, specifically visualized as a pseudosphere, which inherently possesses properties conducive to handling hierarchical or tree-like data structures.

Both the logarithmic transformation and hyperbolic mapping aim to stabilize variance and linearize relationships, through different geometric transformations. To understand the benefits of hyperbolic mapping, consider the effect of each transformation on the variance of the response variable. Starting with $Y = \beta X + \epsilon$, applying the log transformation yields

$$\log Y = \beta X + \epsilon.$$

451 Assuming $\epsilon \sim \mathcal{N}(0, \sigma^2)$, The variance of $\log Y$ remains $\sigma^2$ which can be advantageous if the original
452 $Y$ exhibits multiplicative noise:

$$\text{Var}(Y) = \text{Var}(\exp(\beta X + \epsilon)) = \exp(2\beta X) \cdot \left(\exp(\sigma^2) - 1\right).$$

453 The transformation effectively decouples the variance from $X$ stabilizing it across different predictor
454 values.

455 For hyperbolic mapping, consider a response variable represented as a point on the sphere. The
456 stereographic projection transforms this spherical representation into the hyperbolic plane. Let
457 $Y$ be the original response mapped to a point $p = (x, y, z)$ on the sphere, and $\psi(p) = (u, v)$ its
458 hyperbolic projection. Assuming small deviations around a mean direction, the hyperbolic mapping
459 can linearize angular variations similarly to how the log transformation linearizes multiplicative
460 variations. Specifically, fluctuations in $Y$ around the mean direction correspond to additive noise
461 in the hyperbolic plane, potentially reducing variance in a manner akin to the log transformation.
462 Formally, if $Y$ is modeled on the sphere with

$$Y = R \cdot p + \epsilon,$$

463 where $\epsilon$ represents angular noise, the hyperbolic projection yields

$$\psi(Y) = \left(\frac{Rx}{R+z}, \frac{Ry}{R+z}\right) + \epsilon',$$

464 whre $\epsilon'$ is the transformed noise. Under specific conditions (e.g., small angular deviations), $\epsilon'$
465 exhibits reduced variance compared to $\epsilon$, analogous to the variance stabilization achieved by the log
466 transformation.

467 **Example 1** (Stabilizing Variance in Hierarchical Data). *Consider a dataset where the response*
468 *variable $Y$ represents hierarchical relationships, such as the popularity of topics in a taxonomy. The*
469 *inherent tree-like structure implies that differences between nodes (topics) grow exponentially with*
470 *depth. Direct regression on $Y$ would face increasing variance as depth increases. By mapping $Y$ into*
471 *hyperbolic space via stereographic projection, the exponential growth inherent in hierarchical data*
472 *is linearized. This transformation stabilizes variance across different levels of the hierarchy, enabling*
473 *more effective regression modeling. Specifically, the hyperbolic mapping aligns the geometric*
474 *properties of the data with the regression framework, similar to how the log transformation aligns*
475 *multiplicative relationships with additive modeling.*

476 *Let $Y$ be mapped to hyperbolic space via stereographic projection:*

$$u = \frac{Rx}{R+z},$$
$$v = \frac{Ry}{R+z}.$$

477 *Assuming $Y$ lies close to the north pole $N = (0, 0, R)$, small perturbations $\epsilon$ around $N$ imply*

$$z = R\cos(\phi) \approx R\left(1 - \frac{\phi^2}{2}\right),$$
$$x = R\sin(\phi)\cos(\theta) \approx R\phi\cos(\theta),$$
$$y = R\sin(\phi)\sin(\theta) \approx R\phi\sin(\theta).$$

478 *Substituting into the projection formulas,*

$$u \approx \frac{R \cdot R\phi\cos(\theta)}{R + R\left(1 - \frac{\phi^2}{2}\right)} = \frac{R^2\phi\cos(\theta)}{2R - \frac{\phi^2}{2}} \approx \frac{R\phi\cos(\theta)}{2},$$
$$v \approx \frac{R \cdot R\phi\sin(\theta)}{R + R\left(1 - \frac{\phi^2}{2}\right)} = \frac{R^2\phi\sin(\theta)}{2R - \frac{\phi^2}{2}} \approx \frac{R\phi\sin(\theta)}{2}.$$

479 *Thus, small angular deviations $\phi$ result in approximately linear changes in $u$ and $v$, effectively*
480 *reducing the variance from multiplicative to additive in the hyperbolic plane:*

$$\text{Var}(u, v) \approx \left(\frac{R}{2}\right)^2 \text{Var}(\phi).$$

481 *Compared to the original spherical variance* $\mathrm{Var}(\phi)$*, the hyperbolic mapping scales and linearizes the*
482 *variance, analogous to the stabilizing effect of the log transformation. Figure 4 shows the illustrative*
483 *example of transformed responses for* $Y = \beta X + \epsilon$ *with heteroscedastic errors* $\epsilon = \mathcal{N}(0, g(\sigma X))$,
484 $\sigma = 0.2$ *and* $\beta = 2$. *This figure shows* $g(\sigma X) = \sigma X$ *and* $g(\sigma X) = \exp(\sigma X)$ *cases.*

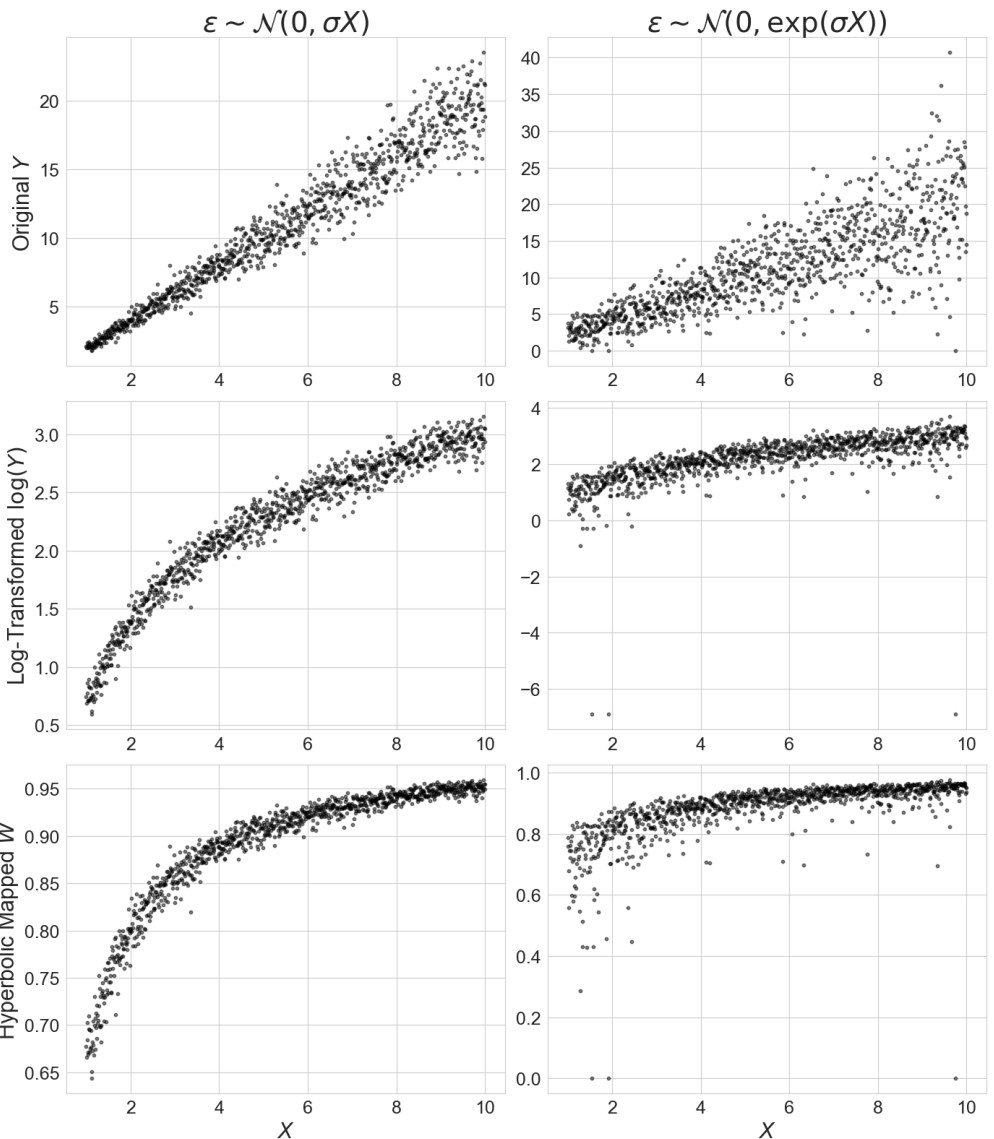

Figure 4: Illustrative example of transformed responses. Under the heteroscedastic errors assumption, the appropriate transformations of response variable yield stabilized variance. In this figure, $Y$ is the original response variables, $\log(Y)$ is the log-transformed variables and $W$ is the hyperbolic mapped variables.

 # B  Proofs

 ## B.1  Proofs for Section 3.1

*Proof for Lemma 1.* To establish the geodesic convexity of the squared distance function $f(x) = d^2(p,x)$ in a CAT($K$) space $(\mathcal{M}, d)$ with $K \leq 0$, one must show that for any two points $x, y \in \mathcal{M}$ and any geodesic $\gamma \colon [0,1] \to \mathcal{M}$ connecting $x$ to $y$, the function $t \mapsto f(\gamma(t))$ is convex on the interval $[0,1]$.

In the model space $\mathbb{M}^2_K$ of constant curvature $K \leq 0$, construct a comparison triangle $\bar{\triangle}$ corresponding to $\triangle = \{p, x, y\}$ in $\mathcal{M}$. Let $\bar{p}, \bar{x}, \bar{y}$ be the vertices of $\bar{\triangle}$ in $\mathbb{M}^2_K$ with side lengths matching those of $\triangle$. Then, for any points $a, b$ on the sides $[x,y]$ and $[p,x]$ or $[p,y]$, the distance $d(a,b)$ in $\mathcal{M}$ is at most the distance $d_{\mathbb{M}^2_K}(\bar{a}, \bar{b})$ in the model space.

Let $\gamma(t)$ corresponds to a point $\bar{\gamma}(t)$ on the side $[\bar{x}, \bar{y}]$ in $\bar{\triangle}$. By the CAT($K$) property,
$$d(p, \gamma(t)) \leq d_{\mathbb{M}^2_K}(\bar{p}, \bar{\gamma}(t)).$$

In $\mathbb{M}^2_K$, which is a uniquely geodesic space, the squared distance satisfies the law of cosines
$$d^2(\bar{p}, \bar{\gamma}(t)) \leq (1-t)d^2(\bar{p}, \bar{x}) + td^2(\bar{p}, \bar{y}) - t(1-t)c_K,$$
where $c_K$ is a non-negative constant dependent on $K$ and the geometry of the triangle. Here, since $K \leq 0$, the space $\mathbb{M}^2_K$ exhibits non-positive curvature, which implies that the term $-t(1-t)c_K$ does not negatively affect the inequality. Therefore,
$$d^2(p, \gamma(t)) \leq d^2_{(\mathbb{M}^2)K}(\bar{p}, \bar{\gamma}(t)) \leq (1-t)d^2(p,x) + td^2(p,y),$$
and $f$ is geodesically convex. $\qquad\square$

*Proof for Lemma 2.* Consider a sequence $\{x_n\}$ in $\mathcal{M}$ that converges to $x \in \mathcal{M}$. Given the continuity of the distance function in metric spaces, for each $y \in \mathcal{M}$, $d(y, x_n) \to d(y, x)$ as $n \to +\infty$. Since $d^2(y, x)$ is continuous in $x$, by Fatou's lemma,
$$\liminf_{n \to +\infty} d^2(y, x_n) \leq d^2(y, x).$$
Integrating both sides with respect to $\nu$,
$$\liminf_{n \to +\infty} \int_{\mathcal{M}} d^2(y, x_n)d\nu(y) \leq \int_{\mathcal{M}} d^2(y, x)d\nu(y).$$
Thus, $F$ is lower semicontinuous. Also, since
$$F(x) = \int_{\mathcal{M}} d^2(y, x)d\nu(y) \geq 0,$$
for any $x \in \mathcal{M}$, $F$ is bounded below by zero. Therefore, there exists a sequence $\{m_m\}$ in $\mathcal{M}$ such that
$$F(m_n) \to \inf_{x \in \mathcal{M}} F(x),$$
as $n \to +\infty$. Let $\{m_n\}$ be called a minimizing sequence. Given that the support of $\nu$, denoted by $\text{supp}(\nu)$, is compact, denote it by $S \subseteq \mathcal{M}$. That is, $S$ is compact and $\nu(S) = 1$.

To ensure that the existence of a convergent subsequence, one need to show that $\{m_n\}$ is contained within a compact subset of $\mathcal{M}$. Since $S$ is compact, it is bounded. Thus, there exists a radius $R > 0$ and a point $p \in \mathcal{M}$ such that $S \subseteq B(p, R)$, where $B(p, R) = \{x \in \mathcal{M} \mid d(p, x) \leq R\}$. Using the triangle inequality in metric spaces,
$$d(y, m_n) \geq d(p, m_n) - d(y, p) \geq d(p, m_n) - R.$$
Then,
$$\begin{aligned}
F(m_n) &= \int_S d^2(y, m_n)d\nu(y) \\
&\geq \int_S \{d(p, m_n) - d(y, p)\}^2 d\nu(y) \\
&= \int_S \{d(p, m_n)^2 - 2d(p, m_n) + d^2(y, p)\} d\nu(y) \\
&= d(p, m_n)^2 - 2d(p, m_n)\int_S d(y, p)d\nu(y) + \int_S d^2(y, p)d\nu(y) \leq C
\end{aligned}$$

515 Let $A = \int_S d(y, p)\nu(y)$ and $B = \int_S d^2(y, p)d\nu(y)$, both finite due to the compactness. Thus,

$$d(p, m_n)^2 - 2Ad(p, m_n) + B \leq C$$
$$d(p, m_n) \leq A \pm \sqrt{A^2 + C - B}.$$

516 Hence, the sequence $\{m_n\}$ lies within the closed ball $\overline{B}(p, A + \sqrt{A^2 + C - B})$, which is compact
517 if $\mathcal{M}$ is proper. Here, $\mathrm{CAT}(K)$ spaces are not necessarily proper in general, bu since $\mathrm{supp}(\nu)$ is
518 compact and $\{m_n\}$ is bounded, one can extract a convergent subsequence under the assumption
519 that $\mathcal{M}$ is complete. Given that $\{m_n\}$ is bounded and $\mathcal{M}$ is complete, one can utilize the Bolzano-
520 Weierstrass theorem in $\mathrm{CAT}(K)$ spaces to extract a convergent subsequence. Specifically, since $\mathcal{M}$
521 is a geodesic space and $\{m_n\}$ is bounded, there exists a subsequence $\{m_{n_k}\}$ that converges to some
522 $m \in \mathcal{M}$.

523 Since $F$ is lower semicontinuous and $m_{n_k} \to m$,

$$F(m) \leq \liminf_{k \to +\infty} F(m_{n_k}) = \inf_{x \in \mathcal{M}} F(x).$$

524 This implies that $m$ achieves the infimum of $F$,

$$F(m) = \inf_{x \in \mathcal{M}} F(x).$$

525 Therefore, $m$ is a minimizer of the Fréchet functional. □

526 *Proof for Lemma 3.* For the sake of contradiction, suppose that there are two distinct points $m_1, m_2 \in$
527 $\mathcal{M}$ such that both are minimizers of the Fréchet functional.

$$m_1 = \arg\min_{x \in \mathcal{M}} \int_{\mathcal{M}} d^2(y, x)d\nu(y),$$
$$m_2 = \arg\min_{x \in \mathcal{M}} \int_{\mathcal{M}} d^2(y, x)d\nu(y),$$

528 with $m_1 \neq m_2$. Since $\mathcal{M}$ is a $\mathrm{CAT}(K)$ space and thus a geodesic metric space, there exists a unique
529 geodesic $\gamma \colon [0, 1] \to \mathcal{M}$ connecting $m_1$ to $m_2$.

$$\gamma(0) = m_1,$$
$$\gamma(1) = m_2,$$
$$d(\gamma(t), \gamma(t')) = |t - t'| \cdot d(m_1, m_2), \quad \forall t, t' \in [0, 1].$$

530 Define a function $F \colon [0, 1] \to \mathbb{R}$ by evaluating the Fréchet functional along the geodesic $\gamma(t)$:

$$F(t) = \int_{\mathcal{M}} d^2(y, \gamma(t))d\nu(y).$$

531 Since both $m_1$ and $m_2$ are minimizers,

$$F(0) = F(1) = \inf_{x \in \mathcal{M}} F(x).$$

532 Given that $\mathcal{M}$ is strictly geodesically convex, the squared distance function $f(x) = d^2(y, x)$ is strictly
533 convex along any geodesic. Therefore, for each fixed $y \in \mathcal{M}$, the function $t \mapsto d^2(y, \gamma(t))$ satisfies

$$d^2(y, \gamma(t)) < (1 - t)d^2(y, m_1) + td^2(y, m_1),$$

534 for all $t \in (0, 1)$.

535 Integrate the strict inequality with respect to the measure $\nu$ yields

$$F(t) = \int_{\mathcal{M}} d^2(y, \gamma(t))d\nu(y)$$
$$< \int_{\mathcal{M}} \left\{ (1 - t)d^2(y, m_1) + td^2(y, m_2) \right\} d\nu(y)$$
$$= (1 - t) \int_{\mathcal{M}} d^2(y, m_1)d\nu(y) + t \int_{\mathcal{M}} d^2(y, m_2)d\nu(y).$$

536 But since $m_1$ and $m_2$ are both minimizers,

$$\int_{\mathcal{M}} d^2(y, m_1) d\nu(y) = \int_{\mathcal{M}} d^2(y, m_2) d\nu(y) = \int_{x \in \mathcal{M}} F(x).$$

537 Thus,

$$F(t) < (1-t) \inf_{x \in \mathcal{M}} F(x) + t \inf_{x \in \mathcal{M}} F(x) = \inf_{x \in \mathcal{M}} F(x).$$

538 However, this is a contradiction because $F(x)$ cannot be less than the infimum $\inf_{x \in \mathcal{M}} F(x)$. The
539 contradiction arises from the assumption that two distinct minimizers $m_1$ and $m_2$ exist. Therefore,
540 there can be at most one minimizer. Given that the Fréchet functional attains its infimum by Lemma 2,
541 this minimizer is unique. □

542 *Proof for Proposition 1.* The Fréchet functional $x \mapsto F_\nu(x)$ for a measure $\nu$ is defined as

$$F_\nu(x) = \int_{\mathcal{M}} d^2(y, x) d\nu(y).$$

543 Given that the squared distance function $d^2(y, x)$ is continuous in $y$ for each fixed $x$, weak conver-
544 gence $\nu_n \Rightarrow \nu$ implies that for each fixed $x \in \mathcal{M}$,

$$\lim_{n \to +\infty} F_{\nu_n}(x) = F_\nu(x).$$

545 In addition, given that $d^2(y, x)$ is continuous and bounded by zero, and assuming that the measures
546 $\nu_n$ and $\nu$ have compact supports, as established in Lemma 2, the convergence $\nu_n \Rightarrow \nu$ implies that

$$\lim_{n \to +\infty} F_{\nu_n}(x) = F_\nu(x), \quad \text{uniformly for } x \in \mathcal{M}.$$

547 This uniform convergence is a consequence of the boundedness of the squared distance function
548 over compact supports, and the equicontinuity provided by the geometric properties of the $\mathrm{CAT}(K)$
549 spaces.

550 Suppose that $m_n$ does not converge to $m$, Then, there exist an $\epsilon > 0$ and a subsequence $\{m_{n_k}\}$ such
551 that

$$d(m_{n_k}, m) \geq \epsilon,$$

552 for all $k$. Since $\mathcal{M}$ is a $\mathrm{CAT}(K)$ space with $K \leq 0$ and hence a geodesic and proper metric space
553 under the assumption of compact support from Lemma 2, the sequence $\{m_{n_k}\}$ has a convergent
554 subsequence. Without loss of generality, assume that $m_{n_k} \to m'$ as $k \to +\infty$. By the continuity of
555 the Fréchet functional,

$$\lim_{k \to +\infty} F_{\nu_{n_k}}(m_{n_k}) = \lim_{k \to +\infty} \inf_{x \in \mathcal{M}} F_{\nu_{n_k}}(x)$$
$$= F_\nu(m),$$

556 since $m$ is the unique minimizer for $\nu$.

557 Consider $\nu_n \Rightarrow \nu$ and $m_{n_k} \to m'$,

$$\lim_{k \to +\infty} F_{\nu_{n_k}}(m_{n_k}) = F_\nu(m').$$

558 Then,

$$F_\nu(m') = F_\nu(m).$$

559 Therefore, $m'$ is also a minimizer of $F_\nu(x)$. Since $\nu$ has a unique Fréchet mean $m$, it must be that
560 $m' = m$. Recall that $d(m_{n_k}, m) \geq \epsilon$ for all $k$, but $m_{n_k} \to m' = m$, which implies that

$$\lim_{k \to +\infty} d(m_{n_k}, m) = d(m', m) = 0,$$

561 contradicting $d(m_{n_k}, m) \geq \epsilon$. Therefore, it must be that

$$m_n \to m, \quad \text{as } n \to +\infty.$$

562 □

*Proof for Proposition 4.* For $K > 0$, the comparison space is the standard sphere $\mathbb{S}^n$ with radius $1/\sqrt{K}$. In $\mathbb{S}^n$, geodesics are great circles, and the distance between two points is given by the central angle multiplied by $1/\sqrt{K}$. The diameter of $\mathbb{S}^n$ is $\pi/\sqrt{K}$, meaning that the maximal distance between any two points is $\pi/\sqrt{K}$.

Given $R < \pi/2\sqrt{K}$, the geodesic ball $B(p, R)$ lies entirely within a hemisphere of $\mathbb{S}^n$. In this setting, any two points $x, y \in B(p, R)$ are separated by a distance $d(x, y)$, satisfying

$$
\begin{aligned}
d(x, y) &\leq d(x, p) + d(p, y) \\
&< \frac{\pi}{2\sqrt{K}} + \frac{\pi}{2\sqrt{K}} \\
&= \frac{\pi}{\sqrt{K}}.
\end{aligned}
$$

Since $d(x, y) < \pi/\sqrt{K}$, there exists a unique minimal geodesic connecting $x$ and $y$ within $\mathbb{S}^n$.

Assume, for contradiction, that the minimal geodesic $\gamma$ between $x$ and $y$ exits $B(p, R)$. Then, there exists a point $z \in \gamma$ such that $d(p, z) = R$. Consider the geodesic triagles $\triangle pzx$ and $\triangle pzy$. Since $d(p, x) < R$ and $d(p, y) < R$, and $\gamma$ is minimal, the angle at $p$ opposite the side $\gamma$ must satisfy certain angular constraints derived from the spherical law of cosines. However, because $R < \pi/2\sqrt{K}$, the triangle $\triangle pzx$ lies within a convex hemisphere, ensuring that the path from $p$ to $z$ to $x$ remains within $B(p, R)$. This contradicts the assumption that $\gamma$ exits $B(p, R)$. Therefore, since any two points in $B(p, R)$ can be connected by a unique minimal geodesic that remains entirely within $B(p, R)$, the geodesic ball $B(p, R)$ is geodesically convex in $\mathbb{S}^n$ for all radius $R < \pi/2\sqrt{K}$. This ensures that $\text{CAT}(K)$ condition preserves the strict convexity.

Given that $\text{diam}(\text{supp}(\nu)) < \pi/2\sqrt{K}$, for any geodesic $t \mapsto \gamma(t)$ connecting two distinct points $m_1, m_2 \in \mathcal{M}$, the Fréchet functional satisfies

$$
F(\gamma(t)) < (1 - t)F(m_1) + tF_2(m_2),
$$

for all $t \in (0, 1)$, provided $m_1 \neq m_2$. Here, strict convexity of $F(x)$ ensures that any local minimum is a global minimum, and further, that such a minimum is unique within the convex neighborhood. $\square$

 **B.2   Proofs for Section 3.2**

584   *Proof for Theorem 1.* Define the population Fréchet functional $F(z)$ and empirical Fréchet functional
585   $F_n(z)$ as follows.

$$F(z) := \mathbb{E}[d^2(Y, m)],$$

$$F_n(z) := \frac{1}{n} \sum_{i=1}^{n} d^2(Y_i, z).$$

586   By definition,

$$\mu = \arg\min_{z \in \mathcal{M}} F(z),$$

$$\hat{\mu}_n = \arg\min_{z \in \mathcal{M}} F_n(z).$$

587   Assume that $\mu$ is unique, which holds if $\mathrm{diam}(\mathcal{M}) < \pi/2\sqrt{K}$ when $K > 0$ or automatically if
588   $K \leq 0$, from Lemmas 2, 3 and Propositions 1, 4.

589   A key geometric fact in $\mathrm{CAT}(K)$ spaces is that the map

$$z \mapsto \mathbb{E}[d^2(Y, z)] = F(z)$$

590   is $\lambda$-strongly geodesically convex around $\mu$, provided $\mathrm{diam}(\mathcal{M})$ is small enough. Concretely, there
591   exists a constant

$$\alpha = \alpha(K, D) > 0,$$

592   such that for every $z \in \mathcal{M}$,

$$F(z) - F(\mu) \geq \alpha d^2(z, \mu).$$

593   A fully explicit formula for $\alpha(K, D)$ can be extracted from standard $\mathrm{CAT}(K)$ lemmas.

594   • If $K \leq 0$, one can take $\alpha(K, D) = \frac{1}{2}$. Indeed, $\mathrm{CAT}(K)$ spaces are sometimes called
595     Hadamard spaces, for which $d^2(y, \cdot)$ is 1-convex along geodesics.

596   • If $K > 0$ but $\mathrm{diam}(\mathcal{M}) = D < \pi/2\sqrt{K}$, one obtains an explicit lower bound

$$\alpha(K, D) \geq \frac{\sin(2\sqrt{K}R)}{2R},$$

597   where $R = D/2$. One often sees, for example,

$$\alpha(K, D) = \frac{2}{\pi} \sqrt{K} \sin\left(\frac{\pi}{2} - \sqrt{K}D\right).$$

598   Since $\hat{\mu}_n$ is the minimizer of $F_n$, one can obtain

$$F_n(\hat{\mu}_n) \leq F_n(\mu).$$

599   Here, rewriting $F_n = F_n - F + F$,

$$F_n(\hat{\mu}_n) - F_n(\mu) = \{F_n(\hat{\mu}_n) - F(\hat{\mu}_n)\} - \{F_n(\mu) - F(\mu)\} + \{F(\mu_n) - F(\mu)\}$$
$$\leq 0,$$
$$F(\hat{\mu}_n) - F(\mu) \leq \{F_n(\mu) - F(\mu)\} - \{F_n(\hat{\mu}_n) - F(\hat{\mu}_n)\}$$
$$\leq |F_n(\mu) - F(\mu)| + |F_n(\hat{\mu}_n) - F(\hat{\mu}_n)|$$
$$\leq 2 \sup_{z \in \mathcal{M}} |F_n(z) - F(z)|.$$

600   On the other hand, by the strong convexity of $F(z)$,

$$F(\hat{\mu}_n) - F(\mu) \geq \alpha(K, D) d^2(\hat{\mu}_n, \mu).$$

601 Therefore, by combining them, if $d(\hat{\mu}_n, \mu) \geq \epsilon$, then

$$
\begin{aligned}
\alpha(K, D)\epsilon^2 &\leq F(\hat{\mu}_n) - F(\mu) \\
&\leq 2 \sup_{z \in \mathcal{M}} |F_n(z) - F(z)|.
\end{aligned}
$$

602 Hence,

$$
\{d(\hat{\mu}_n, \mu) \geq \epsilon\} \subseteq \left\{ \sup_{z \in \mathcal{M}} |F_n(z) - F(z)| \geq \frac{\alpha(K, D)}{2}\epsilon^2 \right\},
$$

603 and

$$
\mathbb{P}\left[d(\hat{\mu}_n, \mu) \geq \epsilon\right] \leq \mathbb{P}\left[ \sup_{z \in \mathcal{M}} |F_n(z) - F(z)| \geq \frac{\alpha(K, D)}{2}\epsilon^2 \right].
$$

604 So, it suffices to control $\sup_{z \in \mathcal{M}} |F_n(z) - F(z)|$ by an exponential tail.

605 Recall that

$$
F_n(z) - F(z) = \frac{1}{n} \sum_{i=1}^{n} \left\{ d^2(Y_i, z) - \mathbb{E}[d^2(Y, z)] \right\}.
$$

606 Define

$$
X_i(z) = d^2(Y_i, z) - \mathbb{E}[d^2(Y, z)].
$$

607 Then, $\mathbb{E}[X_i(z)] = 0$ and

$$
F_n(z) - F(z) = \frac{1}{n} \sum_{i=1}^{n} X_i(z).
$$

608 Because $\mathcal{M}$ has diameter $\mathrm{diam}(\mathcal{M}) \leq D$, $d^2(\cdot, \cdot) \leq D^2$. Hence, for any $z$,

$$
X_i(z) \in [-D^2, D^2].
$$

609 By Hoeffding's inequality, for a fixed $z$,

$$
\begin{aligned}
\mathbb{P}\left[|F_n(z) - F(z)| \geq t\right] = \mathbb{P}\left[\left|\sum_{i=1}^{n} X_i(z)\right| \geq nt\right] \\
\leq 2\exp\left(-\frac{nt^2}{2D^4}\right).
\end{aligned}
$$

610 Here, for every fixed $\epsilon$, one obtains a bound of the form

$$
\mathbb{P}\left[ \sup_{z \in \mathcal{M}} |F_n(z) - F(z)| \geq t \right] \leq c_1' \exp\left(-c_2' n t^2\right),
$$

611 for constants $c_1', c_2' > 0$ depending on $K, D$ and on the metric complexity of $\mathcal{M}$,

$$
\begin{aligned}
c_1' &= 2\left(\frac{\alpha(K, D)D}{\delta}\right)^m, \\
c_2' &= \frac{\alpha(K, D)}{8D^2},
\end{aligned}
$$

612 that are from standard references in manifold-valued statistics.

613 Putting it all together,

$$
\begin{aligned}
\mathbb{P}\left[d(\hat{\mu}_n, \mu) \geq \epsilon\right] &\leq \mathbb{P}\left[ \sup_{z \in \mathcal{M}} |F_n(z) - F(z)| \geq \frac{\alpha(K, D)}{2}\epsilon^2 \right] \\
&\leq c_1' \exp\left\{-c_2 n \left(\frac{\alpha(K, D)}{2}\epsilon^2\right)^2\right\}.
\end{aligned}
$$

614 This concludes the required proof. $\qquad\square$

*Proof for Proposition 2.* By Theorem 1, there exist positive constants $c_1 = c_1(K, D)$ and $c_2 = c_2(K, D)$, such that for every $\epsilon > 0$,

$$\mathbb{P}\left[d(\hat{\mu}_n, \mu) > \epsilon\right] \leq c_1 \exp\left(-c_2 n \epsilon^2\right).$$

For any nonnegative random variable $Z$ and any $p \geq 1$, one has the standard identity

$$\mathbb{E}[Z^p] = \int_0^\infty p\epsilon^{p-1} \mathbb{P}(Z > \epsilon) d\epsilon.$$

This follows from writing $\mathbb{E}[Z^p] = \int_0^\infty p\epsilon^{p-1} \mathbb{1}(Z > \epsilon) d\epsilon$ and exchanging expectation and integral. Applying this to $Z = d(\hat{\mu}_n, \mu)$,

$$\mathbb{E}[d^p(\hat{\mu}_n, \mu)] = \int_0^\infty p\epsilon^{p-1} \mathbb{P}[d(\hat{\mu}_n, \mu) > \epsilon] d\epsilon.$$

Therefore,

$$\mathbb{E}[d^p(\hat{\mu}_n, \mu)] \leq \int_0^\infty p\epsilon^{p-1} \left[c_1 \exp(-c_2 n\epsilon^2)\right] d\epsilon$$

$$= c_1 \int_0^\infty p\epsilon^{p-1} \exp(-c_2 n\epsilon^2) d\epsilon.$$

Let $u = \sqrt{n}\epsilon$. Then, $\epsilon = u/\sqrt{n}$ and $d\epsilon = \frac{1}{\sqrt{n}} du$. Also,

$$\epsilon^{p-1} = \left(\frac{u}{\sqrt{n}}\right)^{p-1} = n^{-(p-1)/2} u^{p-1},$$

$$\exp(-c_2 n\epsilon^2) = \exp(-c_2 u^2).$$

So,

$$\int_0^\infty \epsilon^{p-1} \exp(-c_2 n\epsilon^2) d\epsilon = \int_0^\infty n^{-(p-1)/2} u^{p-1} \exp(-c_2 u^2) \frac{1}{\sqrt{n}} du$$

$$= n^{-\frac{p-1}{2}} n^{-\frac{1}{2}} \int_0^\infty u^{p-1} \exp(-c_2 u^2) du$$

$$= n^{-\frac{p}{2}} \int_0^\infty u^{p-1} \exp(-c_2 u^2) du.$$

Now, evaluate $\int_0^\infty u^{p-1} \exp(-c_2 u^2) du$. This is a known integral that can be expressed via the Gamma function. Indeed,

$$\int_0^\infty u^{p-1} \exp(-c_2 u^2) du = \frac{1}{2} c_2^{-\frac{p}{2}} \Gamma\left(\frac{p}{2}\right),$$

and

$$\int_0^\infty \epsilon^{p-1} \exp(-c_2 n\epsilon^2) d\epsilon = n^{-\frac{p}{2}} \left[\frac{1}{2} c_2^{-\frac{p}{2}} \Gamma\left(\frac{p}{2}\right)\right].$$

Therefore,

$$\mathbb{E}\left[d^p(\hat{\mu}_n, \mu)\right] \leq c_1 p \left\{n^{-\frac{p}{2}} \left[\frac{1}{2} c_2^{-\frac{p}{2}} \Gamma\left(\frac{p}{2}\right)\right]\right\}.$$

Collecting constants and it gives the proof. $\qquad\square$

*Proof for Theorem 2.* Fix a point $x \in \mathbb{R}^d$. Define the weighted empirical measure of $Y$ given $x$ as

$$\nu_{n,x} := \sum_{i=1}^n w_{n,i}(x) \delta_{Y_i},$$

where $\delta_{Y_i}$ denotes the Dirac measure at $Y_i$. Because $\sum_{i=1}^n w_{n,i}(x) = 1$, this is indeed a probability measure on $\mathcal{M}$. Similarly, let $\nu_x$ be the true conditional distribution of $Y$ given $X = x$ as

$$\nu_x := \mathbb{P}\left[Y \in A \mid X = x\right],$$

for Borel sets $A \subseteq \mathcal{M}$. Then, observe that the estimator $\hat{\mu}_n^*(x)$ can be written as

$$\hat{\mu}_n^*(x) = \arg\min_{z \in \mathcal{M}} \sum_{i=1}^{n} w_{n,i}(x) d^2(Y_i, z)$$

$$= \arg\min_{z \in \mathcal{M}} \int_{-\infty}^{+\infty} d^2(y, z) d\nu_{n,x}(y).$$

That is, $\hat{\mu}_n^*(x)$ is precisely the Fréchet mean of the measure $\nu_{n,x}$. Meanwhile, $\mu^*(x)$ is the Fréchet mean of $\nu_x$:

$$\mu^*(x) = \arg\min_{z \in \mathcal{M}} \int_{-\infty}^{+\infty} d^2(y, z) d\nu_x(y).$$

Hence, the problem reduces to showing that as $n \to +\infty$, $\nu_{n,x}$ converges to $\nu_x$ in a sense strong enough to force their Fréchet means to converge.

From Assumption 1, one can expect that for any bounded function $f \colon \mathcal{M} \to \mathbb{R}$,

$$\int f d\nu_{n,x} = \sum_{i=1}^{n} w_{n,i}(x) f(Y_i) \overset{a.s.}{\underset{n \to \infty}{\to}} \mathbb{E}[f(Y) \mid X = x] = \int f d\nu_x.$$

Thus, $\nu_{n,x}$ converges to $\nu_x$ in the weak topology on probability measures.

For each measure $\nu$, define its Fréchet functional $F_\nu \colon \mathcal{M} \to \mathbb{R}$ by

$$F_\nu(z) := \int d^2(y, z) d\nu(y).$$

Here,

$$\hat{\mu}_n^*(x) = \arg\min_{z \in \mathcal{M}} F_{\nu_{n,x}}(z),$$

$$\mu^*(x) = \arg\min_{z \in \mathcal{M}} F_{\nu_x}(z).$$

One want $F_{\nu_{n,x}} \to F_{\nu_x}$ in a suitable sense that implies $\arg\min$ convergence. In fact, for pointwise consistency, it suffices to show that for each $z \in \mathcal{M}$,

$$F_{\nu_{n,x}}(z) = \sum_{i=1}^{n} w_{n,i}(x) d^2(Y_i, z) \overset{a.s.}{\to} \int d^2(y, z) d\nu_x(y) = F_{\nu_x}(z).$$

By Assumption 1, this convergence holds for each $z \in \mathcal{M}$.

To pass from pointwise convergence of $F_{\nu_{n,x}}$ to convergence of the minimizers $\hat{\mu}_n^*(x) \to \mu_{(}^*x)$, one can rely on the strict geodesic convexity of $d^2(\cdot, \cdot)$ in a $\mathrm{CAT}(K)$ space with small diameter. Concretely, from earlier arguments, there is a constant $\alpha(K, D)$ such that

$$F_{\nu_x}(z) - F_{\nu_x}(\mu^*(x)) \geq \alpha(K, D) d^2(z, \mu^*(x)),$$

for all $z \in \mathcal{M}$. This follows from the strong geodesic convexity of $z \mapsto \int d^2(y, z) d\nu_x(y)$. Equivalently, if $z$ is $\epsilon$-far from $\mu^*(x)$, then $F_{\nu_x}(z)$ exceeds the global minimum $F_{\nu_x}(\mu^*(x))$ at least $\alpha(K, D)\epsilon^2$.

Now, let $\epsilon > 0$. Suppose, contrary to what one want, that

$$d(\hat{\mu}_n^*(x), \mu^*(x)) \geq \epsilon.$$

By $\mathrm{CAT}(K)$-convexity,

$$F_{\nu_x}(\hat{\mu}_n^*(x)) - F_{\nu_x}(\mu^*(x)) \geq \alpha(K, D)\epsilon^2.$$

On the other hand,

$$F_{\nu_x}(\hat{\mu}_n^*(x)) - F_{\nu_x}(\mu^*(x)) = \left\{ F_{\nu_{n,x}}(\hat{\mu}_n^*(x)) - F_{\nu_{n,x}}(\mu^*(x)) \right\} + (F_{\nu_x} - F_{\nu_{n,x}})(\hat{\mu}_n^*(x)) - (F_{\nu_x} - F_{\nu_{n,x}})(\mu^*(x)).$$

Since $\hat{\mu}_n^*(x)$ minimizes $F_{\nu_{n,x}}$,

$$F_{\nu,x}(\hat{\mu}_n^*(x)) \leq F_{\nu_{n,x}}(\mu^*(x)).$$

Thus,
$$F_{\nu_{n,x}}(\hat{\mu}_n^*(x)) - F_{\nu_x}(\mu^*(x)) \le (F_{\nu_x} - F_{\nu_{n,x}})(\hat{\mu}_n^*(x)) - (F_{\nu_x} - F_{\nu_{n,x}})(\mu^*(x)).$$

Hence,
$$\alpha(K, D)\epsilon^2 \le \left|(F_{\nu_x} - F_{\nu_{n,x}})(\hat{\mu}_n^*(x))\right| + \left|(F_{\nu_x} - F_{\nu_{n,x}})(\mu^*(x))\right|.$$

But as $n \to +\infty$,
$$F_{\nu_{n,x}}(z) \to F_{\nu_x}(z),$$

pointwise for each $z$, so the difference $|F_{\nu_x}(z) - F_{\nu_{n,x}}(z)| \to 0$. By dominated convergence theorem,
$$\sup_{z \in \{\hat{\mu}_n^*(x), \mu^*(x)\}} \left|F_{\nu_{n,x}}(z) - F_{\nu_x}(z)\right| \overset{a.s.}{\underset{n \to 0}{\to}} 0.$$

Hence, for large $n$, the right-hand side in the above inequality is smaller than $\frac{1}{2}\alpha(K, D)\epsilon^2$, which is incompatible. Thus, for large $n$,
$$d(\hat{\mu}_n^*(x), \mu^*(x)) < \epsilon,$$

and
$$\hat{\mu}_n^*(x) \overset{a.s.}{\to} \mu^*(x).$$

This completes the proof of pointwise consistency. $\qquad\square$

*Proof for Theorem 3.* For each $x$, define the empirical weighted measure as follows.
$$\nu_{n,x} := \sum_{i=1}^n w_{n,i}(x)\delta_{Y_i},$$

where $\delta_y$ is the Dirac measure at $y$. Then,
$$\hat{\mu}_n^*(x) = \arg\min_{z \in \mathcal{M}} \int d^2(y, z)d\nu_{n,x}(y).$$

Simultaneously, define the local population measure near $x$:
$$\pi_{n,x} := \frac{\mathbb{E}\left[W\left(\frac{\|x-X\|}{h_n}\right)\mathbb{1}(Y \in \cdot)\right]}{\mathbb{E}\left[W\left(\frac{\|x-X\|}{h_n}\right)\right]},$$

which is the ideal measure that the kernel weighting is trying to approximate. Then define the local population Fréchet mean as
$$\tilde{\mu}_n^*(x) = \arg\min_{z \in \mathcal{M}} \int d^2(y, z)d\pi_{n,x}(y).$$

Here, $\tilde{\mu}_n^*(x)$ is the minimizer of the population version of the local kernel functional, and $\hat{\mu}_n^*(x)$ is the minimizer of the empirical version. Then one can write
$$d(\hat{\mu}_n^*(x), \mu^*(x)) \le d(\hat{\mu}_n^*(x), \tilde{\mu}_n^*(x)) + d(\tilde{\mu}_n^*(x), \mu^*(x)).$$

Squaring and taking expectation, and applying $2ab \le a^2 + b^2$, one can get a bias–variance decomposition:
$$\mathbb{E}[d^2(\hat{\mu}_n^*(x), \mu^*(x))] \le 2\mathbb{E}[d^2(\hat{\mu}_n^*(x), \tilde{\mu}_n^*(x))] + 2d^2(\tilde{\mu}_n^*(x), \mu^*(x)).$$

The first term in the right-hand side is the variance term, capturing how the empirical local measure $\nu_{n,x}$ fluctuates around $\pi_{n,x}$. The second term in the right-hand side is the bias term, capturing how the local population mean $\tilde{\mu}_n^*(x)$ differs from $\mu^*(x)$.

Recall that in a $\mathrm{CAT}(K)$ space, of diameter $\mathrm{diam}(\mathcal{M}) \le D$, there is a strong geodesic convexity constant $\alpha(K, D)$ such that
$$\int d^2(y, z)d\nu(y) - \int d^2(y, z^*)d\nu(z^*) \ge \alpha(K, D)d^2(z, z^*),$$

for all probability measures $\nu$ on $\mathcal{M}$, provided the measure is fully supported in a ball of diameter $\text{diam}(\mathcal{M}) \leq D$. Hence, for the local measure $\pi_{n,x}$,

$$\int d^2(y, \hat{\mu}_n^*(x)) d\pi_{n,x} - \int d^2(y, \tilde{\mu}_n^*(x)) d\pi_{n,x}(y) \geq \alpha(K, D) d^2(\hat{\mu}_n^*(x), \tilde{\mu}_n^*(x)).$$

Because $\hat{\mu}_n^*(x)$ minimizes $\int d^2(y, z) d\nu_{n,x}(y)$,

$$\int d^2(y, \hat{\mu}_n^*(x)) d\nu_{n,x}(y) \leq \int d^2(y, \tilde{\mu}_n^*(x)) d\nu_{n,x}(y).$$

By subtracting the corresponding population measure integrals,

$$[\nu_{n,x} - \pi_{n,x}] d^2(\cdot, \hat{\mu}_n^*(x)) - [\nu_{n,x} - \pi_{n,x}] d^2(\cdot, \tilde{\mu}_n^*(x)) \leq \int d^2(y, \tilde{\mu}_n^*(x)) d\pi_{n,x}(y) - \int d^2(y, \hat{\mu}_n^*(x)) d\pi_{n,x}(y)$$

$$\int d^2(y, \hat{\mu}_n^*(x)) d\pi_{n,x}(y) - \int d^2(y, \tilde{\mu}_n^*(x)) d\pi_{n,x}(y) \leq \Delta_n(x),$$

where

$$\Delta_n(x) := \left| [\nu_{n,x} - \pi_{n,x}] d^2(\cdot, \hat{\mu}_n^*(x)) \right| + \left| [\nu_{n,x} - \pi_{n,x}] d^2(\cdot, \tilde{\mu}_n^*(x)) \right|.$$

Combining with the strong convexity inequality,

$$\alpha(K, D) d^2(\hat{\mu}_n^*(x), \tilde{\mu}_n^*(x)) \leq \Delta_n(x)$$

$$d^2(\hat{\mu}_n^*(x), \tilde{\mu}_n^*(x)) \leq \frac{\Delta_n(x)}{\alpha(K, D)}.$$

Taking expectation with respect to the sample $\{(X_i, Y_i)\}_{i=1}^n$,

$$\mathbb{E}[d^2(\hat{\mu}_n^*(x), \tilde{\mu}_n^*(x))] \leq \frac{\mathbb{E}[\Delta_n(x)]}{\alpha(K, D)}.$$

Recall that

$$\Delta_n(x) = \left| [\nu_{n,x} - \pi_{n,x}] d^2(\cdot, \hat{\mu}_n^*(x)) \right| + \left| [\nu_{n,x} - \pi_{n,x}] d^2(\cdot, \tilde{\mu}_n^*(x)) \right|$$

$$= \left| \sum_{i=1}^n w_{n,i}(x) \left\{ d^2(Y_i, \hat{\mu}_n^*(x)) - \mathbb{E}[d^2(Y, \tilde{\mu}_n^*(x) \mid X \approx x]\right\} \right|$$

$$+ \left| \sum_{i=1}^n w_{n,i}(x) \left\{ d^2(Y_i, \hat{\mu}_n^*(x)) - \mathbb{E}[d^2(Y, \tilde{\mu}_n^*(x) \mid X \approx x]\right\} \right|.$$

Since $\hat{\mu}_n^*$ itself depends on the sample, a straightforward application of Hoeffding's inequality is tricky. However, one can use Efron–Stein or Bennett–type inequalities for U-statistics, or the bounded differences approach, carefully analyzing how a single $Y_i$ affects $\hat{\mu}_n^*$. Such arguments appear in standard references on manifold-valued kernel regression. Thus, one can obtain

$$\mathbb{E}[\Delta_n(x)] = O\left((nh_n^d)^{-1/2}\right).$$

Hence,

$$\mathbb{E}[d^2(\hat{\mu}_n^*(x), \tilde{\mu}_n^*(x))] \leq \frac{C_{\text{var}}}{\alpha(K, D)} (nh_n^d)^{-1/2},$$

where $C_{\text{var}}$ is a constant depending on the kernel shape, the distribution of $(X, Y)$ near $x$ and the geometry constants $(K, D)$.

Next, recall that

$$\tilde{\mu}_n^*(x) = \arg\min_{z \in \mathcal{M}} \int d^2(y, z) d\pi_{n,x}(y),$$

$$\mu^*(x) = \arg\min_{z \in \mathcal{M}} \int d^2(y, z) d\nu_x(y),$$

where $\nu_x(\cdot) = \mathbb{P}[Y \in \cdot \mid X = x]$. As one move from $X = x$ to a local neighborhood $\{x' \mid \|x - x'\| \leq O(h_n)\}$, it can be expected that $\tilde{\mu}_n^*(x)$ to approximate $\mu^*(x')$ for some $x' \approx x$. Then $\mu^*(x')$ is close to $\mu^*(x)$ if $\mu^*$ is $\beta$-Hölder.

Because $\pi_{n,x}$ is essentially the distribution of $Y \mid X \in \{x' \mid \|x' - x\| \leq ch_n\}$, let $x^\natural$ be some effective point near $x$. Then by using smoothness or local Lipschitz condition on the conditional distributions,

$$d(\tilde{\mu}_n^*(x), \mu^*(x')) \leq C_{\mathrm{bias}}(h_n^\beta),$$

for some constant $C_{\mathrm{bias}} > 0$. Then one adds

$$d(\mu^*(x'), \mu^*(x)) \leq L \cdot \|x' - x\| \approx Lh_n^\beta.$$

Hence,

$$d(\tilde{\mu}_n^*(x), \mu^*(x)) \leq d(\tilde{\mu}_n^*(x), \mu^*(x')) + d(\mu^*(x'), \mu^*(x)) = O(h_n^\beta),$$

and

$$d^2(\tilde{\mu}_n^*(x), \mu^*(x)) = O(h_n^{2\beta}).$$

Putting it all together in the bias–variance decomposition, it completes the required proof. $\qquad\square$

 **B.3 Proofs for Section 3.3**

702 *Proof for Lemma 6.* Let $y'$ be a point on the geodesic segment $[xy$ such that $y'$ is very close to $x$.
703 Similarly, pick $z'$ on $[xz]$. So,

$$d(x, y') = \delta,$$
$$d(x, z') = \delta,$$

704 for some $\delta > 0$. Thi triangle $\triangle xy'z'$ has perimeter $\leq d(x, y) + d(y, z) + d(z, x)$, which is assumed
705 $\leq \pi/\sqrt{K}$ if $K > 0$. For $\delta$ small enough, the side lengths of $\triangle xy'z'$ are also $\leq \pi/\sqrt{K}$. By the
706 CAT$(K)$ definition,

$$d(y', z') \leq d_{\mathbb{M}_k}(\bar{y}', \bar{z}'),$$

707 and

$$d(x, y') = d(\bar{x}, \bar{y}') = \delta,$$
$$d(x, z') = d(\bar{x}, \bar{z}') = \delta.$$

708 The triangle $\triangle \bar{x}\bar{y}'\bar{z}'$ is in the same model plane as $\triangle \bar{x}\bar{y}\bar{z}$, but its typically much smaller near $\bar{x}$.

709 By definition of the Alexandrov angle,

$$\angle_x(y, z) = \lim_{\delta \to 0} \angle_x^{(\text{sec})}(y', z'),$$

710 where $\angle_x^{(\text{sec})}(y', z')$ is the secular angle of $\triangle xy'z'$ at $x$. Equivalently, it is the Euclidean angle
711 $\angle_{\bar{x}}(\bar{y}', \bar{z}')$ in the comparison triangle $\triangle \bar{x}\bar{y}'\bar{z}'$. Thus,

$$\angle_x(y, z) = \lim_{\delta \to 0} \angle_{\bar{x}}(\bar{y}', \bar{z}').$$

712 One also have the angle $\angle_{\bar{x}}(\bar{y}, \bar{z})$ in the large triangle $\triangle \bar{x}\bar{y}\bar{z}$, and want to show

$$\angle_{\bar{x}}(\bar{y}', \bar{z}') \leq \angle_{\bar{x}}(\bar{y}, \bar{z}),$$

713 for each small $\delta$, from which it will follow in the limit that $\angle_x(y, z) \leq \angle_{\bar{x}}(\bar{y}, \bar{z})$.

714 The CAT$(K)$ condition states that $\triangle xy'z'$ is no thicker than the model $\triangle \bar{x}\bar{y}'\bar{z}'$. More precisely, if
715 one place $\triangle xy'z'$ and $\triangle \bar{x}\bar{y}'\bar{z}'$ side by side so that $x \leftrightarrow \bar{x}$, $y' \leftrightarrow \bar{y}'$, $z' \leftrightarrow \bar{z}'$ correspond, one have

$$d(y', z') \leq d_{\mathbb{M}_K}(\bar{y}', \bar{z}').$$

716 Meanwhile, $\triangle \bar{x}\bar{y}'\bar{z}' \subset \triangle \bar{x}\bar{y}\bar{z}$ or can be inscribed in it, with the property that $asy' \to x$ and $z' \to x$,
717 the points $\bar{y}' \to \bar{x}$ and $\bar{z}' \to \bar{x}$.

718 Geometrically, on the model side, it is known (from classical geometry in constant curvature) that

$$\angle_{\bar{x}}(\bar{y}', \bar{z}') \leq \angle_{\bar{x}}(\bar{y}, \bar{z}). \tag{15}$$

719 This is because in a convex geometry (like a sphere of radius $1/\sqrt{K}$ or a Euclidean plane if $K = 0$),
720 drawing smaller radii $\bar{x}\bar{y}'$ and $\bar{x}\bar{z}'$ inside the bigger radii $\bar{x}\bar{y}$ and $\bar{x}\bar{z}$ yields smaller or equal angles
721 from the center $\bar{x}$.

722 More precisely, if one revolve the segment $\bar{y}'\bar{z}'$ about $\bar{x}$ within the triangle $\triangle \bar{x}\bar{y}\bar{z}$, the angle $\angle_{\bar{x}}(\bar{y}', \bar{z}')$
723 cannot exceed $\angle_{\bar{x}}(\bar{y}, \bar{z})$.

724 One thus have, for each small $\delta > 0$,

$$\angle_{\bar{x}}(\bar{y}', \bar{z}') \leq \angle_{\bar{x}}(\bar{y}, \bar{z}).$$

725 By the definition,

$$\angle_x(y, z) = \lim_{\delta \to 0} \angle_{\bar{x}}(\bar{y}', \bar{z}') \leq \angle_{\bar{x}}(\bar{y}, \bar{z}).$$

726 This completes the proof. Thus the angle at $x$ in the real triangle $\triangle xyz$ is bounded above by the
727 corresponding angle at $\bar{x}$ in the comparison triangle $\triangle \bar{x}\bar{y}\bar{z}$. $\square$

*Proof for Lemma 7.* Let $\triangle pqr \subset \mathcal{M}$ have side lengths

$$a = d(p, q), \quad b = d(q, r), \quad c = d(r, p),$$

and let $\angle_p(q, r)$ denote the Alexandrov angle at $p$. Similarly, let $\triangle p'q'r'$ have side lengths

$$a' = d(p', q'), \quad b' = d(q', r'), \quad c' = d(r', p'),$$

with angle $\angle_{p'}(q', r')$.

Assume that both triangles have perimeter $\leq \pi/\sqrt{K}$ if $K > 0$, ensuring they can be compared to triangles in the simply connected model space of curvature $K$ (sphere of radius $1/\sqrt{K}$ if $K > 0$, Euclidean plane if $K = 0$, or hyperbolic plane if $K < 0$). Then, the goal is to show that

$$|\angle_p(q, r) - \angle_{p'}(q', r')| \leq C\left[d(p, p') + d(q, q') + d(r, r')\right],$$

for some constant $C$ depending on $\alpha(K, D)$ or directly $\pi/\sqrt{K}$.

From the triangle inequality, one get for instance

$$\begin{aligned}
|a - a'| = |d(p, q) - d(p', q')| \\
\leq d(p, p') + d(q, q'),
\end{aligned}$$

and similarly,

$$\begin{aligned}
|b - b'| &\leq d(q, q') + d(r, r'), \\
|c - c'| &\leq d(r, r') + d(p, p').
\end{aligned}$$

Hence, each difference in corresponding side lengths is at most

$$\max\{|a - a'|, |b - b'|, |c - c'|\} \leq d(p, p') + d(q, q') + d(r, r') =: \delta_{pp'qq'rr'}.$$

Then,

$$|a - a'| \leq \delta_{pp'qq'rr'}, \quad |b - b'| \leq \delta_{pp'qq'rr'}, \quad |c - c'| \leq \delta_{pp'qq'rr'}.$$

In classical geometry of constant curvature $K$ (sphere, Euclidean plane, and hyperbolic plane), the side lengths $(a, b, c)$ uniquely determine the shape of a triangle (up to rigid motion) provided $a, b, c$ satisfy the triangle inequality. The angle $\eta := \angle_p(q, r)$ (or its model-space counterpart $\bar{\eta}$) is a continuous function of $(a, b, c)$.

- If $K = 0$ (Euclidean), one have the law of cosines

$$c^2 = a^2 + b^2 - 2ab\cos(\eta),$$

    so

$$\cos(\eta) = \frac{a^2 + b^2 + c^2}{2ab}.$$

    This is a rational, continuous function of $(a, b, c)$.

- If $K > 0$ (spherical), the spherical law of cosines yield

$$\cos(\sqrt{K}c) = \cos(\sqrt{K}a)\cos(\sqrt{K}b) + \sin(\sqrt{K}a)\sin(\sqrt{K}a)\sin(\sqrt{K}b)\cos(\eta).$$

- If $< 0$ (hyperbolic), one have similar hyperbolic law of cosines with $\cosh$ and $\sinh$.

$$\cosh(c/K) = \cosh(a/K)\cosh(b/K) - \sinh(a/K)\sinh(b/K)\cos(\eta).$$

In each case, as long as $a, b, c \leq \pi/\sqrt{|K|}$, one remain in a region where the side-length–angle relation is well-defined and continuously differentiable. Then, there exists a function

$$F: \{(a, b, c)\} \subset \mathbb{R}^3_{>0} \to [0, \pi],$$

so that if $\triangle xyz$ in the model space has sides $(a, b, c)$, then the angle at $x$ is $F(a, b, c)$. Moreover, $F$ is Lipschitz continuous on the domain $\{(a, b, c) \mid a + b + c \leq \pi/\sqrt{K}\}$. Hence, if $(a, b, c)$ and $(a', b', c')$ are close in $\mathbb{R}^3$, then

$$|F(a, b, c) - F(a', b', c')| \leq K_0 \left(|a - a'| + |b - b'| + |c - c'|\right),$$

for some constant $K_0$ depending only on $\max(a, b, c) \leq \pi/\sqrt{K}$.

Now connect the actual angles $\angle_p(q, r)$, $\angle_{p'}(q', r')$ in CAT$(K)$ to their comparison angles $\bar{\alpha}$, $\bar{\alpha}'$ in the model space. For $\triangle pqr \subset M$, choose the comparison triangle $\triangle \bar{p}\bar{q}\bar{r} \subset \bar{M}$ in the model space of curvature $K$, with side lengths $\bar{p}\bar{q} = a$, $\bar{q}\bar{r} = b$, $\bar{r}\bar{p} = c$. Let $\bar{\eta} = \angle_{\bar{p}}(\bar{q}, \bar{r})$. For $\triangle p'q'r' \subset M$, choose $\triangle \bar{p}'\bar{q}'\bar{r}' \subset \bar{M}$ similarly with side lengths $a', b', c'$. Let $\bar{\eta}' = \angle_{\bar{p}'}(\bar{q}', \bar{r}')$.

By Lemma 6 in CAT$(K)$:

$$\angle_p(q, r) \leq \bar{\eta},$$
$$\angle_{p'}(q', r') \leq \bar{\eta}'.$$

Symmetrically reversing the roles, one also get

$$\bar{\eta} \leq \angle_p(q, r).$$

Here, $\angle_p(q, r) \approx \bar{\eta}$ and $\angle_{p'}(q', r') \approx \bar{\eta}'$. Hence

$$|\angle_p(q, r) - \angle_{p'}(q', r')| \leq |\bar{\alpha} - \bar{\eta}'| + |\angle_p(q, r) - \bar{\eta}| + |\angle_{p'}(q', r') - \bar{\eta}'|.$$

But each difference $|\angle_p(q, r) - \bar{\eta}|$ is known to be small by the usual CAT$(K)$ thin triangle property. Specifically, if the perimeter is $\leq \pi/\sqrt{K}$, the difference $\angle_p(q, r) - \bar{\eta}$ can be bounded by a constant times the diameter of $\triangle pqr$; but that diameter is $\leq \max(a, b, c)$, already controlled.

In fact, in standard statements, one typically get an inequality of the form

$$|\angle_p(q, r) - \bar{\eta}| \leq \varepsilon_1(a, b, c) \quad \text{with } \varepsilon_1 \to 0 \text{ as } a, b, c \to 0,$$

and similarly for $\angle_{p'}(q', r')$. Since one are only after a linear bound in the final statement, it suffices that each difference is bounded by a universal constant (depending on $\pi/\sqrt{K}$). Thus, effectively

$$|\angle_p(q, r) - \angle_{p'}(q', r')| \leq 2 \, (\text{const}) + |\bar{\eta} - \bar{\eta}'|.$$

Hence collecting all,

$$\left|\angle_p(q, r) \ - \ \angle_{p'}(q', r')\right| \leq C_1 + C_2 \Delta$$

for constants $C_1$ and $C_2$. In typical statements of the lemma, one either arranges that $\Delta$ is small so that the additive constant $C_1$ is overshadowed, or uses a slightly refined thinness difference argument to show $\angle_p(q, r)$ and $\bar{\eta}$ differ by $\leq \tilde{C} \cdot \Delta$. In either case, one get a final bound of the form

$$\left|\angle_p(q, r) \ - \ \angle_{p'}(q', r')\right| \leq C \Delta = C(d(p, p') + d(q, q') + d(r, r')).$$

This completes the proof. $\qquad \square$

*Proof for Proposition 3.* First, from the geodesic convexity, if $\nu_x$ and $\nu_{x'}$ are close in distribution, then

$$d\big(\mu^*(x), \mu^*(x')\big) = C'' \epsilon,$$

for some constant $C''$ depending on $\alpha(K, D)$ and distributional assumptions (e.g. sub-Gaussianity or bounded diameter ensuring all integrals are finite).

Compare angles $\angle_{\mu^*(x)}(u, v)$ and $\angle_{\gamma^*(x')}(u, v)$. Let $[\mu^*(x), u]$ be the geodesic from $\mu^*(x)$ to $u$, $[\mu^*(x'), u]$ be the geodesic from $\mu^*(x')$ to $u$, and similarly for $[\mu^*(x), v]$ and $[\mu^(x'), v]$. Consider two triangles $\triangle\big(\mu^*(x), u, \mu^*(x')\big)$ and $\triangle\big(\mu^*(x), v, \mu^*(x')\big)$. Observe that $\mathrm{diam}(\mathcal{M}) \leq D$, so if $\mu^*(x)$ and $\mu^*(x')$ are also $\leq O(\epsilon)$ apart, then each of these triangles has perimeter $2D + O(\epsilon)$. If $K > 0$, $2D + O(\epsilon) < \pi/(\sqrt{K})$ by the initial assumption $D < \frac{\pi}{2\sqrt{K}}$ and $\epsilon$ small enough. Hence, each triangle is validly contained in a region where one can apply CAT$(K)$ angle comparisons (and the model-space comparison).

Let
$$p = \mu^*(x), \quad q = u, \quad r = \mu^*(x'),$$
and
$$p' = \mu^*(x'), \quad q' = u, \quad r' = \mu^*(x).$$
Then the pair $\triangle pqr$ and $\triangle p'q'r'$ have corresponding points:
$$p \leftrightarrow p', \quad q \leftrightarrow q', \quad r \leftrightarrow r'.$$
Notice that $q = q'$ is actually the same point $u$. The sum of vertex perturbations is
$$d(p,p') + d(q,q') + d(r,r') = d\big(\mu^*(x), \mu^*(x')\big) + 0 + d\big(\mu^*(x'), \mu^*(x)\big)$$
$$= 2d\big(\mu^*(x), \mu^*(x')\big),$$
and $d(\mu^*(x), \mu^*(x')) \leq C'' \epsilon$. By Lemma 7,
$$\big|\angle_p(q,r) - \angle_{p'}(q',r')\big| \leq C_1\big[d(p,p') + d(q,q') + d(r,r')\big].$$
Hence
$$\left|\angle_{\mu^*(x)}\big(u, \mu^*(x')\big) - \angle_{\mu^*(x')}\big(u, \mu^*(x)\big)\right| \leq C_1\big(2\,d(\mu^*(x), \mu^*(x'))\big)$$
$$\leq 2\,C_1\,C''\,\epsilon.$$
Similarly, for $\triangle \mu^*(x)\,v\,\mu^*(x')$, one get the same type of bound in terms of $\epsilon$.

Recall that $\angle_{\mu^*(x)}(u,v)$ is the Alexandrov angle between geodesics $[\mu^*(x)u]$ and $[\mu^*(x)v]$. In a CAT($K$) space, the angle $\angle_{\mu^*(x)}(u,v)$ can be added or compared if we know angles involving a third point $\mu^*(x')$. Thus,
$$\big|\angle_{\mu^*(x)}(u,v) - \big(\angle_{\mu^*(x)}(u, \mu^*(x')) + \angle_{\mu^*(x')}(u,v) - \pi\big)\big| \leq C_2 \cdot d(\mu^*(x), \mu^*(x')),$$
for some constant $C_2$.

Putting all these small angle increments together, conclude that
$$\big|\angle_{\mu^*(x)}(u,v) - \angle_{\mu^*(x')}(u,v)\big| \leq C\,d(\mu^*(x), \mu^*(x')) = O(\epsilon).$$
Hence the angles at $\mu^*(x)$ versus $\mu^*(x')$ differ by a linear factor in $\epsilon$. $\qquad\square$

*Proof for Theorem 4.* From Proposition 3, if $\nu_x \approx \nu_{x'}$ (i.e. their distance is $\leq \epsilon$), then for any pair $(u,v)$,
$$\left|\angle_{\mu^*(x)}(u,v) - \angle_{\mu^*(x')}(u,v)\right| \leq C_1\,\epsilon,$$
for some constant $C_1 > 0$. Hence for one pair of directions $(u,v)$, one get a linear-in-$\epsilon$ bound on how much the angle can change.

Now consider not just one pair, but all pairs $(u_i, u_j)$ with $1 \leq i < j \leq m$. But since each $\angle_{\mu^*(x)}(u_i, u_j)$ is covered by the same result,
$$\left|\angle_{\mu^*(x)}(u_i, u_j) - \angle_{\mu^*(x')}(u_i, u_j)\right| \leq C_1\,\epsilon,$$
for each pair $(u_i, u_j)$. Then the supremum over $i < j$ is also $\leq C_1\,\epsilon$. In fact, it is not even needed a union bound in probability sense, and each pair is bounded by the same linear factor $C_1\,\epsilon$. Hence
$$\sup_{1 \leq i < j \leq m}\left|\angle_{\mu^*(x)}(u_i, u_j) - \angle_{\mu^*(x')}(u_i, u_j)\right| \leq C_1\,\epsilon.$$
Thus one immediately extend from one pair to all $\binom{m}{2}$ pairs $(u_i, u_j)$.

In the hypothesis, it is typically stated that whenever $\|x - x'\|$ is small, then $\nu_x$ and $\nu_{x'}$ differ by $\epsilon(\|x - x'\|)$. For instance, in a classical kernel or smoothing scenario, if $\|x - x'\| \leq \delta$, then
$$d_W\big(\nu_x, \nu_{x'}\big) \leq \epsilon(\delta).$$
Hence setting $\epsilon = \epsilon(\delta)$, for $\|x - x'\| \leq \delta$,
$$\sup_{1 \leq i < j \leq m}\left|\angle_{\mu^*(x)}(u_i, u_j) - \angle_{\mu^*(x')}(u_i, u_j)\right| \leq C_1\,\epsilon(\delta).$$
Thus the angle difference is a function of $\delta$. Hence define $C := C_1$ (it might also absorb small distributional constants if needed), and putting it all together yields the proof. $\qquad\square$

## B.4 Proofs for Section 3.4

*Proof for Lemma 8.* In a smooth Riemannian manifold, for sufficiently close $u$ and $v$, the unique geodesics $\gamma_u \colon [0, \|U\|] \to \mathcal{M}$ and $\gamma_v \colon [0, \|V\|] \to \mathcal{M}$ from $z$ to $u$, respectively from $z$ to $v$, have well-defined initial velocity vectors at $z$. Let $\dot{\gamma}_u(0) \in T_z\mathcal{M}$ be the tangent vector to $\gamma_u$ at $z$. By construction, this is precisely $U$ if we identify $U \in T_z\mathcal{M}$ with the velocity vector in normal coordinates. Similarly, $\dot{\gamma}_v(0) = V \in T_z\mathcal{M}$.

In Riemannian geometry (without singularities around $z$), one then have:

$$\angle_z(u,v) \;=\; \angle\Big(\dot{\gamma}_u(0), \dot{\gamma}_v(0)\Big) \;=\; \cos^{-1}\Big(\frac{g_z\big(\dot{\gamma}_u(0), \dot{\gamma}_v(0)\big)}{\|\dot{\gamma}_u(0)\|\,\|\dot{\gamma}_v(0)\|}\Big).$$

Here $g_z(\cdot,\cdot)$ is the Riemannian metric at $z$. In simpler notation, if one identify $\dot{\gamma}_u(0) = U$ and $\dot{\gamma}_v(0) = V$, then

$$\angle_z(u,v) \;=\; \cos^{-1}\Big(\frac{g_z(U,V)}{\sqrt{g_z(U,U)\,g_z(V,V)}}\Big).$$

Use a geodesic coordinate system $\Phi \colon T_z\mathcal{M} \supset B_\delta(0) \to \mathcal{M}$ around $z$, with $\Phi(0) = z$ and $\mathrm{d}\Phi|_0 = \mathrm{Id}$. Concretely, $\Phi(U) = \exp_z(U)$. In these coordinates, the metric $g_{ij}(X)$ at a point $X$ in a small ball around $0 \in T_z\mathcal{M}$ has the well-known expansions:

$$g_{ij}(X) \;=\; \delta_{ij} \;-\; \tfrac{1}{3} R_{ikj\ell}(0)\, X^k\, X^\ell \;+\; O(\|X\|^3),$$

where $R_{ikj\ell}$ is the Riemann curvature tensor at $z$. The $-\tfrac{1}{3}$ factor is a standard convention from normal coordinate expansions; the main point is that the first non-trivial corrections appear at second order in $\|X\|$.

Hence, for vectors $U, V \in T_z\mathcal{M}$ with small norms, the inner product in the manifold at $z$ is

$$g_z(U,V) \;=\; \delta_{ij}\, U^i\, V^j \;-\; \tfrac{1}{3}\sum_{k,\ell}\Big(\tfrac{1}{2}R_{ikj\ell}(0)\Big) \;\ldots\; +\; O\big(\|U\|\|V\| \max(\|U\|,\|V\|)\big).$$

In simpler notation:

$$g_z(U,V) \;=\; \langle U,V\rangle_{\mathrm{Eucl}} \;+\; O\big(\|U\|\,\|V\|\,\max(\|U\|,\|V\|)\big).$$

From the above expansions,

$$\sqrt{g_z(U,U)} \;=\; \|U\|_{\mathrm{Eucl}}\big[1 + O(\|U\|^2)\big]^{1/2} \;=\; \|U\| + O(\|U\|^3).$$

Similarly for $\|V\|$. In addition,

$$g_z(U,V) \;=\; \langle U,V\rangle_{\mathrm{Eucl}} \;+\; O(\|U\|\,\|V\|\,\max(\|U\|,\|V\|)).$$

Thus

$$\frac{g_z(U,V)}{\sqrt{g_z(U,U)\,g_z(V,V)}} \;=\; \frac{\langle U,V\rangle}{\|U\|\,\|V\|} \;+\; O(\|U\|^2 + \|V\|^2),$$

since each correction is second-order in $\|U\|$ or $\|V\|$. Moreover,

$$\angle_z(u,v) \;=\; \cos^{-1}\Big(\frac{g_z(U,V)}{\sqrt{g_z(U,U)\,g_z(V,V)}}\Big) \;=\; \cos^{-1}\Big(\frac{\langle U,V\rangle}{\|U\|\,\|V\|} \;+\; O(\|U\|^2 + \|V\|^2)\Big).$$

When $\theta_0 = \angle_0(U,V)$ denotes the Euclidean angle in the tangent space,

$$\cos(\theta_0) \;=\; \frac{\langle U,V\rangle}{\|U\|\,\|V\|}.$$

Then

$$\cos(\angle_z(u,v)) \;=\; \cos(\theta_0) + O(\|U\|^2 + \|V\|^2).$$

833 Since $\cos$ is locally invertible around angles not equal to $0, \pi$ (and we assume $\theta_0$ is not degenerate or
834 extremely close to $\pi$ for typical use), a standard expansion yields:

$$\angle_z(u, v) \; = \; \theta_0 \; + \; O(\|U\|^2 + \|V\|^2).$$

835 Concretely, if $\theta_1 = \theta_0 + \delta$ satisfies $\cos(\theta_1) = \cos(\theta_0) + \eta$, then $\delta = O(\eta)$ for small $\eta$. Here,
836 $\eta = O(\|U\|^2 + \|V\|^2)$.

837 Hence,

$$\angle_z(u, v) \; = \; \theta_0 \; + \; O(\|U\|^2 + \|V\|^2),$$

838 where $\theta_0 = \angle_0(U, V)$ is the Euclidean angle of $U$ and $V$ in $T_z M$. This completes the proof. $\qquad\square$

839 *Proof for Proposition 4.* Let $\gamma(t)$ be a geodesic in $(\mathcal{M}, g)$ with $\gamma(0) = \mu^*$ and $\dot{\gamma}(0) = v$. Consider
840 $F(\gamma(t))$. Then

$$\frac{d}{dt} F(\gamma(t))\Big|_{t=0} = \frac{d}{dt} \int d^2(y, \gamma(t)) \, d\nu(y)\Big|_{t=0}$$
$$= \int \frac{d}{dt} d^2(y, \gamma(t))\Big|_{t=0} d\mu(y).$$

841 By standard Riemannian geometry formulas, if $\sigma(s)$ is the geodesic $[\, y \, \gamma(t)]$, then

$$\frac{d}{dt} d^2(y, \gamma(t)) \; = \; 2 \, d(y, \gamma(t)) \left\langle \dot{\gamma}(t), \, \dot{\sigma}(0) \right\rangle_{g_{\gamma(t)}}.$$

842 At $t = 0$, since $\gamma(0) = \mu^*$, one interpret $\dot{\sigma}(0)$ as the initial velocity from $\mu^*$ toward $y$. If $\mu^*$ is a
843 minimizer, the directional derivative must vanish for all directions $v$. Formally, this implies

$$\nabla F(\gamma^*) \; = \; 0.$$

844 Hence the first-order term in the expansion of $F(z)$ around $z = \mu^*$ vanishes.

845 Next, examine the second derivative (or Hessian) of $F$ at $\gamma^*$.

$$\mathrm{Hess}_z(F)(v, v) \; = \; \frac{d^2}{dt^2} F(\exp_z(t \, v))\Big|_{t=0}.$$

846 When $z = \mu^*$, and $\mu^*$ is the unique minimizer, these second derivatives measure how strongly $F$
847 curves upward around $\mu^*$.

848 In fact, the Gauss–Manasse–Busemann formula for second variation of distance shows that

$$\mathrm{H}_{\mu^*}(F)(v, v) \; = \; \int \mathrm{H}_{\mu^*}\big[d^2(y, \cdot)\big](v, v) \, d\mu(y).$$

849 Each term $\mathrm{H}_{\mu^*}\big[d^2(y, \cdot)\big](v, v)$ can be computed from the second variation of $\rho(\mu^*, y) = d(\mu^*, y)$.
850 In standard curvature conditions (especially nonpositive curvature or small diameter in positive
851 curvature), this Hessian is positive semidefinite, ensuring local convexity around $\mu^*$. If $\mathrm{CAT}(0)$
852 or if $\mathrm{diam} < \pi/(2\sqrt{K})$ in $\mathrm{CAT}(K)$, then $d^2(y, \cdot)$ is geodesically convex with a definite strong
853 convexity modulus $\alpha > 0$. Integrating preserves that positivity, giving $\mathrm{H}_{\mu^*}(F) \succeq 0$. Hence there is a
854 well-defined linear operator $H_{\mu^*}$ on $T_{\mu^*}\mathcal{M}$ representing $\mathrm{H}_{\mu^*}(F)$.

855 Because $F$ is at least $C^2$, one can write the remainder $R(v)$ in a standard Taylor expansion form:

$$R(v) \; = \; O\big(\|v\|^3\big) \quad \text{as } v \to 0.$$

856 Concretely, one can show this by analyzing the third derivative of $F$ in normal coordinates:

$$\frac{d^3}{dt^3} F\big(\exp_{\mu^*}(t \, v)\big)$$

857 remains bounded as $t \to 0$, so the third-order term is well-defined.

858 Hence the local expansion is

$$F\big(\exp_{\mu^*}(v)\big) \; = \; F(\mu^*) \; + \; \underbrace{\big\langle \nabla F(\mu^*), \, v \big\rangle}_{=0} \; + \; \tfrac{1}{2} \big\langle H_{\mu^*} \, v, \, v \big\rangle \; + \; R(v), \quad R(v) = O(\|v\|^3).$$

859 That is precisely the jet expansion for the Fréchet functional around $\mu^*$. $\qquad\square$

## B.5 Proofs for Section 3.5

*Proof for Proposition 5.* From the local Riemannian (or $\mathrm{CAT}(K)$) law of cosines in $\triangle \mu^* \, y \, z$:

$$d^2(y, z) = d^2(y, \mu^*) + d^2(z, \mu^*) - 2\, d(y, \mu^*)\, d(\mu^*, z)\, \cos\big(\angle_{\mu^*}(y, z)\big).$$

Rewriting as

$$d^2(y, z) - d^2(y, \mu^*) = d^2(z, \mu^*) - 2\, d(y, \mu^*)\, d(\mu^*, z)\, \cos\big(\angle_{\mu^*}(y, z)\big).$$

Here, let

$$\Delta_{\mathrm{dist}}(y, z, \mu^*) := d^2(\mu^*, z) - 2\, d(y, \mu^*)\, d(\mu^*, z),$$

$$\Delta_{\mathrm{angle}}(y, z, \mu^*) := 2\, d(y, \mu^*)\, d(z, \mu^*)\Big[1 - \cos\big(\angle_{\mu^*}(y, z)\big)\Big].$$

Observe that

$$-2\, d(y, \mu^*)\, d(\mu^*, z)\, \cos(\angle_{\mu^*}(y, z)) = \big[\Delta_{\mathrm{dist}} - d^2(\mu^*, z)\big] - \Delta_{\mathrm{angle}},$$

and

$$d^2(y, z) = d^2(y, \mu^*) + \Delta_{\mathrm{dist}}(y, z, \mu^*) + \Delta_{\mathrm{angle}}(y, z, \mu^*).$$

So the desired identity is obtained. $\qquad\square$

*Proof for Proposition 6.* Let

- $r_0 = d\big(\mu^*(x),\, u_0\big)$. (A constant for each $x$ if $u_0$ is fixed.)

- $r(y) = d\big(\mu^*(x),\, y\big) = R_x(y)$. (A variable depending on $y$.)

- $\alpha(y) = d(u_0, y)$. Another side of the triangle.

Then from the local law of cosines,

$$r(y)^2 = r_0^2 + \alpha(y)^2 - 2\, r_0\, \alpha(y)\, \cos\big(\angle_{\mu^*(x)}(u_0, y)\big).$$

But $\angle_{\mu^*(x)}(u_0, y) = \phi_x(y)$. So

$$r(y)^2 = r_0^2 + \alpha(y)^2 - 2\, r_0\, \alpha(y)\, \cos\big(\phi_x(y)\big).$$

We write it as

$$\Psi_x(y) = r(y)^2 = r_0^2 + \alpha(y)^2 - 2\, r_0\, \alpha(y)\, \cos\big(\phi_x(y)\big).$$

Now, to link $\alpha(y) = d(u_0, y)$ with $r(y)$ and $\phi_x(y)$, we may do yet another small expansion or an additional law-of-cosines approach. If the manifold is small enough in diameter, we can treat $\alpha(y)$ also as a function of $(r(y), \phi_x(y))$.

Also, let

$$\alpha(y)^2 = r_0^2 + r(y)^2 - 2\, r_0\, r(y)\, \cos\big(\angle_{u_0}(\mu^*(x),\, y)\big).$$

But $\angle_{u_0}(\mu^*(x), y)$ is not necessarily the same as $\phi_x(y)$. Then,

$$\alpha(y) = \alpha\big(r(y),\, \phi_x(y)\big) = r_0 + O\big(r(y)\big)$$

plus terms involving $\phi_x(y)$. In a small neighborhood, these expansions typically become second-order in $\phi_x(y)$. Hence, $\alpha(y)$ is not an independent variable; it's determined once $\phi_x(y)$ and $r(y) = R_x(y)$ are known.

In addition,

$$r(y)^2 = r_0^2 + \alpha(y)^2 - 2\, r_0\, \alpha(y)\, \cos\big(\phi_x(y)\big).$$

884 This yields a final expression of form

$$r(y)^2 = r_0^2 + \left(\text{some linear or quadratic function in } r(y)\right) + \left(\text{terms in } \phi_x(y)\right).$$

885 In short, the function $\Psi_x(y) = r(y)^2$ can be viewed as

$$\Psi_x(y) = \underbrace{f_{\text{radial}}\big(r(y)\big)}_{\text{part ignoring angles}} + \underbrace{f_{\text{angle}}\big(r(y),\, \phi_x(y)\big)}_{\text{angle corrections}},$$

886 where $f_{\text{angle}}$ is typically second-order or cross-term in $\phi_x(y)$.

887 Consider

$$\mathbb{E}_{\nu_x}\big[\Psi_x(Y)\big] = \int r(y)^2\, d\nu_x(y).$$

888 Let

889      • $\mathbb{E}_{\nu_x}\big[r(Y)\big]$ as some average radius.

890      • $\mathbb{E}_{\nu_x}[\phi_x(Y)]$ as average angle.

891 One obtains expansions, where

$$\Psi_x(Y) - r(y)^2\Big|_{\phi_x(Y)=0}$$

892 is some cross or higher-order term in $\phi_x(Y)$.

893 Then,

$$\mathbb{E}\big[\Psi_x(Y)^2\big] = \int \big[r(y)^2\big]^2\, d\nu_x(y).$$

894 Expanding $\big[r(y)^2\big]^2$ yields

$$\big[r(y)^2\big]^2 = r(y)^4 = \Big(f_{\text{radial}}(r(y)) + f_{\text{angle}}(r(y),\, \phi_x(y))\Big)^2.$$

895 One obtains terms:

896      • $\big[f_{\text{radial}}(r)\big]^2$,

897      • cross terms $2\, f_{\text{radial}}(r)\, f_{\text{angle}}(r, \phi)$,

898      • $\big[f_{\text{angle}}(r, \phi)\big]^2$.

899 By taking expectation,

$$\mathbb{E}\big[r(y)^4\big] = \mathbb{E}\Big(\big[f_{\text{radial}}(r)\big]^2\Big) + 2\,\mathbb{E}\Big(f_{\text{radial}}(r)\, f_{\text{angle}}(r, \phi)\Big) + \mathbb{E}\Big(\big[f_{\text{angle}}(r, \phi)\big]^2\Big).$$

900 Then, $\text{Var}[\Psi_x(Y)] = \mathbb{E}[\Psi_x(Y)^2] - (\mathbb{E}[\Psi_x(Y)])^2$ can be rearranged, grouping the radial part of the
901 variance from the angle cross terms:

$$\text{Var}\big[\Psi_x(Y)\big] = \text{Var}\Big(\underbrace{f_{\text{radial}}(r(Y))}_{\text{like } r(Y)^2 \text{ ignoring angles}}\Big) + \text{Cov}\big[\phi_x(Y),\, r(Y)^2\big] + \left(\text{smaller or higher-order expansions in } \phi_x(Y)\right).$$

902 Explicitly, let

$$A_x(Y) = f_{\text{radial}}\big(r(Y)\big) \quad (\text{often } = r(Y)^2)$$

903 ignoring angular corrections, and

$$B_x(Y) = f_{\text{angle}}\big(r(Y), \phi_x(Y)\big) \quad (\text{some function capturing dependence on angle } \phi_x(Y)).$$

904  Then

$$\Psi_x(Y) \;=\; A_x(Y) \;+\; B_x(Y).$$

905  Using

$$\mathrm{Var}[A + B] = \mathrm{Var}[A] + \mathrm{Var}[B] + 2\,\mathrm{Cov}(A, B),$$

906  one have

$$\mathrm{Var}[\Psi_x(Y)] \;=\; \mathrm{Var}[A_x(Y)] \;+\; \mathrm{Var}[B_x(Y)] \;+\; 2\,\mathrm{Cov}\big(A_x(Y),\, B_x(Y)\big).$$

907  If $B_x(Y)$ is small or mostly depends on $\phi_x(Y)$ with some bounding condition, one can inter-
908  pret $\mathrm{Var}[B_x(Y)]$ and $\mathrm{Cov}(A_x(Y),\, B_x(Y))$ as cross/higher-order expansions. Here, $\mathrm{Var}[A_x(Y)]$
909  is the purely radial piece $\mathrm{Var}[R_x(Y)^2]$.  The cross terms or expansions in $\phi_x(Y)$ become
910  $\mathrm{Cov}\big(\phi_x(Y),\, R_x(Y)^2\big)$. Hence we get the claimed partial decomposition.  $\square$

 # C  Additional Analysis on $\epsilon$-Approximate $\mathrm{CAT}(K)$ Space

912 In comparison geometry framework, the theoretical statements are provided on the model space with
913 constant curvature. In practice, however, real-world datasets may lie in spaces that only approximately
914 satisfy the curvature conditions. Below we introduce an $\epsilon$-approximate version of $\mathrm{CAT}(K)$ space,
915 and derive perturbed versions of existence, uniqueness, and convexity-type results.

916 **Definition 9** ($\epsilon$-Approximate $\mathrm{CAT}(K)$ Space). *Let $\epsilon > 0$. A geodesic metric space $(\mathcal{M}, d)$ is said to*
917 *be $\epsilon$-approximate $\mathrm{CAT}(K)$ space if for every geodesic triangle $\triangle pqr$ of perimater less than $2D_K$*
918 *(where $D_K = \pi/\sqrt{K}$ if $K > 0$, otherwise $D_K = \infty$), and for any points $x$ and $y$ on the edges $[pq]$*
919 *and $[qr]$, respectively, one has*

$$d(x, y) \le d_{\mathbb{M}_K^2}(\bar{x}, \bar{y}) + \epsilon, \tag{16}$$

920 *where $\triangle \bar{p}\bar{q}\bar{r} \subset \mathbb{M}_K^2$ is the usual comparison triangle in the simply connected model space of constant*
921 *curvature $K$.*

922 This definition allows a small additive slack $\epsilon$ in the usual comparison inequality. When $\epsilon = 0$, we
923 recover the standard definition of $\mathrm{CAT}(K)$.

924 **Theorem 5** (Approximate Geodesic Convexity of Squared Distance). *Let $(\mathcal{M}, d)$ be an $\epsilon$-approximate*
925 $\mathrm{CAT}(K)$ *space with $K < 0$. Fix any $p \in \mathcal{M}$, and define $f(x) = d^2(p, x)$. Then, for any geodesic*
926 $\gamma \colon [0, 1] \to \mathcal{M}$,

$$f(\gamma(t)) \le (1 - t)f(\gamma(0)) + tf(\gamma(1)) + O(\epsilon D), \tag{17}$$

927 *where $D$ is the diameter of the relevant geodesic segment under consideration, or the whole space if*
928 *bounded.*

929 *Proof.* Let $\gamma \colon [0, 1] \to \mathcal{M}$ be a geodesic from $\gamma(0) = x$ to $\gamma(1) = y$. Define $\gamma(t)$ as the point at
930 parameter $t$. We form a (possibly degenerate) triangle $\triangle pxy$ in $\mathcal{M}$. Then, $\triangle \bar{p}\bar{x}\bar{y}$ is the comparison
931 triangle in the model space $\mathbb{M}_K^2$ that has side lengths

$$d_{\mathbb{M}_K^2}(\bar{p}, \bar{x}) = d(p, x), \quad d_{\mathbb{M}_K^2}(\bar{x}, \bar{y}) = d(x, y), \quad {}_{\mathbb{M}_K^2}(\bar{y}, \bar{p}) = (y, p).$$

932 Let $\bar{\gamma}(t)$ be the point on $[\bar{x}, \bar{y}] \subset \triangle \bar{p}\bar{x}\bar{y}$ at fraction $t$. Because $\gamma$ is a geodesic and $[\bar{x}, \bar{y}]$ is also a
933 geodesic in $\mathbb{M}_K^2$, the pair $\gamma(t) \leftrightarrow \bar{\gamma}(t)$ correspond naturally for the sub-segment ratio $t$. Here, we
934 have

$$d(p, \gamma(t)) \le d_{\mathbb{M}_K^2}(\bar{p}, \bar{\gamma}(t)) + C_1 \epsilon,$$

935 for some constant $C_1$. By taking squares,

$$d^2(p, \gamma(t)) \le \left(d_{\mathbb{M}_K^2}(\bar{p}, \bar{\gamma}(t))\right)^2 + 2C_1 \epsilon d_{\mathbb{M}_K^2}(\bar{p}, \bar{\gamma}(t)) + (C_1 \epsilon)^2.$$

936 Since $K < 0$, the model space $\mathbb{M}_K^2$ is either Euclidean or hyperbolic. In both cases, it is known that

$$\{\bar{\gamma}(t) \mid t \in [0, 1]\} \subset [\bar{x}, \bar{y}],$$

937 which yields $\bar{\gamma}(t)$ satisfying the usual convexity of the squared distance in a non-positive curvature
938 setting.

$$\left(d_{\mathbb{M}_K^2}(\bar{p}, \bar{\gamma}(t))\right)^2 \le (1 - t)\left(d_{\mathbb{M}_K^2}(\bar{p}, \bar{x})\right)^2 + t\left(d_{\mathbb{M}_K^2}(\bar{p}, \bar{y})\right)^2.$$

939 Therefore,

$$d_{\mathbb{M}_K^2}(\bar{p}, \bar{\gamma}(t))^2 \le (1 - t)d^2(p, x) + td^2(p, y),$$

940 and

$$\begin{aligned} d^2(p, \gamma(t)) &\le (1 - t)d^2(p, x) + td^2(p, y) + 2C_1 \epsilon \left(d_{\mathbb{M}_K^2}(\bar{p}, \bar{\gamma}(t))\right) + (C_1 \epsilon)^2 \\ &\le (1 - t)d^2(p, x) + td^2(p, y) + 2C_1 \epsilon D' + (C_1 \epsilon)^2 \\ &\le (1 - t)d^2(p, x) + td^2(p, y) + C_2 \epsilon D, \end{aligned}$$

941 for some constant $C_2 > 0$, where $D'$ is the diameter of the model space, and can be bounded by local
942 diameter $D$. This can be written as

$$f(\gamma(t)) = d^2(p, \gamma(t)) \le (1 - t)f(\gamma(0)) + tf(\gamma(1)) + C_2 \epsilon D,$$

943 and it exactly states the approximate geodesic convexity for $f(x) = d^2(p, x)$. $\qquad \square$

**Corollary 1** (Approximate Uniqueness of Fréchet Mean). *Under the same $\epsilon$-approximate* $\mathrm{CAT}(K)$ *assumptions, consider the Fréchet functional*

$$F(x) = \int_{\mathcal{M}} d^2(y, x) d\nu(y), \tag{18}$$

*for a compactly supported probability measure $\nu$. Then, one has the following.*

- *A minimizer of $F$ exists for any $\epsilon > 0$.*

- *If $\epsilon$ is small, any two minimizers $m_1$ and $m_2$ must lie within a small neighborhood of each other:*

$$d(m_1, m_2) \leq O(\sqrt{\epsilon}). \tag{19}$$

  *Hence, strict uniqueness is replaced by an $\epsilon$-dependent bound.*

**Proposition 7** (Local Existence and Uniqueness). *Let $\mathcal{M}$ be a geodesic metric space that is $\mathrm{CAT}(K)$ (or $\epsilon$-approximately $\mathrm{CAT}(K)$ space) locally in a geodesic ball $B(p_0, R)$. That is, for any geodesic triangle fully contained in $B(p_0, R)$, the usual $\mathrm{CAT}(K)$ (or approximate) triangle comparison property holds. Suppose $\nu$ is a probability measure on $\mathcal{M}$ whose support $\mathrm{supp}(\nu)$ is contained in $B(p_0, R)$. Define the Fréchet functional*

$$F(x) = \int_{\mathcal{M}} d^2(y, x) d\nu(y).$$

*Then, one has the following.*

- *The function $F(x)$ attains its minimum at some $m \in B(p_0, R)$.*

- *If $K > 0$ but $\mathrm{diam}(\mathrm{supp}(\nu)) < \frac{\pi}{2\sqrt{K}}$, or if $K \leq 0$ (no diameter restriction), then $m$ is unique within $B(p_0, R)$.*

In other words, the Fréchet mean $m$ exists in the local ball $B(p_0, R)$ and is unique when the (local) curvature constraints enforce strict geodesic convexity.

**Proposition 8** (Heavy-Tailed Distributions and Slower Convergence). *Let $\mathcal{M}$ be either a strict $\mathrm{CAT}(K)$ space or an $\epsilon$-approximate $\mathrm{CAT}(K)$ space of diameter $\leq D$. Suppose $Y_1, Y_2, \ldots, Y_n$ are i.i.d. random points in $\mathcal{M}$ with common distribution $\nu$. Denote by*

$$\mu = \arg\min_{z \in \mathcal{M}} \mathbb{E}[d^2(Y, z)]$$

$$\hat{\mu} = \arg\min_{z \in \mathcal{M}} \frac{1}{n} \sum_{i=1}^{n} d^2(Y_i, z).$$

*Assume that*

1. *$\nu$ has finite second moments $\mathbb{E}[d^2(Y, z_0)] < \infty$ for some reference point $z_0$, and*

2. *the random variable $d^2(Y, z_0)$ satisfies a sub-exponential-type tail bound: there exist constants $\alpha \geq 0$, $\gamma \in (0, 1]$ such that*

$$\mathbb{P}\left(d^2(Y, z_0) > t\right) \leq \exp(-\alpha t^\gamma), \tag{20}$$

   *for all $t > 0$.*

*Then, there exist constants $c, C$ such that for all $n \geq 1$ and all $\epsilon > 0$,*

$$\mathbb{P}\left(d(\hat{\mu}_n, \mu) \geq \epsilon\right) \leq C \exp\left(-cn\epsilon^{2\gamma}\right). \tag{21}$$

*Hence $\hat{\mu}_n$ converges to $\mu$ in probability, and its deviation tails decay sub-exponentially with arte $\epsilon^{2\gamma}$.*

*Proof.* Define the population and empirical Fréchet functionals

$$F(z) = \mathbb{E}[d^2(Y, z)], \quad F_n(z) = \frac{1}{n} \sum_{i=1}^{n} d^2(Y_i, z).$$

By definition,

$$\mu = \arg\min_{z \in \mathcal{M}} F(z), \quad \hat{\mu}_n = \arg\min_{z \in \mathcal{M}} F_n(z).$$

Observe that

$$
\begin{aligned}
F(\hat{\mu}_n) - F(\mu) &= \{F(\hat{\mu}_n) - F_n(\hat{\mu}_n)\} + \{F_n(\hat{\mu}_n) - F_n(\mu)\} + \{F_n(\mu) - F(\mu)\} \\
&\le \{F(\hat{\mu}_n) - F_n(\hat{\mu}_n)\} - \{F(\mu) - F_n(\mu)\}, \\
|F(\hat{\mu}_n) - F(\mu)| &\le |F(\hat{\mu}_n) - F_n(\hat{\mu}_n)| + |F(\mu) - F_n(\mu)|.
\end{aligned}
$$

Therefore,

$$\{d(\hat{\mu}_n, \mu) \ge \epsilon\} \subseteq \left\{F(\hat{\mu}_n) - F(\mu) \ge \alpha(K,D)\epsilon^2\right\} \subseteq \left\{\sup_{z \in \mathcal{M}} \left| F_n(z) - F(z) \ge \frac{\alpha(K,D)}{2}\epsilon^2 \right| \right\}.$$

Here,

$$\sup_{z \in \mathcal{M}} |F_n(z) - F(z)| \le \max_{1 \le j \le N_\delta} |F_n(z_j) - F(z_j)| + \eta(\delta),$$

where $N_\delta \le \exp(C_1(D/\delta)^m)$ is a $\delta$-net for some $m$ and $\eta(\delta) \to 0$ as $\delta \to 0$. Taking $\delta \to 0$,

$$
\begin{aligned}
\mathbb{P}\left(\sup_{z \in \mathcal{M}} |F_n(z) - F(z)| \ge t\right) &\le N_\delta \cdot 2\exp\left(-c' n t^\gamma\right) + \mathbb{P}(\eta(\delta) \ge t/2) \\
&\approx \exp(\ln N_\delta - c' n t^\gamma).
\end{aligned}
$$

For fixed $D$, $\log N_\delta$ is polynomial in $(1/\delta)$ so we can absorb that into a constant factor. $\qquad\square$

# D   Relation to Geodesic Regression

A Riemannian manifold $(\mathcal{M}, g)$ is a smooth manifold endowed with a Riemannian metric $g$, which locally induces a norm on each tangent space $T_p\mathcal{M}$. In such a setting, the geodesic distance between two points $p, q \in \mathcal{M}$ is given by

$$d(p,q) = \inf_\gamma \int_0^1 \sqrt{g\big(\dot{\gamma}(t), \dot{\gamma}(t)\big)}\, dt,$$

where the infimum is taken over all smooth curves $\gamma$ joining $p$ and $q$. For points $q$ in a normal neighborhood of $p$, the exponential map $\exp_p \colon T_p\mathcal{M} \to \mathcal{M}$ is a diffeomorphism and we have the local relation

$$d^2(p,q) = \|\exp_p^{-1}(q)\|^2.$$

Moreover, assuming the sectional curvatures of $\mathcal{M}$ are bounded above by $K$, the manifold is also a $\mathrm{CAT}(K)$ space. In this smooth setting, one can use differential calculus; for example, the Fréchet functional $F(z) = \int_{\mathcal{M}} d^2(y, z)\, d\nu(y)$ is differentiable (at least locally), with gradient $\nabla F(z) = -2 \int_{\mathcal{M}} \exp_z^{-1}(y)\, d\nu(y)$, and a second-order expansion

$$F(\exp_z(v)) = F(z) + \langle \nabla F(z), v \rangle + \frac{1}{2}\langle H_z\, v, v \rangle + O(\|v\|^3).$$

Here, a $\mathrm{CAT}(K)$ space is a geodesic metric space $(\mathcal{M}, d)$ satisfying a comparison condition: for any geodesic triangle $\triangle pqr$ with perimeter less than a critical value (for $K > 0$) and any points $x$ and $y$ on two of its sides, the distance $d(x, y)$ is bounded above by the corresponding distance in the model space $\mathbb{M}_K^2$ of constant curvature $K$. In particular, if $\gamma \colon [0,1] \to \mathcal{M}$ is a geodesic, one has the following (strong) convexity inequality for the squared distance function:

$$d^2\big(y, \gamma(t)\big) \le (1-t)d^2\big(y, \gamma(0)\big) + t\, d^2(y, \gamma(1)) - \alpha\, t(1-t)\, d^2\big(\gamma(0), \gamma(1)\big),$$

where $\alpha = \alpha(K, D)$ is a constant depending on the curvature bound $K$ and the diameter $D$ of the region under consideration. The above inequality replaces the role of second-order (Hessian) information.

In geodesic regression on a Riemannian manifold, we assume that the regression function follows a geodesic curve. For example, for a predictor $x \in \mathbb{R}^d$, one common formulation is:

$$\mu(x) = \exp_p\Big((\alpha + \beta^\top x)\, v\Big), \quad \text{with } v \in T_p\mathcal{M},$$

or equivalently, writing the geodesic $\gamma$ from $p$ with initial velocity $v$,

$$\mu(x) = \gamma\big(\alpha + \beta^\top x\big).$$

Here, $p \in \mathcal{M}$ is a base point, $v \in T_p\mathcal{M}$ is a tangent vector at $p$, $\exp_p$ is the Riemannian exponential map, and $\alpha, \beta$ are the regression parameters. This model implies that the conditional mean of $Y$ given $X = x$ lies exactly on the geodesic determined by $p$ and $v$. Fréchet regression is defined more generally and does not restrict the mean to lie on a pre-specified geodesic. For each $x$, the conditional Fréchet mean is given by

$$\mu(x) = \underset{z \in \mathcal{M}}{\arg\min} \ \mathbb{E}\Big[d^2(Y, z) \mid X = x\Big].$$

If $\mathcal{M}$ is a Riemannian manifold and the conditional distribution of $Y$ given $X = x$ is concentrated and symmetric around a geodesic curve, then one may find that the minimizer satisfies

$$\mu(x) = \exp_p\Big((\alpha + \beta^\top x)\, v\Big),$$

thus recovering the geodesic regression solution. However, in general, Fréchet regression allows for much more flexible conditional mean structures. In summary, we can relate these two concepts (Fréchet regression and geodesic regression). Riemannian manifolds allow a local linearization via the exponential map and a full Taylor expansion, making geodesic regression a natural parametric model, and $\mathrm{CAT}(K)$ spaces provide a more general setting where one relies on strong convexity properties of the squared distance function rather than differentiability. Both approaches are unified under the Fréchet regression framework, with geodesic regression emerging as a parametric case when the conditional means lie on a geodesic.

# E    Details of Experiments

This section describes the details of experiments in Section 4.

**Model Details**    Throughout the experiment, we use an implementation of Fréchet regression based on the Nadaraya-Watson estimator [14, 21, 38].

$$\mu^*(x) = \underset{z \in \mathcal{M}}{\arg\min} \frac{1}{n} \sum_{i=1}^{n} K_h(X_i - x) d^2(Y_i, z),$$

where $K_h$ is a smoothing kernel that corresponds to a probability density with $K_h(\cdot) = h^{-1} K(\cdot/h)$. For the optimization, we use Limited-memory BFGS [30].

```python
import numpy as np
from scipy.optimize import minimize

# Kernel function (Gaussian kernel)
def gaussian_kernel(x, x_data, bandwidth):
    dists = np.linalg.norm(x_data - x, axis=1)
    weights = np.exp(-0.5 * (dists / bandwidth) ** 2)
    return weights / np.sum(weights)

# Fréchet objective function
def frechet_objective(y, responses, weights, distance_func):
    dists = np.array([distance_func(y, r) for r in responses])
    return np.sum(weights * dists**2)

# Fréchet regression function
def frechet_regression(X, Y, x_query, bandwidth, distance_func):
    weights = gaussian_kernel(x_query, X, bandwidth)
    y_init = np.mean(Y, axis=0)
    result = minimize(
        frechet_objective,
        y_init,
        args=(Y, weights, distance_func),
        method='L-BFGS-B'
    )
    return result.x
```

Listing 1: Python code for the Fréchet regression.

**Stereographic Projection**    Listing 2 shows the Python code for the stereographic projection from sphere surface to hyperbolic plane.

```python
# Define the stereographic projection function
def stereographic_projection(x, y, z, R):
    u = R * x / (R + z)
    v = R * y / (R + z)
return u, v
```

Listing 2: Python code for the stereographic projection.

 **E.1   Details for Illustrative Example 4.1**

 **Data Generating Process**   To assess the performance of the Fréchet regression estimator, consider
 to generate simulated data. The regression function is

$$\mu(x)(\cdot) = ((1 - x^2)1/2 \cos(\pi x), (1 - x^2)^{1/2} \sin(\pi x), x), \quad x \in (0, 1),$$

 which maps a spiral on the sphere. To generate a random sample $\{(X_i, Y_i)\}_{i=1}^n$, let $X_i \sim \mathcal{U}(0, 1)$
 followed by a bivariate normal random vector $U_i$, and

$$Y_i = \cos(\|U_i\|)\mu(X_i) + \sin(\|U_i\|)\frac{U_i}{\|U_i\|}.$$

 The sample size of the simulation data is $n = 50$, and Gaussian noise with variance $0.4$ is added to
 each instance.

 **E.2   Details for Experiments on Real-world Datasets 4.2**

 **Details of Datasets**

   • **HYG Stellar**: The HYG Stellar Database is a comprehensive star catalog that amalgamates
     data from several prominent astronomical catalogs, including HIPPARCOS, the Yale Bright
     Star Catalog, and the Gliese Catalog of Nearby Stars. This integration provides detailed
     information on stars' positions, brightness, spectral types, and various identifiers such as
     traditional names and Bayer designations. It contains detailed information on 119,614 stars
     including position data, photometric data and luminosity and variability.

   • **USGS Earthquake**: The USGS Earthquake catalogue provides information on earthquakes
     worldwide with a magnitude of 2.5 and above that have occurred over the past week, and it
     contains 300 instances.

   • **NOAA Climate**: The NOAA Climate data provides Two-Line Element (TLE) sets for
     weather satellites, including those operated by NOAA, and contains 72 instances. A TLE
     consists of two 69-character lines of data, each containing specific parameters that describe
     the satellite's orbit.

 Table 3 shows the detailed breakdown of variables $X$ and $Y$ for each dataset.

| Dataset | Sample size | Predictor $X$ | Response $Y$ |
|---|---|---|---|
| HYG Stellar | 119,614 | • Observation time $t$
• Brightness of the star $m$
• Absolute Magnitude $m'$
• Spectral type $s$ | Position on the celestial sphere |
| USGS Earthquake | 300 | • Observation time $t$
• Magnitude of the earthquake $m$
• Depth of the earthquake $d$ | Earthquake location |
| NOAA Climate | 72 | • Timestamp of the TLE $t$
• Orbital parameters $\theta$
• Inclination $i$ | Satellite position |

Table 3: Detailed breakdown of variables for each dataset.

 **Visualizations of Real-world Spherical Datasets**   Figure 5 shows the additional visualizations of
 real-world spherical datasets, and Figure 6 shows the heteroscedasticity in the NOAA and USGS
 datasets. In addition, Python code in Listing 3 shows the implementation for the visualization of
 HYG Steller dataset.

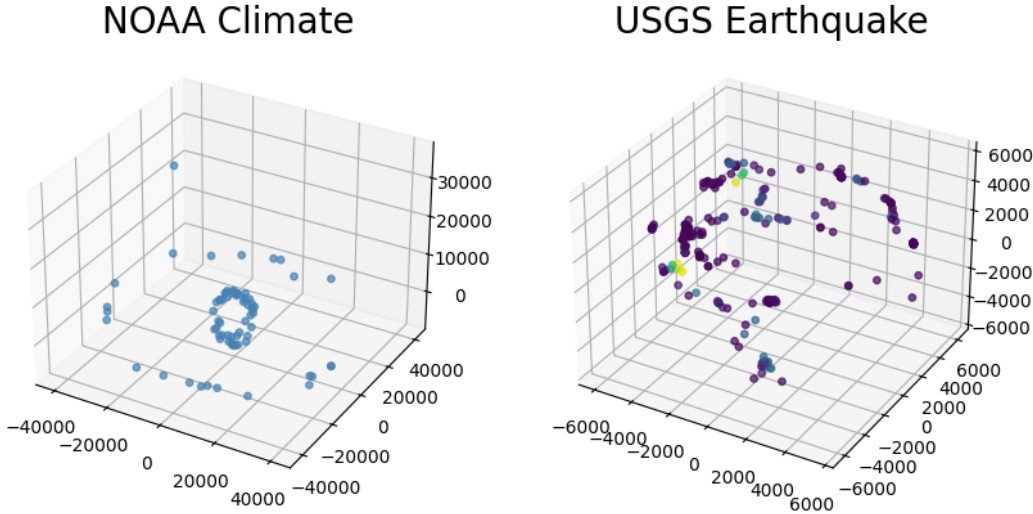

Figure 5: Visualizations for USGS Earthquake catalogue and NOAA Climate dataset.

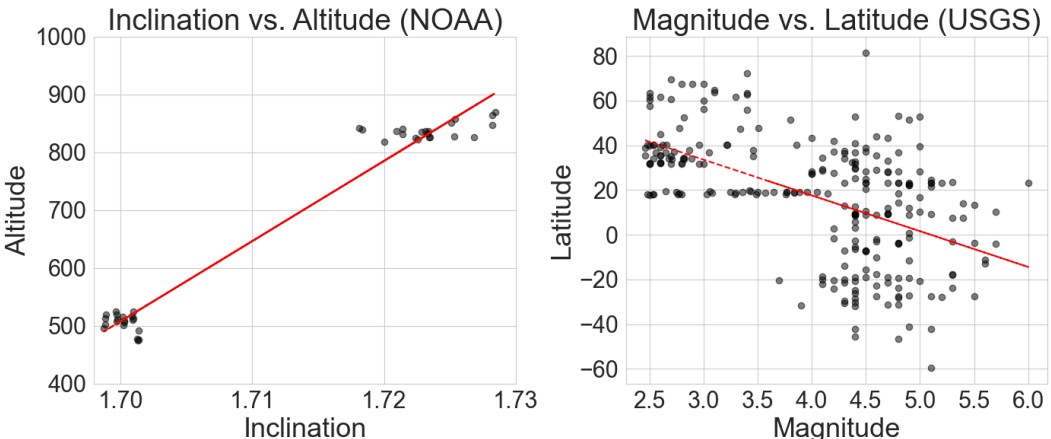

Figure 6: Heteroscedasticity in the NOAA and USGS datasets.

```python
import numpy as np
import matplotlib.pyplot as plt
from astropy.io import ascii

# Load the Bright Star Catalog
url = '{Data URL}' # URL for HYG Steller database
data = ascii.read(url)

# Extract Right Ascension and Declination
ra = np.array(data['ra'])   # in hours
dec = np.array(data['dec'])   # in degrees

# Convert RA from hours to degrees
ra_deg = ra * 15

# Convert RA and Dec to radians for plotting
ra_rad = np.radians(ra_deg)
dec_rad = np.radians(dec)

# Create a 3D scatter plot
fig = plt.figure(figsize=(12, 8))
ax = fig.add_subplot(111, projection='3d')

# Convert spherical coordinates to Cartesian for plotting
x = np.cos(dec_rad) * np.cos(ra_rad)
y = np.cos(dec_rad) * np.sin(ra_rad)
z = np.sin(dec_rad)

# Plot the stars
ax.scatter(x, y, z, color='white', s=0.01, label="data points")

ax.xaxis.set_ticklabels([])
ax.yaxis.set_ticklabels([])
ax.zaxis.set_ticklabels([])

# Set plot parameters
ax.set_facecolor('black')
ax.set_xlabel('X')
ax.set_ylabel('Y')
ax.set_zlabel('Z')
plt.legend(markerscale=80, fontsize=30)
plt.show()
```

Listing 3: Python code for the visualization of HYG Steller database.

