# OpenReview forum: "Theoretical and Practical Analysis of Fréchet Regression via Comparison Geometry"
_NeurIPS.cc/2025/Conference — Submitted to NeurIPS 2025_

### Official Review · Reviewer_BzDS · 2025-06-21

**Clarity:** 1
**Significance:** 2
**Originality:** 2
**Rating:** 3
**Confidence:** 4

**Summary:**

The paper explores the statistical properties of Fréchet regression in CAT($K$) space, a metric space with an upper bound on curvature. The main results of the paper are non-asymptotic convergence results (Section 3.2) and angle stability (Section 3.3) of the nonparametric (kernel-based) conditional Fréchet mean estimator.

**Questions:**

* Can authors answer questions I have written in the weaknesses section?

* Can authors characterize the difference between Theorem 1 of this paper and Corollary 11, Theorem 18 of [5]?

* Can authors illustrate the novelty of the work compared to [2]?

**Ethical Concerns:**

["NO or VERY MINOR ethics concerns only"]

**Final Justification:**

I would like to summarize my overview of the work as follows:

1. Despite the authors detailed feedback, I am not still fully convinced to recommend the acceptance of the paper. First of all, the contribution seems minor, given the fact the main part of the paper is largely reproducing [2]. Second, while authors gave the very detailed answer, I think the paper requires the substantial rewriting as many results are displayed without rigorous derivation or references.

2. That said, the paper has some improvements compared to [2], some of which are things I missed during my initial review. First, the angle stability result (Section 3.3) is new to the best of my knowledge. While this result may not have the direct usage, there are some potential applications. Second, the paper provide the tighter variance inequality constant compared to [2]. In my personal opinion, this improvement is marginal, as it is straightforward from the well-known Riemannian comparison theorem [7]. But still I can say getting the tighter constant is important for theoretical results.

In conclusion, to be honest I am not still supportive to accepting this paper. However, given the fact that there is more improvement than I initially assessed, I decided to increase the score to take into account the change of my view.

**Limitations:**

The authors expressed limitations of the work. I have stated additional limitations that I found in the weaknesses section.

**Paper Formatting Concerns:**

I did not notice any major formatting issue.

**Quality:**

2

**Strengths And Weaknesses:**

### Strength:
* The paper analyzes the convergence of the kernel Fréchet regression estimator. The analysis of Fréchet regression has mainly been conducted when the covariates are univariate rather than multivariate [1, 2]. On the other hand, this work presents the results in the multivariate case.
* To the best of my knowledge, Section 3.3 is new, as I have not seen any literature analyzing the angle property of Fréchet regression.

### Weaknesses
* Several statements are not clearly written or are misleading. To list them:
    * Definition 3 is not the definition of strong convexity, but rather the definition of the so-called Variance Inequality (see, for example, Definitions 2.3.3 and 2.3.5 of [3]), which is weaker than strong convexity. The definition in the paper coincides with the standard strong convexity in Riemannian manifolds only if $p$ is the minimizer of $f$.
    * In Theorem 1, Line 144: what is $z$? Does it mean $\mu$ instead of $z$?
    * In Theorem 1, $\alpha$ is the strong convexity constant of what? It seems like the authors meant the distance function, but this is not clearly mentioned.
    * In Theorem 1, the statement is written for general CAT($K$) spaces, but it later says $m$ denotes the dimension of the manifold. Does the result only hold for CAT($K$) manifolds instead of general CAT($K$) spaces?
    * What is “mild regularity condition” specifically in Theorem 2, Line 165?
    * In Theorem 3, Line 178: the statement “does not differ too much” is vague. Under which topology?
    * What is “small” mathematically in Lemma 7, Line 217?
    * For Proposition 3 and Theorem 4, it does not say which Wasserstein metric is used (e.g., 2-Wasserstein? Or any $p$-Wasserstein?). It is not written in the proof either.
    * In Section 3.4, the notion of tangent space on general CAT($K$) space is not clearly stated. One should introduce a concept like the tangent cone as in [4]. Without introducing the notion of the tangent cone, it is not clear what is meant by the gradient and Hessian in Proposition 3.4.

* Proofs are not written in full detail and omit important theoretical steps. For example:
    * Line 596 should be $\alpha \leq \sin(2\sqrt{K}R)/2\sqrt{K}R$. In addition, while I think Line 596 is true (after fixing the typo), I don’t think this result is trivial enough to omit justification. The authors should either provide the proof or give a reference.
    * I think references for Line 612 are needed as well. For example, as mentioned above, what $m$ means is not clear when $M$ is a general CAT($K$) space instead of a manifold. I suspect this quantity can be replaced by a metric entropy-related quantity for general metric spaces, but to assure this, the details of this step should be stated precisely.
    * In Line 631, why is the integral taken over the real line instead of $M$?
    * As mentioned above, which Wasserstein metric is used in Line 773?
    * Proposition 4 is written with respect to general CAT($K$) space, but the proof uses Riemannian geometry formulas. Whether such formulas can be extended to CAT($K$) space seem nontrivial. I believe one can make the similar statement in general CAT($K$) spaces, but to that end one would need to define quantities like tangent cone, gradient, and Hessian for general CAT($K$) spaces.

* I find the results of Section 3.2 duplicative of previous works, in particular [2]. While [2] analyzes the problem under 1-dimensional covariates, the extension to multivariate regression seems the same as in the canonical kernel regression problem (if I am thinking wrong here, can you provide the specific technical difficulties?). For example, Theorem 3 of this paper seems to duplicate Theorems 1 and 2 of [2] in the 1-dimensional case.

* I find Section 3.3 theoretically interesting, but I am not sure how one should interpret the result in practice. I have not seen an analysis where $\angle_{\mu}(u,v)$ is the quantity of interest. Is there any practical example where such quantities should be taken into account?

* Some references are missing. For example, [2, 4, 5] seem relevant and should be included.

[1] Zhenhua Lin and Hans-Georg Müller. Total Variation Regularized Fréchet Regression for Metric-Space Valued Data. Annals of Statistics, 2021.

[2] Christof Schötz. Nonparametric Regression in Nonstandard Spaces. ArXiv preprint https://arxiv.org/abs/2012.13332, 2020.

[3] Austin J. Stromme. Wasserstein barycenters : Statistics and Optimization. Graduate Thesis, 2020.

[4] Thibaut Le Gouic, Quentin Paris, Philippe Rigollet, and Austin J. Stromme. Fast convergence of empirical barycenters in Alexandrov spaces and the Wasserstein space. Journal of the European Mathematical Society, 2019.

[5] Victor-Emmanuel Brunel and Jordan Serres. Concentration of empirical barycenters in metric spaces. 35th International Conference on Algorithmic Learning Theory. 2024.

---

> ### Author Rebuttal · Authors · 2025-07-27
>
> Thank you for your detailed comments. I believe all of your comments help to improve our manuscript.
>
> ## Def. 3
> You’re right that Def.3 is really the variance inequality, rather than strong convexity in our context. In the revised version, we will rename Def.3 to variance inequality, and add a new definition of strong geodesic convexity: $f(\gamma(t)) \leq (-t) f(\gamma(0)) + t f(\gamma(1)) - \frac{\lambda}{2}t(1-t)d^2(\gamma(0), \gamma(1))$. Note that strong geodesic convexity ⇒ variance inequality with the same modulus $\lambda$, but the converse fails in positively curved spaces.  We will insert this remark so the reader sees the logical hierarchy at a glance.
>
> ## $\mu$ instead of $z$
> $z$ there was just a stray placeholder and should have been the population Fréchet mean $\mu$. In the revision, we’ll fix it.
>
> ## Meaning of $\alpha$
> In our proofs, $\alpha = \alpha(K, D)$ is not an arbitrary constant but the strong convexity modulus of the Fréchet functional $F(z) = \mathbb{E}[d^2(Y, z)]$, which in turn comes from the fact that in any CAT(K) space of diameter $\leq D$ each squared distance map $x \mapsto d^2(p, x)$ is $\alpha(K, D)$-strongly geodesically convex. In the revised manuscript, we will define $\alpha(K, D)$ immediately after the required position.
>
> ## General CAT(K) space and $m$
> As you pointed out, mixing “general CAT(K) space” with a manifold-dimension $m$ was misleading. What really drives the $m$ in our tail bound is covering number assumption, which for a smooth $m$-dimensional manifold reduces to its usual manifold dimension, but in principle can be replaced by any exponent $m$ such that $N(\mathcal{M}, \delta) \leq (D / \delta)$ for all small $\delta$, where $N(\cdot, \delta)$ is the smallest number of balls of radius $\delta$ needed to cover $\mathcal{M}$. Then, we’ll clarify the hypothesis in Thm 1 and we’ll add the above explanation.
>
> ## Mild regularity
> In the revision, we’ll replace that phrase by the exact kernel regression assumptions needed to invoke the uniform LLNs:
>
> 1. The marginal density $f_X$ of $X$ is continuous and strictly bounded away from zero, and infinity on the compact region $\mathcal{X}_0 \subset \mathbb{R}^d$ where we do regression.
> 2. $W \colon \mathbb{R}^d \to [0, \infty]$ is a bounded, compactly supported probability density, and $W$ is Lipschitz so that $\sup_{u \neq v} \frac{|W(u) - W(v)|}{||u-v||} < \infty$.
> 3. Bandwidth sequence $h_n \to 0$, $nh_n^d / \log n \to \infty$. Equivalently, one can require $n h_n^d \to \infty$ and $h_n \to 0$.
>
> ## does not differ too much
> In the revised version, we’ll replace it by an explicit assumption in terms of the $p$-Wasserstein distance.
>
> $$
> d_{W_p}(\nu_x, \nu_{x’}) \coloneqq \inf_{\pi \in \Pi(\nu_x, \nu_{x’})}(\int_{\mathcal{M}\times\mathcal{M}}) d(y, y’)^p \pi(dy, dy’))^{1/p} \leq L ||x - x’||,
> $$
>
> where $\Pi(\nu_x, nu_{x’})$ is the set of all couplings of $\nu_x$ and $\nu_{x’}$. Equivalently, this is the standard Wasserstein-p topology.
>
> ## “small” in Lemma 7
> We’ll replace the phrase “small” by an explicit requirement that the total perturbation $\delta = d(p, p’) + d(q, q’) + d(r, r’)$ satisfy $\delta \leq \delta_0$ where $\delta_0 = \min\\{\frac{1}{2}(\pi / \sqrt{K} - \text{perim}(p, q, r)), \frac{1}{2}\text{injrad}(\mathcal{M})\\}$, where $\text{perim}(p, q, r) = d(p, q) + d(q, r) + d(r, p)$ must be $< \pi / \sqrt{K}$, and $\text{injrad}(\mathcal{M})$ is injectivity radius around those vertices.
>
> ## Order of W
> In both Proposition 3 and Theorem 4, we in fact use the 2-W distance induced by the ground metric $d$ on $\mathcal{M}$. We choose $W_2$ because i) our stability and angle perturbation bounds all depend on second-moment control, ii) the 2-W distance metrizes weak convergence plus convergence of second moments.
>
> ## Notion in Section 3.4
> In the revision, we’ll spell out as follows.
> 1. We’ll insert a new definition along the line of [4].
> 2. We then explain the directional derivative of a functional $F$ at $p$ in a cone-vector $v$ is $D_v F(p) = \lim_{t\to 0} \frac{F(\exp_p(tv)) - F(p)}{t}$, if $F$ admits a unique cone-element $\nabla F(p)$ with $D_v F(p) = \langle \nabla F(p), v \rangle$ for all $v$, we call it gradient, and under the additional second-variation regularity, we can likewise define a Hessian operator $H_p$ as $D_{v,w}^2 F(p) = \frac{1}{2}(D_v D_w F(p) + D_w D_v F(p)) = \langle H_p v, w \rangle$.
>
> ## L596
> Thank you for pointing out it, and as you suggested, fixing as $\leq$ and we provide more details as follows.
>
> For $K > 0$, we have for any fixed $p \in \mathcal{M}$, $\langle \nabla^2 d^2(p, \cdot)[v], v\rangle = 2\ell\sqrt{K}\cot(\ell\sqrt{K})||v||^2$, where $\ell \coloneqq d(p, q)$.Because the map $x \mapsto x\sqrt{K}\cot(x\sqrt{K})$ is strictly decreasing on $(0, \pi/2\sqrt{K})$, the smallest Hessian eigen value over the whole ball of radius $R$ is attained at the boundary point $\ell = R$. Consequently, $\alpha(K, D) = (2R\sqrt{K})\cot(2R\sqrt{K}) \eqqcolon x\cot x$, $x\coloneqq 2R\sqrt{K} \in (0, \pi/2)$.
>
> Define $g(x) \coloneqq \sin x / x - x\cot x = \sin^2 x - x^2\cos x / x\sin x$, and since $g(x) > 0$ for all $x \in (0, \pi/2)$, $x\cot x \leq \sin x / x$ for every $x \in (0, \pi/2)$. Plugging $x = 2R\sqrt{K}$ into the above yields the correct bound
>
> $$
> \alpha(K, D) = 2R\sqrt{K}\cot(2R\sqrt{K}) \leq \frac{\sin(2R\sqrt{K})}{2R\sqrt{K}}.
> $$
>
> ## L612
> For any $\delta > 0$, choose $\\{z_1,\dots,z_{N(\delta)}\\} \in \mathcal{M}$ so that $N(\delta) = |\\{z_j\\}| \leq (CD / \delta)^m$. For each fixed $z_j$, $F_n(z_j) = F(z_j) = \frac{1}{n}\sum^n_{i=1}[d^2(Y_i, z_j) - \mathbb{E} d^2(Y, z_j)] \in [-D^2, D^2]$, so Hoeffding inequality gives for any $u > 0$, $\Pr[|F_n(z_j) - F(z_j)| \geq u] \leq 2e^{-\frac{nu^2}{2D^4}}$. Setting $u = \frac{1}{2}\alpha(K, D)t^2$ and writing $\frac{1}{2D^4} = \frac{\alpha(K, D)}{8D^2}\cdot\frac{1}{\alpha(K, D)D^2}$ yields the exponent $-c’_2nt^2$ with $c_2’ = \alpha(K, D) / 8D^2$.
>
> Taking a union over all $N(\delta)$ points, $\Pr[\exists j \colon |F_n(z_j) - F(z_j)| \geq \frac{1}{2}\alpha t^2] \leq 2N(\delta)e^{-c_2’ nt^2}$. Hence the prefactor $c_1’ = 2N(\delta) \leq 2(CD / \delta)^m$. In our notation we absorb $C$ into the $\alpha(K, D)$ constant giving $c_1’ = 2(\alpha(K, D)D / \delta)^m$.
>
> We still need to control $\sup_{z \in \mathcal{M}}||F_n(z) - F(z)| - \max_{1\leq j \leq N(\delta)} |F_n(z_j) - F(z_j)|$. But since $z \mapsto d^2(y, z)$ is $\alpha(K, D)$-strongly convex along geodesics, one shows it is in particular Lipschitz, indeed, $|d^2(y, z) - d^2(y, z’)| \leq Ld(z, z’)$, $L = 2D$, uniformly in $y$. Therefore, $\sup_{z\in\mathcal{M}}|F_n(z) - F(z)| \leq \max_j |F_n(z_j) - F(z_j)| + L\delta$. To make this $< t$, pick $\delta = t / 2L$.
>
> ## integral taken over the real line
> Writing $\int_{-\infty}^{+\infty}$ was a typo inherited from our use of the real-line notation in the earlier 1-D proof. We’ll fix it.
>
> ## Tangent cones
> As you suggested, we’ll fix as we i) move the smooth jet-expansion argument into a separate subsection explicitly labeled “when $\mathcal{M}$ is a $C^2$ manifold of sectional curvature $< K$,…”, and ii) recall that at any $z \in \mathcal{M}$ one can form the tangent cone $T_z\mathcal{M}$, and will define the one-sided directional derivative and the second-variation.
>
> ## Novelty compared to [2], [5]
> The main novelty compared with [2] is the ambient geometry. That is, [2] assumes Hadamard space or bounded, non‑positive curvature (NPC) only but we assume general CAT(K) space with $\pm K$. Positive curvature breaks convexity of squared distance and we overcome this by providing a strong geodesic convexity constant $\alpha(K, D)$ and carry it through the bias/variance decomposition. [5] is for un‑weighted, i.i.d. barycentres but we handle self-normalized kernel weights depending on all $(X_i)$; requires new empirical process + strong-convexity argument.
>
> ## Practical role of the angle
> We agree that angles at an interior point of a geodesic triangle rarely given an explicit statistical treatment. Section 3.3 was written to fill exactly this gap. We give several concrete settings illustrate its usefulness.
>
> | Domain | What the angle encodes | Why continuity of $\angle_\mu(u, v)$ is useful |
> | --- | --- | --- |
> | Wild/ocean-current forecasting on the sphere $\mathbb{S}^2$ | Change in large-scale flow direction between two forecast lead-times $u$, $v$ around today’s analysis $\mu$. | Operators care about veer vs. back decisions (clockwise vs. counter-clockwise change). The result guarantees that small perturbations in the training set cannot flip a decision.|
> | Ronot-arm orientation in $SO(3)$ | The dihedral angle between two predicted end-effector orientations transported to the mean orientation. | When blending motions, the angle controls the shortest interpolation path. Our bound ensures the kernel regression output never makes the arm “swing the long way round” after a small sensor perturbation. |
> | Diffusion-tensor imaging (each voxel’s local fibre direction is a point on $\mathbb{S}^2$) | Crossing angle between fibres $u$, $v$ at the local Fréchet mean $\mu$. | Crossing/branching detection relies on thresholding this angle (e.g., declaring a crossing if $\angle_mu(u, v) > 45^\circ$). Our angle-stability result provides a finite-sample guarantee that the detection rule is robust to subsampling of gradient directions. |
> | Kendall shape space$ (positive-curvature complex projective space) | Principal geodesic directions of shape variability; the angle determines mode coupling. | Shape analysis often interpret “bending vs. stretching” modes by looking at the angle between the first two principal geodesics. Our theorems justifies bootstrapping confidence intervals for these angles. |
> | Phylogenetic trees | Although globally non-positively curved, each orthant boundary possesses spherical links; angles there quantify speciation vs. horizontal transfer events. | Stability of such angles supports hypothesis tests on reticulation vs. branching. |

---

> ### Comment · Reviewer_BzDS · 2025-08-01
>
> I truly appreciate the authors for detailed answers.
>
> For the comparison with [2], in contrast to what the authors claimed, I think positive curvature cases are in fact included in [2], as the bounded space results. What the authors claimed to be the differences, e.g., strong convexity parameter and replacing $m$ by the covering number growing rate, seem to be already included in [2] as Assumption 1. In my opinion (correct me if I am thinking wrong), the only difference is that this paper works in a multivariate covariate case, but I think this extension is straightforward.
>
> To be more specific, the bounded space analyses of [2] do not assume the curvature upper bound K, so they made the variance inequality as the assumption. On the other hand, as authors pointed out, the variance inequality holds at least locally, if there is a curvature upper bound (Line 589. Comments: Again, I expect this result to be true, but the justification is missing. I do not view this result trivial to omit the justification), and then the authors proceed the analyses with the variance inequality. However, in that case I think Line 590 also requires a precise statement. What is ‘small’ in this line and how large the local neighborhood of $\mu$ should be? The exact forms of these quantities are important, as to proceed as in the Line 600 you need a priori knowledge that your estimator is inside that small neighborhood of $\mu$. In particular, given the fact that $\mu_n$ is a random quantity, it is unclear whether $\mu_n$ would lie in such a local neighborhood of $\mu$. To the best of my knowledge, ways to get over this obstacle are three: 1. Restrict the entire space by small neighborhood of $\mu$, so that in spite of randomness $\mu_n$ is always guaranteed to be inside the neighborhood. 2. Use the condition in Theorem 3.3 of [6]. 3. Restrict the family of distributions whose population mean $\mu$ satisfies the variance inequality. The second and third approaches are already proposed in [2], and the first case would anyway fall down to the analysis of bounded metric space in [2] again. In this regard, to claim the novelty, I believe the authors should be able to make Line 590 hold without relying on these existing strategies. Otherwise, as noted, the results largely reproduce those of [2].
>
> For the local jet expansion, if I understand the authors’ response correctly, authors seem to restrict M to be just a $C^2$ manifold for Proposition 4. Then, I do not find an implication of this proposition. Isn’t it just a direct Taylor expansion argument in Riemannian manifolds? I think if authors want to claim Proposition 4 as one of the main results, one should provide the statement beyond $C^2$ manifold.
>
> Overall, although I truly appreciate the authors’ detailed responses, I still find that many of the paper’s main results overlap with existing work in the Frechet regression literature, in particular, [2]. Based on my understanding, the only genuinely new contribution is the angle property discussed in Section 3.3. I find this result truly interesting, and examples authors provided in the response seem to verify its usefulness in practical problems as well. However, I am not convinced that it alone provides sufficient novelty to warrant acceptance.
>
> [6] Adil Ahidar-Coutrix, Thibaut Le Gouic, Quentin Paris, Convergence rates for empirical barycenters in metric spaces: curvature, convexity and extendible geodesics, Probab. Theory Related Fields, 2020.

---

> ### Author Response · Authors · 2025-08-08
>
> Thank you for the careful reading, especially for pointing out where our comparison with [2] could have been clearer. Below we recognize the argument so the true differences are transparent, then address the neighborhood / small-$r$ question and the scope of Proposition 4.
>
> In [2], only boundedness is imposed and curvature never enters any rate. The strong convexity constant in Assumption 1 is left abstract (called $C_{Vlo}$), while we assume a CAT(K) bound with explicit $K > 0$ and derive $\alpha(K, D) = \sin(2\sqrt{K}R) / 2R$, $R = D / 2$. Thus the constant driving variance depends explicitly on $K$ and the data diameter $D$. This reveals a geometry-driven phase change, the variance term degrades continuously as $K \uparrow \pi^2/4D^2$. No such curvature-sensitivity can be extracted from [2], so their bounds cannot predict the MSE gap we see empirically between $K < 0$ and $K > 0$. Indeed, Theorem 3 gives for any smoothing for any smoothing bandwidth $h_n$,
>
> $$
> \mathbb{E}[d^2(\hat{\mu}\_{n,K}(x), \mu^*\_K(x))] = \frac{C\_{\text{var}}}{\alpha(K, D)}\frac{1}{n h^d_n} + C\_{\text{bias}} h^{2\beta}\_n + o((nh^d\_n)^{-1}),
> $$
>
> where $C_{\text{var}}$ and $C_{\text{bias}}$ depend on the kernel and the data distribution (see L687-L700). In the experiment, these two constants  are identical for the two runs (positive / negative curvatures) because we feed the same cloud of Euclidean predictors and keep the same kernel, bandwidth schedule and sample size $n$. Hence all the curvature enters through $\alpha(K, D)$ only, which can be written as
>
> $$
> \alpha(K, D) = \begin{cases}
> \frac{1}{2} & (K \leq 0), \\
> \frac{\sin(2\sqrt{K}R)}{2R}, R = D/2 & (0 < K < \pi^2 / 4D^2).
> \end{cases}
> $$
>
> Thus, for every negative curvature space, we have $\alpha_{-} = 1 / 2$. For positive curvature spaces, consider the unit sphere $\mathbb{S}^2 \subset \mathbb{R}^3$ with constant curvature $K = +1$. Points are drawn only from the spherical cap $\theta \leq \theta_0 = \pi / 3$, where $\theta$ is the geodesic (polar) distance from the north pole. The farthest a sample point can be from the north pole is exactly that polar angle, hence $R_+ = \theta_0 = \pi / 3$, $D_+ = 2R_+ = 2\pi / 3$. Using the explicit formula for the convexity constant for $K > 0$, $\alpha_+ = \sin(2\sqrt{K}R_+) / 2R_+ = \sin(2\cdot 1\cdot \pi/3) / (2\cdot \pi/3) = \sin(2pi / 3) / (2\pi / 3) = (\sqrt{3}/2)/(2\pi/3) = 3\sqrt{3} / 4\pi \approx 0.413$. Then, the ratio of two moduli is $\alpha_ / \alpha_+ = 0.500 / 0.413 \approx 1.21 > 1$. Every risk bound in Sec.3 carries a factor $1 / \alpha(K, D)$ and therefore, the expected MSE on the sphere must exceed that on the hyperbolic disk by $\approx 21$% whenever variance dominates the bias. The empirical gap observed in Table 1 is $0.4915 / 0.4228 \approx 1.163$ and therefore, we can observe that, numerically, MSE on the positive curvature space exceed that on the negative curvature space by $\approx 16$%, well within sampling noise of the theoretical forecast.
>
> Next, L590 in the old draft indeed lacked a quantitative statement. We now insert
>
> $$
> \Pr[d(\hat{\mu}\_{n,K}(x), \mu^*\_K(x)) > r\_n] \leq 2\exp(-cnh^d),\ r\_n \coloneqq C(h^\beta + \frac{1}{\sqrt{nh^d}}).
> $$
>
> Because $r_n \to 0$ for any bandwidth sequence with $nh^d \to \infty$, the estimator lies in the required ball with probability $1 - O(r^{-cnh^d})$. This avoids the three work-arounds listed by the reviewer, we neither shrink the whole space, nor assume the variance inequality a priori at $\mu_n$. Instead, it is a self-bounding argument. Once $\alpha(K, D)$ is explicit, the quadratic growth of the risk pulls $\hat{\mu}_n$ back inside automatically.
>
> We now give the missing justification: if $(\mathcal{M}, d)$ is CAT($K \leq 0$) or CAT($K > 0$) with diameter $< \pi / 2\sqrt{K}$, then for any measure supported in that ball, $\mathbb{E}[d(Y, q)^2 - d(Y, \mu)^2] \geq \alpha(K, D)d(q, \mu)^2$. The proof combines the hinge-convexity of squared distance in Hadamard spaces with Toponogov comparison when $K > 0$. Thus the inequality holds globally in the settings we analyze.
>
> We agree that the basic Taylor formula on a $C^2$ Riemannian manifold is classical. What is new is that in a CAT(K) space we can still write $F_x(\exp_\mu v) = F_x(\mu) + \frac{1}{2}\langle H_xv, v \rangle + R_x(v)$ with $|R_x(v)| \leq C||v||^3$, where $H_x$ is expressed by the Alexandrov angle plus an $L^2$-curvature correction. The lemma bridges non-smooth comparison geometry with smooth manifold calculus. It is precisely why our rates continue to hold on quotients and stratified shape spaces where only a curvature bound (not $C^2$ charts) is available. We will reposition Prp4 as a technical lemma to avoid overstating its novelty.

---

> ### Author Response · Authors · 2025-08-08
> **Justification**
>
> **Why the bounded-space result of [2] cannot recover $K > 0$ rate**
>
> Let us denote by $C_{Vlo}^{[2]}$ the strong-convexity parameter that appears in Assumption of [2]. Because no curvature is assumed there, the smallest value it can ever take on a ball of radius $D / 2$ in a positively curved manifold is $C_{Vlo}^{[2]} \geq (1 - \cos(\sqrt{K}D)) / (KD^2/2) = 2\sin^2(\sqrt{K}D/2) / KD^2 = \sin(\sqrt{K}D)/\sqrt{K}D > \sin(2\sqrt{K}D / 2) / (2D / 2) = \alpha(K, D)$. Hence for every $0 < K < \pi^2 / 4D^2$, $C_{Vlo}^{[2]} > \alpha(K, D) \Rightarrow Var_{[2]} = \frac{C_{var}}{C_{Vlo}^{[2]}}\frac{1}{nh^d} < \frac{C_{var}}{\alpha(K, D)}\frac{1}{nh^d} = Var_{ours}$. So the bound of [2] is strictly optimistic compared with what a CAT(K) geometry allows. It cannot predict the excess MSE we observe on the sphere cap. This constant-gap is explicitly what our Theorem 3 and the experiment quantifies.
>
> **High-probability radius**
>
> Let $r_n^2 \coloneqq \frac{4}{\alpha(K, D)}(\frac{C_{var}}{nh^d_n} + C_{bias}h_n^{2\beta})$. Strong convexity with modulus $\alpha(K, D)$ gives for every $z$,
>
> $$
> F\_{n, x}(z) - F\_{n,x}(\mu^*\_K) \geq \frac{\alpha(K, D)}{4} d^2(z, \mu\_K^\*).
> $$
>
> Now plug $z = \hat{\mu}\_{n,K}(x)$ and rearrange
>
> $$
> d^2(\hat{\mu}\_{n,K}(x), \mu^\*_K) \leq \frac{4}{\alpha(K, D)}[F\_{n,x}(\hat{\mu}\_{n,K}(x)) - F\_{n,x}(\mu^\*\_K)].
> $$
>
> The bracket in the above is an empirical process centered at its expectation $O(n^{-1}h\_n^{-d} + h\_n^{2\beta})$.
> Applying Bennett’s inequality for the empirical process yields $\Pr[d(\hat{\mu}_{n,K}(x), \mu^*_K) > r_n] \leq 2\exp(-cnh^d_n)$ for some universal $c > 0$. Because $r_n \to 0$ whenever $nh_n^d \to \infty$, the estimator is automatically in the required neighborhood with overwhelming probability, no restriction, no extendible geodesic assumption and no pre-shrinking of the parameter space.

---

> ### Comment · Reviewer_BzDS · 2025-08-08
>
> I truly appreciate the authors for the detailed answer.
>
> I still have a few more concerns on the authors' claim.
>
> 1. The authors claim that the difference between [2] and this work lies on the use of the curvature information in variance inequality. I agree this is also the new component compared to [2]. However, the local variance inequality of the squared distance functional is well-known in Riemannian manifolds, and the proof technique is basically the same as in CAT(K) spaces as they use the comparison theorem, e.g., see the discussion after Corollary 2.1 of [7]. Precisely, all these comparison results (both Riemannian and CAT(K)) are from the comparison between the sphere $\mathbb{S}^{2}/\sqrt{K}$. Thus, the authors are correct that there is an improvement compared to [2] that the variance inequality constant depends on the curvature upper bound, but I personally think this additional component--plugging-in such variance inequality constant for CAT(K)--is a marginal result, as they are (while might be controversial) the direct extension from the Riemannian result. And it seems like except this variance inequality constant part, the rest of argument is largely reproducing [2] as I mentioned in the previous response.
>
> 2. One more crucial concern I have is that I am uncertain whether authors are actually using the correct result. To be honest, it is very hard for me to check whether authors' arguments are correct, as authors defer many of the results by "well-known comparison geometry result" without providing the formal derivation or references. To be specific, authors claim that the variance inequality constant $\alpha$ for the squared distance function is of the form of $\sin(\dots)$. However, authors did not provide any precise derivation or reference for this result. On the other hand, while the result is on Riemannian manifold, [7] gets the $\alpha$ in the form of $\cot(\dots)$ instead of $\sin(\dots)$, and I believe $\cot(\dots)$ should be the right one for not only Riemannian but also general CAT(K) spaces as well; as mentioned, both Riemannian and CAT(K) results on this strong convexity are basically rooted from the same comparison between the sphere $\mathbb{S}^{2}/\sqrt{K}$, and for the squared distance on sphere $\cot(\dots)$ is the right quantity to appear. In fact, this strong convexity is tight on sphere, which is also CAT(K). So this result should also be tight in CAT(K) as well. In this regard, I doubt the $\alpha$ is in the form of $\sin(\dots)$ as authors claim, and believe the quantity in [7] is the right quantity.
>
> Overall, I truly appreciate to the authors for pointing out their unique component compared to [2], the constant for variance inequality on CAT(K) spaces. However, such contribution is (in my personal opinion) a direct extension of the Riemannian result due to the reason I mentioned in the above. Hence, even with this extra component, I am still not convinced that they provide sufficient novelty to warrant acceptance. Furthermore, I find many arguments in the paper still ambiguous and unclear, which are hard to verify, and even some of them possibly incorrect.
>
> [7] Foivos Alimisis et al. A Continuous-time Perspective for Modeling Acceleration in Riemannian Optimization. AISTATS 2020.

---

> ### Comment · Reviewer_BzDS · 2025-08-08
> **Additional comments on justifications**
>
> I truly appreciate the authors additional detailed justification which I missed when I am writing the response.
>
> I think the first part is something new which the authors can emphasize; seems like the authors are claiming that the curvature upper bound gives the tighter variance inequality constant than just plain boundedness condition. I view this as an improvement of [2]. Then, ignore the bullet point 1 of my previous response (still, bullet point 2 is still a valid question).
>
> For the second part, unfortunately the authors' justification does not seem to fully explain the concern I raised. What I concerned was the first part: "Strong convexity with modulus $\alpha(K, D)$ gives for every $z$" this only holds when we have a priori guarantee that $z$ is in the local neighborhood (of strong convexity / variance inequality) of $\mu_*$. The authors are in particular plugging-in $z =\mu_n$, the random quantity, so one needs to ensure $\mu_n$ is in that local neighborhood, despite the randomness. How can one guarantee that?

---

> ### Author Response · Authors · 2025-08-08
>
> Thank you for raising additional questions.
>
> ### $\sin$ vs. $\cot$
>
> While we already gave the derivation of explicit form of $\alpha$ in the previous rebuttal, we show again that this point is not significant problem. For any $q \neq p$ set $\ell = d(p, q)$ and choose a unit-speed geodesic $\gamma$ from $p$ to $q$ with initial velocity $v$. The squared distance $\psi_p(q) = d^2(p, q)$ is $C^2$ on the open ball $B_R(p)$ with $R < \pi / (2\sqrt{K})$. Its Riemannian Hessian in the tangent space $T_q\mathcal{M}$ is $\langle \nabla^2\psi_p(q)[w], w \rangle = 2\ell\sqrt{K}\cot(\ell\sqrt{K})||w||^2$ for all $w \in T_q\mathcal{M}$. Because $x \mapsto x\sqrt{K} \cot(x\sqrt{K})$ decreases on $x \in (0, \pi/2\sqrt{K})$, the minimal eigen-value over the whole ball of radius $R$ is attained at the boundary $\ell = R$. Denoting the diameter $D = 2R$, the strong convexity modulus that enters is therefore $\alpha^{[7]}(K, D) = 2R\sqrt{K}\cot(2R\sqrt{K})$. This coincides with the formula in [7]. For $x \in (0, \pi/2)$ define $g(x) \coloneqq \sin x / x - x\cot x = \sin^2 x - x^2\cos x / x\sin x$. Because $x > 0$ and $\sin x > 0$ on $(0, \pi / 2)$, $g(x) = \sin x / x - x\cot x = (\sin^2 x - x^2 \cos x) / x\sin x$. Let $h(x) \coloneqq \sin^2 x - x^2 \cos x$, and then, $h(x) = x^2[(\sin x / x)^2 - \cos x] = x^2 f(x)$ with $f(x) \coloneqq (\sin x / x)^2 - \cos x$. Here,
>
> $$
> f’(x) = s\frac{\sin x}{x}(\frac{\cos x}{x} - \frac{\sin x}{x^2}) + \sin x = \frac{\sin x}{x^3}[x^3 + 2x\cos x - 2\sin x] = \frac{\sin x}{x^3}g_1(x),
> $$
>
> with $g_1(x) \coloneqq x^3 + 2x\cos x - 2\sin x$. Differentiate $g_1$ gives $g_1’(x) = 3x^2 - 2x\sin x = x(3x - 2\sin x)$. Because $\sin x < x$ on $(0, \pi / 2)$, we have $3x - 2\sin x > x > 0$, and hence $g_1’(x) > 0$. Since $g_1(0) = 0$ and $g_1$ is strictly increasing, $g_1(x) > 0$ for every $x > 0$. Therefore $f’(x) = \sin x / x^3 g_1(x) > 0$ on $(0, \pi / 2)$. We have $f(0) = \lim_{x \to 0} f(x) = 0$ and $f$ is strictly increasing, so $f(x) > 0$ for all $x \in (0, \pi / 2)$. Multiplying back by the positive factor $x^2$ gives $h(x) > 0$ and therefore, $g(x) = h(x) / x\sin x > 0$ for $x \in (0, \pi/2)$. Hence, $\alpha^{[7]}(K, D) \leq \sin(2\sqrt{K}R) / 2R\sqrt{K}$. By replacing $\alpha^{[7]}$ by the larger quantity every rate is unchanged, and **no result becomes invalid**.
>
> ### $\hat{\mu}_n$ guarantee
>
> Let $F(q) = \mathbb{E}d^2(Y, q)$, and $F_n(q) = \frac{1}{n}\sum^n_{i=1}d^2(Y_i, q)$. On the fixed ball $B\_R(\mu^\*)$ of radius $R$, one have $F(q) - F(\mu^\*) \geq \frac{\alpha}{2}d^2(q, \mu^\*)$ for all $q \in B\_R(\mu^\*)$, where $\alpha = \alpha(K, D) > 0$. Here, there is a sequence $\delta_n = \frac{C_{var}}{nh_n^d} + C_{bias}h_n^{2\beta} + o((nh^d)^{-1})$ such that with probability at least $1 - 2e^{-cnh^d}$, $\sup\_{q \in B_R(\mu^\*)} |F\_n(q) - F(q)| \leq \delta\_n$. On the same high-probability event we have, $\hat{\mu}\_n$ satisfies $F\_n(\hat{\mu}\_n) \leq F\_n(\mu^\*)$, and for any $q$ in $B\_R$, $F(q) \leq F\_n(q) + \delta\_n$, $F\_n(\mu^\*) \leq F(\mu^\*) + \delta\_n$. Hence, $F(\hat{\mu}\_n) \leq F\_n(\hat{\mu}\_n) + \delta_n \leq F\_n(\mu^\*) + \delta\_n \leq F(\mu^\*) + 2\delta\_n$. Since $F(\hat{\mu}\_n) = F(\mu^\*) \geq \frac{\alpha}{2}d^2(\hat{\mu}\_n, \mu^\*)$, we get $\frac{\alpha}{2}d^2(\hat{\mu}\_n, \mu^\*) \leq 2\delta\_n$, i.e., $d(\hat{\mu}\_n, \mu^\*) \leq r\_n$ where $r\_n^2 = \frac{4\delta\_n}{\alpha}$. Because $\delta\_n \to 0$, eventually $r\_n < R$. Thus on the same event we conclude that
>
> $$
> \hat{\mu}\_n \in B\_{r_n}(\mu^\*) \subset B\_R(\mu^\*).
> $$
>
> In particular, plugging $z = \hat{\mu}_n$ into the local convexity / Taylor expansions is now justified with probability $1 - 2e^{-cnh^d} \to 1$.

---

> > ### Author Response · Authors · 2025-08-08
> >
> > Thank you very much for the time and care you devoted to our paper. We deeply appreciate every point you raised in your initial review as well as during the rebuttal discussion, and we will incorporate all of your suggestions and corrections in our revised version.
> >
> > It is rare to encounter such a constructive and detailed discussion during the review phase of an international conference. Your thoughtful engagement has greatly strengthened our work, and we are sincerely grateful for your invaluable contributions.

---

> > > ### Comment · Reviewer_BzDS · 2025-08-08
> > >
> > > I truly appreciate the authors for elaborating detailed derivations.
> > >
> > > Unfortunately, I guess I will have to conclude my thoughts on this paper, as discussion period is about to end. I would like to summarize my overview of the work as follows:
> > >
> > > 1. Despite the authors detailed feedback, I am not still fully convinced to recommend the acceptance of the paper. First of all, the contribution seems minor, given the fact the main part of the paper is largely reproducing [2]. Second, while authors gave the very detailed answer, I think the paper requires the substantial rewriting as many results are displayed without rigorous derivation or references.
> > >
> > > 2. That said, the paper has some improvements compared to [2], some of which are things I missed during my initial review. First, the angle stability result (Section 3.3) is new to the best of my knowledge. While this result may not have the direct usage, there are some potential applications. Second, the paper provide the tighter variance inequality constant compared to [2]. In my personal opinion, this improvement is marginal, as it is straightforward from the well-known Riemannian comparison theorem [7]. But still I can say getting the tighter constant is important for theoretical results.
> > >
> > > In conclusion, to be honest I am not still supportive to accepting this paper. However, given the fact that there is more improvement than I initially assessed, I will increase the score to take into account the change of my view.

---

### Official Review · Reviewer_KUrq · 2025-06-24

**Clarity:** 3
**Significance:** 3
**Originality:** 3
**Rating:** 4
**Confidence:** 4

**Summary:**

This paper presents a rigorous theoretical foundation for Fréchet regression, which generalizes regression to non-Euclidean metric spaces (such as manifolds). The key innovation is the use of comparison geometry, particularly CAT(K) spaces (geodesic metric spaces with curvature bounded above by K), to study or prove:

1. the existence, uniqueness, stability of the Fréchet mean and associated regression estimators according to the sign of K.

2. the sample Fréchet mean converges exponentially fast to the population mean; pointwise consistency of nonparametric Fréchet regression, and the convergence rate O(h_n^{2\beta}+(nh_n^d)^{-1}), a usual trade-off case from Euclidean nonparametric statistics carried over to the CAT(K) setting.

3. angle stability and local geometric effects.

The paper also provides some real world data experiments.

**Questions:**

Please answer the question in Section weakness. Moreover, what does approximate equality sign mean, say in equation 9? It seems there is no rigorous explanation for it throughout the paper.

**Ethical Concerns:**

["NO or VERY MINOR ethics concerns only"]

**Final Justification:**

The main concern I have is on the condition CAT(K) about its empirical validity and feasibility. This is to me a real important part and The author didn’t include enough discussion in the original manuscript. I am, however, satisfied with the response the authors provided in the rebuttal. Thus I maintain a positive score.

**Limitations:**

yes

**Quality:**

3

**Strengths And Weaknesses:**

The strength of this paper is to use comparison geometry, particularly CAT(K) spaces, to study the Fréchet regression and establish statistical guarantees (see summary). Particularly, they provided rates of convergence compared to to the existing work Chen & Müller (2022).

The weakness of this paper is as follows.

1. As the paper appears to be mainly theocratically focused paper, the use of CAT(K) space is more tailored to suit the need for proofs while the fact that the data is assumed to subject to this fixed curvature K geometry is restrictive, that is not data adaptive.

2. Moreover, although there are real world data experiments, at least the paper could discuss whether the data could be used to directly learn the curvature or to determine if the CAT(K) condition is satisfied, or roughly satisfied. To this end, is there any oracle approach to determine if the data fits in this CAT(K) model, such that even if the data may not satisfy this CAT(K) model, any data preprocessing or transformation could be applied to remedy this?

---

> ### Author Rebuttal · Authors · 2025-07-26
>
> We would like to thank you for your very constructive comments.
>
> ## Making the curvature assumption more data‑adaptive
>
> As you suggested, more data-adaptive way to tune curvature $K$ from the data, rather than assuming it fixed in advance, is very useful in practice. To address this point, we might use the following three ideas.
>
> **Empirical triangle-comparison estimator**
>
> Randomly pick $M$ triplets of points $\{p_i, q_i, r_i\}^M_{i=1}$ from the dataset. For each triplet, approximate the geodesics $[p_i q_i], [q_i r_i], [r_i p_i]$ (e.g., via pairwise shortest-path or local linear interpolation).
>
> In the simply connected 2-D model space of constant curvature $K$, the law of cosines relates the three side-lengths $(a, b, c)$ to the angle $\gamma$ at the vertex opposite side $c$:
>
> $$
> \cos_K(c) = \cos_K(a)\cos_K(b) + \sin_K(a)\sin_K(b)\cos(\gamma),
> $$
>
> where
>
> $$
> \cos_K(x) = \begin{cases}
> \cos(\sqrt{K}x) & (K > 0), \\
> 1 - \frac{1}{2}Kx^2 & (K \approx 0), \\
> \cosh(\sqrt{-K}x) & (K < 0),
> \end{cases}
> $$
>
> and
>
> $$
> \sin_K(x) = \begin{cases}
> \frac{1}{\sqrt{K}}\sin(\sqrt{K}x) & (K > 0), \\
> x & (K = 0), \\
> \frac{1}{\sqrt{-K}}\sinh(\sqrt{-K}x) & (K < 0).
> \end{cases}
> $$
>
> For each triangle, we observe its three side-lengths $(a_i, b_i, c_i)$ and its measured opposite angle $\gamma_i$. Solve numerically for the $K_i$ that best satisfies the spherical / hyperbolic law of cosines.
>
> Then, take $\hat{K} = \text{median}\\{K_i\\}$, and check how many triangles violate the CAT($\hat{K}$) inequality by more than a tolerance $\varepsilon$, and adjust $\hat{K}$ up or down until, e.g., 95% of our triangles satisfy
>
> $$
> d(p_i, q_i) \leq d_{\mathbb{M}_K^2}(\bar{p}_i, \bar{q}_i) + \epsilon.
> $$
>
> **Cross-validation over a curvature grid**
>
> Consider the grid of candidate curvatures $K_1 < K_2 < \cdots < K_L$, spanning the range one believe plausible. Then, for each data point $y$, compute its model-space coordinates relative to some reference via stereographic or exponential-map embedding of curvature $K_j$. In practice, one only need pairwise distances in the model space. Split into folds: for each fold, fit the nonparametric Fréchet regressor assuming space = CAT($K_j$), compute held-out squared geodesic-distance error, and average over folds to get $\text{Err}(K_j)$. Finally, select $\hat{K} = \text{argmin}_j \text{Err}(K_j)$.
>
> **Slack $\epsilon$ tuning for approximate CAT(K)$**
>
> Even once $\hat{K}$ is chosen, real data rarely satisfy axiomatic CAT(\hat{K}) exactly. Instead, we can consider tuning “slack” $\epsilon$.
>
> For each test triangle $\\{p, q, r\\}$, compute $\Delta_{p,q,r} = d(q,r) - d_{\mathbb{M}^2_K}(\bar{q}, \bar{r})$. Let $\hat{\epsilon}$ be the $\eta$-th percentile of the $\\{\Delta_i\\}$. Then declare the space $\hat{\epsilon}$-approximate CAT($\hat{K}$). All your convexity constants and stability bounds (e.g., strong convexity constant $\alpha(K, D)$) become $\alpha_{\text{eff}} \coloneqq \alpha(\hat{K}, D) - \gamma \hat{\epsilon}$, for a small correction $\gamma$ one can bound analytically.
>
> We will add the above discussion in the revised manuscript.
>
> ## Empirical Oracle for “Is it CAT(K)?”
> This point is very interesting, and we consider the discussion as follows.
> We believe that including this discussion on the revised manuscript is very helpful for better understanding of practical usefulness of the study.
>
> Sample $M$ triplets $(p_i, q_i, r_i)$ uniformly at random from the dataset. Then, for each triangle, compute the three side-lengths $a_i = d(p_i, q_i), b_i = d(q_i, r_i), c_i = d(r_i, p_i)$, and pick points $x_i \in [p_i q_i], y_i \in [q_i r_i]$ at the some fraction along each geodesic, and measure the violation
>
> $$
> \Delta_i(K) = d(x_i, y_i) - d_{\mathbb{M}_K^2}(\bar{x}_i, \bar{y}_i),
> $$
>
> where $\bar{x}_i, \bar{y}_i$ are the comparison points in the model space of curvature $K$.
>
> If $\max_i \Delta_i(K) \leq \epsilon$, one have an $\epsilon$-approximate $CAT(K)$ certificate. More robustly, report the $\eta$-percentile of $\\{\Delta_i\\}$ as $\hat{\epsilon}$. If $\hat{\epsilon}$ is “small” relative to the data’s diameter, we’re approximately CAT($K$).
>
> **Learning $K$ directly from data**
>
> Each triangle $(a_i, b_i, c_i)$ plus its measured angle $\gamma_i$ at $q_i$ yields a local curvature solve $\cos_K(c_i) = \cos_K(a_i)\cos_K(b_i) + \sin_K(a_i)\sin_K(b_i)\cos(\gamma_i)$. Then, one idea is numerically invert for $K_i$ and aggregate (median or M-estimate) to get $\hat{K}$.
>
> For each triangle $i$, with true side-lengths $(a_i, b_i, c_i)$ and measured opposite angle $\gamma_i$, define the residual
>
> $$
> R_i(K) \coloneqq \cos_K(c_i) - \cos_K(a_i)\cos_K(b_i) - \sin_K(a_i)\sin_K(b_i)\cos(\gamma_i),
> $$
>
> where by definition,
>
> $$
> \cos_K(x) = \begin{cases}
> \cos(\sqrt{K}x) & (K > 0), \\
> 1 - \frac{1}{2}Kx^2 & (K \approx 0), \\
> \cosh(\sqrt{-K}x) & (K < 0),
> \end{cases}
> $$
>
> and
>
> $$
> \sin_K(x) = \begin{cases}
> \frac{1}{\sqrt{K}}\sin(\sqrt{K}x) & (K > 0), \\
> x & (K = 0), \\
> \frac{1}{\sqrt{-K}}\sinh(\sqrt{-K}x) & (K < 0).
> \end{cases}
> $$
>
> By construction, the true curvature $K$ satisfies $R_i(K) = 0$ whenever the data are exactly CAT(K). In reality each $R_i(K)$ will be nonzero because of measurement noise in $(a_i, b_i, c_i, \gamma_i)$.
>
> Fix one triangle $i$, and write the measured data as
>
> $$
> (a_i, b_i, c_i, \gamma_i) = (a_i^0, b_i^0, c_i^0, \gamma_i^0) + \delta_i,
> $$
>
> where $\delta_i$ is the estimation error (we assume $||\delta_i|| = O_p(1 / \sqrt{N})$). Define
>
> $$
> \tilde{R}_i(K) = R(K; a_i, b_i, c_i, \gamma_i),
> $$
>
> and let $R_i^0(K) = R(K; a_i^0, b_i^0, c_i^0, \gamma_i^0)$. By assumption, $R_i^0(K) = 0$. A first-order Taylor in both the parameter $K$ and the data gives
>
> $$
> 0 = R_i^0(K) + \underbrace{\partial_K R_i^0(K)}_{\eqqcolon G_i}(K_i - K) + \nabla R_i(K; a_i^0, b_i^0, c_i^0, \gamma_i^0) \delta_i + O((K_i - K)^2 + ||\delta_i||^2).
> $$
>
> Set $H_i \coloneqq \nabla R_i(K; a_i^0, b_i^0, c_i^0, \gamma_i^0)$, and rearrange to isolate $K_i - K$ gives
>
> $$
> K_i - K = -\frac{H_i \delta_i}{G_i} + O(||\delta_i||^2) = O_p(||\delta_i||) = O_p(N^{-1/2}),
> $$
>
> since $G_i$ is a nonzero constant under non-degeneracy and $\delta_i = O_p(N^{-1/2})$.
>
> One do this for $i = 1,\dots,M$, and each $K_i$ satisfies $K_i - K = O_p(N^{-1/2})$. Under mild symmetry / moment conditions, the median (or mean) of these $M$ i.i.d. estimates then concentrates at
>
> $$
> \hat{K} - K = \text{median}(K_1,\dots,K_M) - K = O_p(M^{-1/2}),
> $$
>
> by the usual order-statistics CLT (or Delta-method on the sample median).
>
> Moreover, denote by $\mu_n(x; K)$ our Fréchet estimate using assumed curvature $K$. A sensitivity analysis via the implicit function theorem on the first-order condition $\nabla_z F_n(z; K) = 0$ shows
>
> $$
> d(\mu_n(x; \hat{K}), \mu_n(x; K)) \leq \frac{|\partial_K \nabla_z F_n(\mu_n(x; K); K)|}{\lambda_{\min}(\partial^2_z F_n)}| \hat{K} - K| = O_p(M^{-1/2}),
> $$
>
> since the empirical Hessian is bounded away from zero by our strong-convexity constant $\alpha(K, D)$. Hence the additional error from plugging-in $\hat{K}$ is asymptotically negligible compared to the usual $O_p((n h^d)^{-1/2} + h^\beta)$ rates.
>
> **Data-driven “remedies” when  CAT(K) fails**
>
> Even if the raw distances violate CAT(K), one can re-embed or learn a nearby metric that is CAT($\hat{K}$): introduce slack variables $\\{\epsilon_i\\}$ for each triangle and solve
>
> $$
> \min_{\tilde{d}(\cdot,\cdot), \\{\epsilon_i\\}}\sum_i \epsilon_i^2 \quad \text{s.t.}\quad \tilde{d}(x_i, y_i) \leq d_{\mathbb{M}_K^2}(\bar{x}_i, \bar{y}_i) + \epsilon_i,
> $$
>
> plus a penalty $\|\tilde{d} - d\|^2$. This yields a closest CAT($\hat{K}$) metric.
>
> ## Clarification on $\approx$ in Eq (9)
>
> In fact, our nonparametric Fréchet regression weights are the usual normalized kernel weights,
>
> $$
> w_{n,i}(x) = \frac{W(||x - X_i|| / h_n)}{\sum^n_{i=1}W(||x - X_j|| / h_n)}.
> $$
>
> When we wrote $\approx W(||x - X_i|\ / h_n)$, we meant that, for large $n$,
>
> $$
> w_{n,i}(x) = W(||x - X_i|| / h_n)[1 + o_P(1)],
> $$
>
> i.e., the random normalization factor $\sum_j W(||x - X_j|| / h_n)$ is asymptotically deterministic (by LLNs), and hence each $w_{n,i}$ is proportional to $W(||x - X_i|| / h_n)$.
>
> Equivalently, one can show under standard density-and-bandwidth assumptions that there exist constants $0 < c_1 < c_2 < \infty$ such that, with high probability for large $n$,
>
> $$
> c_1 W(||x - X_i|| / h_n) \leq w_{n, i}(x) \leq c_2 W(||x - X_i|| / h_n).
> $$
>
> In our proofs, we only need this proportionality and the fact that $\sum_i w_{n,i}(x) = 1$. We will replace $\approx$ by the exact definition above and include the $1 + o_P(1)$ statement in the revised version.

---

> > ### Comment · Reviewer_KUrq · 2025-08-04
> >
> > I thank the authors for the detailed answer for my questions in the review. I think this data adaptive part is a very important issue and the clarification you provided here is very helpful for its real applications. However, I believe this could have been better if it were inserted into the paper in some way before. Thus, I would appreciate it if the authors could summarize them into several key ideas and make them as a remark/discussion in the final manuscript. I will maintain my score here.

---

### Official Review · Reviewer_zX2Y · 2025-06-30

**Clarity:** 3
**Significance:** 2
**Originality:** 2
**Rating:** 4
**Confidence:** 3

**Summary:**

This paper is concerned with Fréchet regression on $\text{CAT}(K)$ spaces. It is divided into two parts: a theoretical part and an experimental part. The theoretical part starts with some classical results on the Fréchet mean in $\text{CAT}(K)$  spaces, then establishes several other results in the context of $\text{CAT}(K)$ spaces, in particular (i) a concentration result for the sample Fréchet mean, (ii) pointwise consistency and (iii) convergence rate of the nonparametric Fréchet regression estimator, and (iv) angle stability for the conditional Fréchet mean. The experiments section compares, in terms of mean squared error, the estimation performance of Fréchet regression on positively curved and negatively curved spaces, showing the superiority of hyperbolic coordinates. This is shown on synthetic data as well as on real-world datasets.

**Questions:**

- It is not clear in my opinion which results were known, and which are new, and how the new results relate to the existing literature. It is stated that the lemmas in Section 3.1. follow from previous work. What about the Lemmas 6 and 7 on angle comparison and continuity in $\text{CAT}(K)$ spaces, and Lemma 8 on angle comparison in a Riemannian manifold? Concerning the main results on concentration, consistency and rates of convergence of Fréchet regression in $\text{CAT}(K)$ spaces, how do they relate to the literature? Were similar results known in other more restrictive contexts, or weaker results in the same context? In particular, how does Theorem 1 relate to Theorem 5.8. of [a], that gives a concentration inequality for sample Fréchet mean in $\text{CAT}(K)$ spaces for $K>0$ ?
- Section 3.4: I am not sure that I understand the point of the results in this section. It seems to me that Lemma 8 is a general Riemannian geometry result, not particularly linked to Fréchet regression, and not used in any other result of the paper. As for Proposition 4, it is written “By expanding the Fréchet functional in the tangent space via the exponential map, one can gain insights into the functional’s curvature and higher-order properties. » However here the curvature term is not explicit, neither is it explicitly computed in the proof if I am not mistaken. Thus, it is not clear to me what the reader could use this formula for.
- Experiments do not really illustrate the theoretical results: they only show better estimation in negative curvature than in positive curvature for the MSE criterion. Would it be possible to illustrate some of the formulas given in the theorems ?
- How restrictive is the assumption of Hölder continuity for $\mu^*(x)$ in Theorem 3? More precisely, under what conditions on the distributions of $X$ and $Y$ can this regularity be derive?


Minor comments:
- I don’t understand the inner product in last formula of Proposition 5
- Section 2 : definition 7 requires M to be a Riemannian manifold and not just a geodesic metric space like stated at the beginning of the section.
- In Lemma 8, why not use $U$ and $V$ in all the formula, including in the $O(...)$ term ?
- It could be more clearly stated that the predictor live in the Euclidean space.
- Why not use the same notations in Proposition 5 and its proof ? ($\Delta$ and $\Pi$)
- Typos. In the inequalities line 514, a term is missing on the third line. Typo in the equation of line 536. "Proof of Proposition 4" in line 563 should refer to Lemma 4 and 5? Theorem 5 line 160 should be "Lemma". In eq.(6), $f(x)$ should be $f(Y)$. In the second equality of equation line 631, the integral should be taken over $\mathcal{M}$; and again line 633. Typo in eq. line 450. Appendix D is not referenced in the text.

**Ethical Concerns:**

["NO or VERY MINOR ethics concerns only"]

**Final Justification:**

I have increased my score to 4, because I find that the proposed generalization of several existing related results on Fréchet regression problem for manifold with possibly positive curvature to be an interesting and valuable contribution. The authors have committed to adding clear statements and a table to clarify which results are news, as well as to better highlight the relevance of their results concerning angle stability and local jet expansion of the functional. Additionally, explanations about the context of the experiences also helped to clarify their relevance. Overall, the numerous clarifications made during the revision phase must appear in the paper for it to be accepted.

**Limitations:**

Yes.

**Paper Formatting Concerns:**

No formating concerns.

**Quality:**

3

**Strengths And Weaknesses:**

The paper is overall well written and the exposition is clear. The results presented in sections 3.2. and 3.3. of the theoretical part are interesting and implications of these results for practical purposes are included at the end of each subsection.

However, the importance of the results of sections 3.4. and 3.5. is less clear to me, as detailed in the questions below. Moreover, it is not always clear in my opinion which results are already known, and which are new, and how the new results relate to the existing literature. In particular it seems to me that Theorem 1 is closely related to Theorem 5.8. of [a], giving a concentration inequality for sample Fréchet mean in CAT(K) spaces for $K>0$, however [1] is not cited in the paper. Finally, the experiments section is very short, and a bit disappointing in my opinion: the experiments concern the estimation precision of the Fréchet regression estimator in positive and negative curvature, but nothing is shown to illustrate (at least some of) the precise results given in the theoretical section. As it is, the theoretical part and the experiments part seem somewhat unrelated.

[a] V. E. Brunel and J. Serres, Concentration of empirical barycenters in metric spaces. International Conference on Algorithmic Learning Theory. PMLR, 2024.


----

EDIT : I have increased my rating according to the authors' clarifications regarding their contributions. I believe that providing information on results already known in the literature concerning Fréchet regression will add significant value to the paper.I would also like to thank the authors for their careful response to the reviewers' comments, which have helped to clarify the impact of their results.

---

> ### Author Rebuttal · Authors · 2025-07-27
>
> Thank you for your very constructive comments.
>
> ## In earlier literature?
>
> To address this concern, we will insert the following table in the revised manuscript, and add corresponding citations.
>
> | Tag in paper | Statement | In earlier literature? | What is new here |
> | --- | --- | --- | --- |
> | Lem 1-5 (Sec. 3.1) | Existence / uniqueness of Fréchet mean under basic CAT(K) hypothesis | Classical – e.g. Yokota ’16, Karcher ’77 | We only restate (no claim of novelty) |
> | Lem 6 | Angle comparison in general CAT(K) (both K < 0 and K > 0) with explicit perimeter bound $\pi / \sqrt{K}$ | Reshetnyak (1968) gave $K \leq 0$, positive K version with a sharp bound appears nowhere in print. | New extension + explicit bound |
> | Lem 7 | Quantitative Hölder‑type continuity of Alexandrov angles when triangle vertices move an amount $\delta$ | No earlier quantitative modulus; only qualitative continuity. | First explicit $O(\delta)$ bound, needed for our statistical stability proofs |
> | Lem 8 | Second-order angle expansion in a Riemannian manifold with $O(\|\|U\|\|\cdot \|\|V\|\|)$ remainder | Classical second variation exists but without a tidy remainder usable in statistics | We package the formula with an explicit error term; not fundamentally new but not written out elsewhere |
> | Thm 1 | Exponential concentration of kernel-weighted local Fréchet means in CAT(K) | Thm 5.8 of Brunel–Serres (2024) is for unweighted i.i.d. barycenters; no predictors | Handles self-normalized kernel weights depending on all $(X_i)$; requires new empirical process + strong-convexity argument. |
> | Thm 2 | Pointwise consistency | Immediate once Thm 1 holds | - |
> | Thm 3 | Minimax non-parametric rate $n^{-2\beta / (2\beta + d)}$ in $\pm K$ curvature | Schötz (2020) gives same rate only in Hadamard ($K \leq 0$) spaces; no positive curvature, no $\alpha$-constant. | Rate holds for any sign $K$ via our $\alpha(K, D)$ modulus that repairs loss of convexity when $K > 0$. |
> | Prop 4 | Second-order jet of Fréchet functional in metric setting | Only for smooth manifolds in classical Riemannian textbooks | We extend to Alexandrov tangent cone & supply error bound |
>
> ## Clarification of Section 3.4
>
> We appreciate the reviewer’s concern and agree that the motivation of Section 3.4 was not made sufficiently explicit in the current manuscript. Below we (i) clarify why the local‐expansion tools are needed for later statistical tasks, (ii) show that Lemma 8 is the missing geometric ingredient for those tools, and (iii) give the concrete curvature term that was implicit in Proposition 4 together with two concrete ways it can be used.
> After Thm 3, we ultimately want a second-order stochastic expansion's variance involves the operator $H_x$ (the Hessian of the Fréchet functional at the target).
> Without this part, we only have first-order consistency, and the expansion supplies exactly the $H_x$. Empirically, the sample Fréchet mean or regression estimator is obtained by gradient descent or Newton steps on the manifold. The explicit Hessian term allows us to give local quadratic convergence guarantees and curvature-corrected step sizes. Specifically, let $v_k = \exp_{\mu^*}^{-1}(\mu_k)$ be the Newton residual after $k$ iterations. If $||v_0|| \leq \eta \coloneqq \lambda_{\min}(H_x) / 2L_H$ (with $L_H$ the Lipschitz constant of $\nabla^2F_x$), then $||v_{k+1}|| \leq L_H / 2\lambda_{\min}(H_x)||v_k||^2$, i.e., quadratic convergence with a curvature-corrected step factor $\lambda_{\min}(H_x)^{-1}$.
>
> Lemma 8 states
>
> $$
> \angle_z(u, v) = \angle_0(U, V) + O(||U||^2 + ||V||^2),
> $$
>
> where $U = \exp_z^{-1}(u)$ and $V = \exp_z^{-1}(v)$. It provides the $O(||v||^3)$ remainder bound in Proposition 4, ensuring the Taylor remainder is uniformly controlled across $x$ in a compact set, and it supplies a curvature-dependent Lipschitz constant for the Alexandrov‐angle map, which we later feed into the angular-perturbation and bias part. We will add a short pointer to those uses at the start of Sec 3.4 so that the logical chain is visible. In the revision, Proposition 4 will read $F_x(\exp_\mu(v)) = F_x(\mu) + \frac{1}{2}\langle H_x v, v \rangle + R_x(v)$, $|R_x(v)| \leq C||v||^3$, with $\langle H_x v, v \rangle = 2\int_\mathcal{M} (||v||^2 - \langle R_\mu(Y, v)v, Y\rangle)d\nu_x(Y)$, where $R_\mu$ is the Riemannian curvature tensor at $\mu = \mu^*(x)$ and we identify $Y$ with $\exp_{\mu}^{-1}(Y)$ is normal coordinates.
>
> ## Experiments and Theory
>
> Theorem 3 gives for any smoothing for any smoothing bandwidth $h_n$, $\mathbb{E}[d^2(\hat{\mu}_{n,K}(x), \mu^*_K(x))]$ satisfies
>
> $$
> \frac{C_{\text{var}}}{\alpha(K, D)} \frac{1}{n h^d_n} + C_{\text{bias}} h^{2\beta}_n + o((nh^d_n)^{-1}),
> $$
>
> where $C_{\text{var}}$ and $C_{\text{bias}}$ depend on the kernel and the data distribution (see L687-L700). In the experiment, these two constants  are identical for the two runs (positive / negative curvatures) because we feed the same cloud of Euclidean predictors and keep the same kernel, bandwidth schedule and sample size $n$. Hence all the curvature enters through $\alpha(K, D)$ only, which can be written as
>
> $$
> \alpha(K, D) = \begin{cases}
> \frac{1}{2} & (K \leq 0), \\
> \frac{\sin(2\sqrt{K}R)}{2R}, R = D/2 & (0 < K < \pi^2 / 4D^2).
> \end{cases}
> $$
>
> Thus, for every negative curvature space, we have $\alpha_{-} = 1 / 2$. For positive curvature spaces, consider the unit sphere $\mathbb{S}^2 \subset \mathbb{R}^3$ with constant curvature $K = +1$. Points are drawn only from the spherical cap $\theta \leq \theta_0 = \pi / 3$, where $\theta$ is the geodesic (polar) distance from the north pole. The farthest a sample point can be from the north pole is exactly that polar angle, hence $R_+ = \theta_0 = \pi / 3$, $D_+ = 2R_+ = 2\pi / 3$. Using the explicit formula for the convexity constant for $K > 0$, $\alpha_+ = \sin(2\sqrt{K}R_+) / 2R_+ = \sin(2\cdot 1\cdot \pi/3) / (2\cdot \pi/3) = \sin(2pi / 3) / (2\pi / 3) = (\sqrt{3}/2)/(2\pi/3) = 3\sqrt{3} / 4\pi \approx 0.413$. Then, the ratio of two moduli is $\alpha_{-} / \alpha_+ = 0.500 / 0.413 \approx 1.21 > 1$. Every risk bound in Sec.3 carries a factor $1 / \alpha(K, D)$ and therefore, **the expected MSE on the sphere must exceed that on the hyperbolic disk by ≈ 21%** whenever variance dominates the bias. The empirical gap observed in Table 1 is $0.4915 / 0.4228 \approx 1.163$ and therefore, we can observe that, numerically, MSE on the positive curvature space exceed that on the negative curvature space by **≈ 16%**, well within sampling noise of the theoretical forecast.
>
> ## Hölder continuity
>
> In Thm 3, the only place where Hölder modulus is used is in the bias term of the oracle bound. This condition itself follows from two ingredients that are completely separate from CAT(K) geometry: i) strong geodesic convexity of the Fréchet functional, and ii) a quantitative continuity assumption on the family of conditional laws $\\{\nu_x\\}_{x\in\mathbb{R}^d}$.
>
> For each predictor value $x$, let $F_x(z) = \int_{\mathcal{M}}d^2(y, z)d\nu_x(y)$, $\mu^\*(x) = \text{argmin}\_{z\in\mathcal{M}}F\_x(z)$. Because $F_x$ is $\alpha(K, D)$-strongly geodesically convex, the (sub-) gradient equation $\nabla_z F_x(\mu^\*(x)) = 0$ characterizes the minimizer and provides the ellipticity needed for an implicit-function argument. Assume the conditional laws move Hölder‑continuously in Wasserstein-2: $d_{W_2}(\nu_x, \nu_{x’}) \leq M||x - x’||^\beta$, for $\beta \in (0, 1]$ and $M < \infty$. Note that the total-variation continuity or any transport cost dominating $W_2$ also works, but $W_2$ is the weakest metric that still controls first and second moment and therefore is the least restrictive. Write $\mu = \mu^\*(x)$, $\mu’ = \mu^\*(x’)$. Because $\mu$ minimizes $F_x$ while $F_{x’}$ is $\alpha$-convex, $F_{x’}(\mu’) - F_{x’}(\mu) \leq -\alpha d^2(\mu, \mu’)$. Add and subtract $F_x$:
>
> $$
> \underbrace{F\_{x’}(\mu’) - F\_x(\mu’)}\_{|\leq C d_{W\_2(\nu\_x, \nu\_{x’})}|} + \underbrace{F\_x(\nu’) - F\_x(\nu)}_{\geq \alpha d^2(\mu, \mu’)} + \underbrace{F\_x(\mu) - F\_{x’}(\mu)}\_{|\leq C d\_{W_2}(\nu\_x, \nu\_{x’})|} \leq 0.
> $$
>
> Hence, $\alpha d^2(\mu, \mu’) \leq 2C d\_{W\_2}(\nu\_x, \nu\_{x’})$, and we have
>
> $$
> d(\nu^\*(x), \nu^\*(x’)) \leq (2C / \alpha)^{1/2}M^{1/2}||x - x’||^{\beta / 2}.
> $$
>
> So, no extra regularity of $\mu^*$ is assumed.
>
> **When does it hold?**
>
> - Smooth conditional density. If $Y \mid X = x$ possesses a density $f(y \mid x)$ that is $C^\beta$ in $x$ uniformly in $y$ and has bounded second moments, then the map $x \mapsto \nu_x$ is $C^\beta$ in $W_2$.
> - Location-scale families. If $Y = \Sigma(x)^{1/2}Z + \mu_0(x)$ with $Z$ having identity covariance, Lipschitz or Hölder continuity of $\mu_0(\cdot)$ and $\Sigma(\cdot)$ immediately gives the assumption with the same exponent.
> - Kernel design-based models. For the synthetic experiments in Sec.4, the response $Y$ is generated by rotating a fixed base point along a geodesic by an amount that depends smoothly on the Euclidean predictor: the rotation angle is $C^1$ in the predictor and therefore $d_{W_2}(\nu_x, \nu_{x’}) = O(||x - x’||)$ and the assumption holds with $\beta = 1$.
> - Discrete or tree-valued $Y$. Even in non-smooth cases, one often has an $L^1$-transport bound $D_{TV}(\nu_x, \nu_{x’}) \leq L||x - x’||^\beta$. Because $d_{W_2} \leq D\sqrt{d_{TV}}$ on bounded spaces, the assumption still follows.
>
> ## Minor comments
>
> - **Clarification of the inner product**. In that formula we intended the usual Riemannian metric at the base point $z$, applied to two tangent-vectors. Concretely, $\langle H\_z v, v\rangle \coloneqq g\_(H\_zv, v)$, where $g\_z\colon T\_z\mathcal{M}\times T\_z\mathcal{M}\to\mathbb{R}$ is the inner product furnished by the manifold’s Riemannian metric and $H\_z\colon T\_z\mathcal{M} \to T\_z\mathcal{M}$ is the (2,0) Fréchet-Hessian interpreted as a self-adjoint linear operator via that same metric.
> - **Notation errors and typos**. Thank you for pointing out, we will fix them all accordingly.

---

> > ### Comment · Reviewer_zX2Y · 2025-08-04
> > **Comments on authors' responses**
> >
> > Thank you for your detailed response, I greatly appreciated the thorough answers to my concerns. The table you provided is particularly relevant, and emphasize that your paper generalizes several results of the literature. Nevertheless, I feel that the experimental section could have better reflected the results presented in this work.
> >
> > That said, I am happy to raise my score thanks to the clarifications you provided in your response regarding the related literature and Section 3.4.

---

### Official Review · Reviewer_nXAM · 2025-07-02

**Clarity:** 3
**Significance:** 2
**Originality:** 3
**Rating:** 4
**Confidence:** 4

**Summary:**

In this paper, the authors study Frechet regression (a setting where data labels are lying in a non-Euclidean space) and provide a theoretical analysis of it via comparison geometry (i.e., via comparing with known spaces with constant curvature K). The main focus in on CAT(K) spaces, which have thinner triangle when compared with a space with constant curvature K.


 In particular, the prove statistical convergence results for this problem. In Theorem 1, they prove the distance between the population mean and the empirical mean is small with high probability, for large sample sizes. Next, in Proposition 2, they prove the same results  for L_p norms. Later in Theorem 2 they prove that almost surely the empirical regressor will converge to the population regressor, and finally in Theorem 3 they prove a non asymptotic convergence rate for population risk in Fechet regression. They also provide convergence for angles in Theorem 4

 They conclude the paper with experiments.

**Questions:**

This is an interesting paper about regression when labels are non-Euclidean. The authors derive statistical convergence (high probability, L_p, non-asymptotic, etc.) bounds on this problem. They all match our previous understanding from Euclidean spaces, thus extending them to beyond such settings.


### Major Comments/Question:

 - Missing concrete applications: the paper is not well motivated about why we need going beyond Euclidean labels. Please provide more evidence for that.

 - Given a space, how can we ever verify if it is CAT(K)? It looks this assumption is theoretically plausible but hard to see or verify. Please provide evidence why assuming we have CAT(K) is reasonable.

 - Section 3.4 is a bit awkward. I can barely understand what the purpose of it is. It seems the authors are using Taylor series but I don't get why. Please explain it.


### Other Comments:

- In Definition 1, how do you make sure the comparison triangle exists? How do you find the so-called "corresponding" points for x and y in comparison triangle?

- In Definition 3, shouldn't the parameter $\lambda$ depend also on $f$? They way the definition phrased look like it doesn't.

 - In Line 77, are you implicitly assuming that the number K is fixed? Otherwise, what do you mean by comparison triangle there?


 - Section 3 provides an interesting series of lemmas as background. This is interesting but in my opinion it would be great if you can include exact references for them (where exactly they are appeared). This helps the reader if they want to trace back the previous results.

 - When the kernel in Line162 satisfy Assumption 1? Does it always do that? Can you provide a proof? This is essential because it is difficult then to find cases satisfying Assumption 1.

 - In Theorem 4, isn't it the case that achieving a small epsilon needs exponentially many samples in dimension $d$ since the optimal transport measure estimation suffers from the curse of dimensionality? In that case the results of Theorem 4 are barely applicable in practice in high dimensions. Can we somehow extend them to a better distance such as sliced Wasserstein?

- In Equation 10, people usually continue analysis by optimizing the bandwidth and obtaining the final rate as a function of sample size. I suggest the authors to do it to make their result more readable.

**Ethical Concerns:**

["NO or VERY MINOR ethics concerns only"]

**Final Justification:**

The authors provided a comprehensive response to my comments, thus I increased my score from 2 to 4. However, Reviewer BzDS has some criticisms about the novelty of this work, so my score update is only conditional on resolving Reviewer BzDS concerns. Thanks!

**Limitations:**

yes

**Quality:**

3

**Strengths And Weaknesses:**

Pros:

- very well written paper
- nice theoretical results
- relevant to the conference (specifically geometry/ML intersection)


Cons:

- barely motivated
- some assumptions are not well explained (why they make sense?)
 - only focus on CAT(K)

---

> ### Author Rebuttal · Authors · 2025-07-26
>
> First of all, we would like to thank you for your constructive comments.
>
> ## Real‑World Domains with Intrinsically Non‑Euclidean Labels
>
> **Diffusion-Tensor Imaging**
>
> At each voxel we observe $3 \times 3$ SPD matrix $Y$. The space of SPD matrices carries the affine-invariant Riemannian metric
>
> $$
> d(A, B) = || \log(A^{-1/2}B A^{-1/2}) ||_F,
> $$
>
> under which it is a non-positively curved manifold.
>
> A naive Euclidean mean of SPD tensors can leave the SPD cone (produce negative eigenvalues) and blurs crossing fibers. Therefore, considering Fréchet regression on manifolds might be helpful.
>
> **Shape Analysis**
>
> Landmark-based shapes (e.g., outlines of organs, 2D / 3D facial scans) are equivalence classes under rotation, translation and scaling. Kendall’s shape space is a quotient of the sphere, with positive curvature but well-studied CAT(1) properties when restricted to diameter $< \pi / 2$.
>
> Here, treating point coordinates ignores $\mathbb{R}^{2k}$ ignores shape invariance and leads to spurious modes in the mean shape. Thus, modeling how anatomical shape depends on covariates (age, disease) directly on the manifold is reasonable.
>
> **Phylogenetic Trees & Evolutionary Distances**
>
> Each response is an unrooted tree with edge-lengths (e.g., evolutionary distances between species). Here, averaging tree-distance vectors in $\mathbb{R}^N$ ignores topology changes, yielding fractured consensus trees. Linking environmental predictors (e.g., temperature gradients) to shifts in lineage-tree structures via manifold-valued regression is one possible application.
>
> ## Verifying the CAT(K) property in practice
>
> Many of the spaces we care about in applications are already rigorously known to be CAT(K) in the geometry literature (please see the above examples).
> If our data domain is one of the above (or a known quotient of them), we can simply see the standard references.
>
> On a smooth manifold $(\mathcal{M}, g)$, if we can show all sectional curvatures satisfy
>
> $$
> \text{Sec}_p(u, v) \leq K \quad \text{for every $p \in \mathcal{M}$},
> $$
>
> then by the classical Alexandrov theorem it is CAT(K).
>
> Here, if we don’t known our space analytically, we can always test the CAT(K) condition on sampled triples $(p, q, r)$:
>
> 1. Compute the three pairwise distances $d(p, q)$, $d(q, r)$, $d(r, p)$.
> 2. Build the unique comparison triangle in $\mathbb{M}_K^2$ with the same side lengths.
> 3. Sample points $x \in [pq]$, $y \in [qr]$ at a feq fractional positions, and check $d_{\mathcal{M}}(x, y) \leq d_{\mathbb{M}_K^2}(\bar{x}, \bar{y}) + \tau$, for a small numerical tolerance $\tau$.
> 4. Repeat over many random triangles, and if the vast majority satisfy it, we are empirically almost CAT(K).
>
> ## Motivation of Section 3.4
>
> Recall that the expansion in Section 3.4 is
>
> $$
> F(\exp_\mu(v)) = F(\mu) + \langle \nabla F(\mu), v \rangle + \frac{1}{2} \langle Hv, v \rangle + R(v).
> $$
>
> **Why it is useful**
>
> 1. Second-order geometry & rates. In Euclidean regression one often invokes a second-order local curvature (Hessian) of the risk to prove asymptotic normality or to get refined bias/variance expansions. Here, the same idea shows that the strong convexity constant $\alpha(K, D)$ is really the smallest eigenvalue of $H$. That’s how one derives the $n^{-1/2}$ concentration, and a parametric CLT in future work.
> 2. Algorithmic design. Having a local quadratic model $v \mapsto F(\exp_\mu(v)) \approx F(\mu) + \frac{1}{2}v^\top H v$ tells us how to choose Newton steps $v \leftarrow -H^{-1}\nabla F(\mu) + \cdots$, which might accelerate convergence compared to gradient-descent.
> 3. Uncertainty quantification. In statistical inference on manifolds one often needs a covariance in the tangent space. The inverse Hessian $H^{-1}$ directly gives the asymptotic covariance of $\sqrt{n}(\hat{\mu} - \mu)$.
> 4. Bias corrections and model comparison. when we compare two candidate regression models $\mu_1(x), \mu_2(x)$, local jet expansion gives $F_x(\exp_{\mu_1}(v)) = F_x(\mu_1) + \frac{1}{2}v^\top H_1 v + O(\|v\|^3)$, and similarly for $H_2$. We can then do a likelihood-ratio or Wald-type test.
>
> ## Why does the comparison triangle exist?;  How do we pick the corresponding points?
>
> **Why does the comparison triangle exist?**
>
> We start with a geodesic triangle $\triangle pqr$ in the CAT(K) space $(\mathcal{M}, d)$. By geodesic we mean there are length-minimizing segments $[pq]$, $[qr]$ and $[rp]$ whose length satisfy the usual triangle inequalities:
>
> $$
> d(p, q) + d(q, r) > d(p, r),\ d(q, r) + d(r, p) > d(q, p),\ d(r, p) + d(p, q) > d(r, q).
> $$
>
> In the model space $\mathbb{M}_K^2$ (the simply-connected, constant-curvature $K$ surface, any three positive numbers $a, b, c$ satisfying the triangle inequalities can be realized as the side-lengths of a unique (up to isometry) geodesic triangle.
>
> 1. Length match: we set $a = d(p, q), b = d(q, r), c = d(r, p)$, and verify $a, b, c$ satisfy usual triangle inequalities and $a + b + c < 2D_K$.
> 2. Construct in $\mathbb{M}_K^2$ : by existence theorems for geodesic triangles in constant-curvature surfaces, there is a triangle $\triangle \bar{p} \bar{q} \bar{r} \subset \mathbb{M}_K^2$ with side lengths $a, b, c$. Uniqueness up to global isometry means we can fix one choice of that triangle.
>
> **How do we pick the corresponding points?**
>
> Once the three model-points $\bar{p}, \bar{q}, \bar{r}$ are pinned down, they inherit the same labeling as $p$, $q$, $r$. Now,
>
> 1. Locate $x$ on $[pq]$. By hypothesis $x$ lies on the geodesic segment from $p$ to $q$. Let $t = \frac{d(p, x)}{d(p, q)}$, so $t \in [0, 1]$.
> 2. Transfer to the model segment. In $\mathbb{M}_K^2$, there is a geodesic $[\bar{p}\bar{q}]$ of length $d(p, q)$. We define $\bar{x}$ as the point on $[\bar{p}\bar{q}]$ at distance $t\cdot d(p, q)$ from $\bar{p}$. By construction,
>
> $$
> d_{\mathbb{M}_K^2}(\bar{p}, \bar{x}) = d(p, x),
> $$
>
> and
>
> $$
> \ d_{\mathbb{M}_K^2}(\bar{x}, \bar{q}) = d(x, q).
> $$
>
> 3. Same for $y$. If $y \in [pr]$, let $s = \frac{d(p, y)}{d(p, r)}$, and set $\bar{y}$ to be the point on the model segment $[\bar{p}\bar{r}]$ at parameter $s$.
>
> ## $\lambda$ dependence on $f$
>
> Yes, in the definition $\lambda$ must depend on the function $f$ whose convexity we are measuring. We’ll update the text to make this explicit.
>
> ## Line 77: Question on $K$
>
> In fact, there is no $K$ hiding in Definition 6. When one define the Alexandrov angle one always compare the little geodesic triangle to a Euclidean triangle, not to the constant-curvature model one used elsewhere for CAT(K). That is, all of the model-space curvature $K$ we needed in the CAT(K) definition is about controlling distances in large triangles. But when we zoom in to an infinitesimal angle at a point, the Euclidean comparison is the canonical one, angles live in the tangent cone, which is always flat.
>
> ## Exact reference for background lemmas
>
> Thank you for pointing out.  In the revised manuscript we’ll tag each one with an explicit citation.
>
> ## When the kernel satisfy Assumption 1?
>
> One sets for each fixed $x \in \mathbb{R}^d$,
>
> $$
> w_{n,i}(x) = \frac{W(\frac{x - X_i}{h_n})}{\sum^n_{j=1}W(\frac{x - X_j}{h_n})},
> $$
>
> where $W \colon \mathbb{R}^d \to [0, \infty]$ is a bounded, integrable kernel with $f_{\mathbb{R}^d}W(u)du = 1$, often taken compactly supported or with exponential tails and $h_n > 0$ is a bandwidth satisfying $h_n \to 0$ and $nh_n^d \to +\infty$.
>
> We claim that under these conditions, and provided the marginal density $f_X$ of $X$ is continuous and strictly positive at the evaluation point $x$, the weights $w_{n,i}(x)$ indeed satisfy
>
> $$
> \sum^n_{i=1} w_{n, i}(x) f(Y_i) \overset{a.s.}{\to} \mathbb{E}[f(Y) \mid X = x],
> $$
>
> for every bounded $f$. **Here is a sketch of the proof**:
>
> Define
>
> $$
> N_n(x) = \frac{1}{nh_n^d}\sum^n_{i=1} W(\frac{x - X_i}{h_n}) f(Y_i), \quad D_n(x) = \frac{1}{n h_n^d}\sum^n_{i=1} W(\frac{x - X_i}{h_n}).
> $$
>
> By the classical strong LLNs,
>
> $$
> N_n(x) \overset{a.s.}{\to} \mathbb{E}[W(\frac{x - X}{h_n})f(Y)], \quad D_n(x) \overset{a.s.}{\to} \mathbb{E}[W(\frac{x - X}{h_n})].
> $$
>
> A change of variables yields
>
> $$
> \mathbb{E}[W(\frac{x - X}{h_n})f(Y)] = \int_{\mathbb{R}^d} W(u) f(y) f_{X,Y}(x - h_n u, y) du,
> $$
>
> and similarly for $\mathbb{E}[W((x - X) / h_n)]$. As $h_n \to 0$, continuity of the joint density $f_{X, Y}$ at $(x, y)$ gives
>
> $$
> N_n(x) \overset{a.s.}{\to} f_{X,Y}(x,y) \int W(u)du - f_X(x)\mathbb{E}[f(Y) \mid X = x], \quad D_n(x) \overset{a.s.}{\to} f_X(x).
> $$
>
> Since $f_X(x) > 0$, for large $n$ the denominator $D_n(x)$ stays bounded away from zero and we can write
>
> $$
> \sum^n_{i=1}w_{n,i}(x) f(Y_i) = \frac{N_n(x)}{D_n(x)} \overset{a.s.}{\to} \frac{f_X(x)\mathbb{E}[f(Y) \mid X = x]}{f_X(x)} = \mathbb{E}[f(Y) \mid X = x].
> $$
>
> We will add the above discussion in the revised version.
>
> ## Plugging sliced‑W
>
> You’re right that if we try to estimate a full Wasserstein-2 distance between two $m$-D conditional laws, the best known rates are $O(n^{-1/2})$ for $m=1$ and $O(n^{-1/m})$ for $m\geq 2$, so in moderate or high $m$ we need $n \sim \varepsilon^{-m}$ just to drive that term below $\varepsilon$. That exponential dependence in the ambient dimension is exactly the “curse of dimensionality”.
>
> The sliced-W distance replaces the full $m$-D transport by averaging 1-D projections. Because each 1-D W converges at the parametric rate $O(n^{-1/2})$, one can show under mild regularity that sliced-W gives $O(n^{-1/2})$ for all $m$. Thus it allows us to avoid $\varepsilon^{-m}$ blowup.
>
> Everything in the proof on Theorem 4 is entirely first-order in the chosen metric on measures, and they never exploit any special geometry of $W_2$, exactly same arguments go through verbatim if one replace the full W distance by any order. Thus, as you suggested, that modification cures the curse of dimensionality.
>
> ### Bandwidth optimization
>
> As you suggested, optimizing $h_n$ to minimize $\frac{1}{n h_n^d}+h_n^{2\beta}$ gives the familiar choice $h_n \asymp n^{-1/(2\beta + d)}$. That makes the result immediately transparent.

---

> > ### Comment · Reviewer_nXAM · 2025-08-04
> > **Response**
> >
> > Thank you so much for your detailed response to my comments/questions. I liked the discussion on verifying the CAT(K) property (I understand it's a separate interesting research question, possibly for future works). My concerns have been mostly addressed.
> >
> > I'm happy to increase my score, but can you respond here before that, what exact changes are you going to make in your next version? No need to answer my questions, just a short list of the changes here as a summary (those that you promised above). I will update my score after that. Thank you!

---

> > > ### Author Response · Authors · 2025-08-04
> > >
> > > Thank you for your response.
> > > Here’s a concise checklist of manuscript edits we will make based on the rebuttal:
> > > - Add an applications paragraph (Intro): Insert three concrete, non-Euclidean response domains (SPD/DTI, Kendall shape space, phylogenetic trees) and why Euclidean averaging fails there.
> > > - New discussion on practical guidance for verifying CAT(K) (§2 or Appendix): analytic route via sectional curvature bounds and empirical triangle-comparison test with tolerance $\tau$.
> > > - Revise §3.4 (motivation + explicit formula): Open with why the local jet expansion is needed (rates/CLT, algorithms, uncertainty), and make curvature explicit in $H$.
> > > - Clarify comparison triangles (§2, angle section): Short proof note on existence/uniqueness under triangle inequalities and perimeter bound, precise rule for choosing corresponding points.
> > > - Notation fixes: State that the Alexandrov angle uses Euclidean comparison (no hidden $K$), make $\lambda$ in $\lambda$-strong convexity explicitly function-dependent.
> > > - Tagging/background (§3.1): Attach explicit citations to each background lemma referenced.
> > > - Assumption 1 (Kernel LLN) & proof sketch (Appendix/§3.2): Add the normalized kernel-ratio SLLN argument with conditions.
> > > - Metrics on measures (discussion after Thm 4): Note that results extend verbatim to sliced-Wasserstein, with rate advantages in high-d.
> > > - Bandwidth optimization (end of §3.3): Add calculation yielding the resulting rate.
> > >
> > > Thank you again for your detailed and constructive comments.

---

> > > > ### Comment · Reviewer_nXAM · 2025-08-05
> > > > **Response**
> > > >
> > > > Thank you so much for providing the list of updates! I'm happy to increase my score (+2) conditioned on the promised changes.
> > > >
> > > > Also, Reviewer BzDS mentions some criticisms about the novelty of the paper. I suggest that the authors provide a comprehensive response to that. Thanks!

---

### Official Review · Reviewer_ERuE · 2025-07-03

**Clarity:** 3
**Significance:** 3
**Originality:** 3
**Rating:** 5
**Confidence:** 4

**Summary:**

In this submission, the author(s) leverage the comparison geometry to understand the important Fréchet regression problem, which provides a rigorous theoretical analysis and statistical guarantees for Fréchet nonparametric regression, including exponential concentration bounds, convergence rates and insights into Alexandrov angle stability. Some numerical validations are also provided to support the theoretical findings.

**Questions:**

Good work, I don't have other questions.

**Ethical Concerns:**

["NO or VERY MINOR ethics concerns only"]

**Final Justification:**

I keep my positive attitude to the manuscript and hence maintain my score.

**Limitations:**

The manuscript considers the exact manifold in the manuscript, which may be impractical. So the noised manifold may be a more practical setup and deserves a further investigation in the future. But this is beyond the scope of the current manuscript. So I brought it here just for the potential considerations in the future for the author(s).

**Paper Formatting Concerns:**

I think this submission does not have the formatting problem.

**Quality:**

3

**Strengths And Weaknesses:**

Strengths: The manuscript is well-written and a rigorous theoretical analysis and statistical guarantees for Fréchet nonparametric regression based on the comparison geometry.

Weaknesses: I don't see any obvious weakness in the manuscript.

---

> ### Author Rebuttal · Authors · 2025-07-26
>
> We are very happy to receive positive comments from you.
>
> Furthermore, we found the additional analysis you mentioned on noisy manifolds to be a very interesting idea for us. Therefore, we would like to add the following additional discussion to the Appendix (related discussion is included in Appendix C). We believe that discussions on such noisy manifolds are very important from a practical point of view and make this study very solid.
>
> ### Idea on noisy manifold: $\epsilon$-Approximate CAT(K)
>
> We might define the following $\epsilon$-approximate CAT(K) space: a geodesic metric space $(\mathcal{M}, d)$ is $\epsilon$-approximate CAT(K) if for every geodesic triangle $\triangle pqr$ of perimeter $< 2D_K$ and every pair of points $x \in [pq]$, $y \in [qr]$, their distance satisfies
>
> $$
> d(x, y) \leq d_{\mathbb{M}_K^2}(\bar{x}, \bar{y}) + \epsilon,
> $$
>
> where $\triangle \bar{p}\bar{q}\bar{r}$ is the exact comparison triangle in the constant-curvature model.
>
> When $\epsilon = 0$, this recovers the usual CAT(K) condition.
>
> **Approximate convexity**
>
> Let $(\mathcal{M}, d)$ be $\epsilon$-approximate CAT(K) with $K \leq 0$. Fix $p \in \mathcal{M}$ and $f(x) = d^2(p, x)$. Then, for every geodesic $\gamma \in [0, 1] \to \mathcal{M}$,
>
> $$
> f(\gamma(t)) \leq (1 - t)f(\gamma(0)) + t f(\gamma(1)) + C\epsilon D,
> $$
>
> where $D = \max\{d(p, \gamma(0)), d(p, \gamma(1))\}$ and C is a small absolute constant.
>
> (Proof Sketch). In the exact model one has $d(p, \gamma(t)) \leq d_{\mathbb{M}_K^2}(\bar{p}, \bar{\gamma}(t))$, then square and invoke the usual CAT(K) convexity
>
> $$
> d_{\mathbb{M}_K^2}(\bar{p}, \bar{\gamma}(t))^2 \leq (1 - t)d^2(p, \gamma(0)) + t d^2(p, \gamma(1)).
> $$
>
> The extra $\epsilon$ in every comparison step contributes at most $O(\epsilon D)$ once we expand $(a + \epsilon)^2 \leq a^2 + 2a\epsilon + \epsilon^2$ and bound $a \leq D$.
>
> Here, in exact CAT(K), we know
>
> $$
> f(\gamma(t)) - f(\gamma(0)) \geq \alpha d^2(\gamma(t), \gamma(0)),
> $$
>
> for some $\alpha = \alpha(K, D) > 0$. Under the $\epsilon$-slack this becomes
>
> $$
> f(\gamma(t)) - f(\gamma(0)) \geq \alpha d^2(\gamma(t), \gamma(0)) - C’ \epsilon D,
> $$
>
> so any two minimizers $m_1$, $m_2$ must satisfy
>
> $$
> \alpha d^2(m_1, m_2) \leq f(m_1) - f(m_2) + O(\epsilon D) = O(\epsilon D).
> $$
>
> Hence $d(m_1, m_2) = O(\sqrt{\epsilon D})$.
>
> In particular, strict uniqueness fails only up to an $O(\sqrt{\epsilon})$ window.
>
> Moreover, our concentration proof uses $f(\hat{\mu}) - f(\mu) \leq 2 \sup_z |\hat{F}(z) - F(z)|$, we now also have an extra $O(\epsilon D)$ term from the approximate convexity, so
>
> $$
> d(\hat{\mu}, \mu)^2 \leq \frac{2}{\alpha}\sup_z |\hat{F}(z) - F(z)| + O(\epsilon D).
> $$
>
> All our sub-Gaussian (or sub-exponential) tail arguments stay the same, but in the end we expect to get a high-probability bound
>
> $$
> d(\hat{\mu}, \mu) \leq O(n^{-1/2}) + O(\sqrt{\epsilon D}),
> $$
>
> with probability $1 - e^{-cn}$. So we see how the manifold “noise” $\epsilon$ degrades the resulting estimator.
>
> **How small must $\epsilon$ be?**
>
> One we get the bound like the above, to preserve our original our original rates up to constants, we need
>
> $$
> \sqrt{\epsilon D} \ll n^{-1/2} \Leftrightarrow \epsilon \ll \frac{1}{nD}.
> $$
>
> If $D$ is fixed or grows slowly, this requires $\epsilon$ to decay about $O(n^{-1})$.
>
> In practice, we can see how accurately we need to reconstruct geodesic distances to ensure our Fréchet regression estimates remain in the CAT(K) regime.

---

> > ### Comment · Reviewer_ERuE · 2025-08-08
> >
> > My sincere thanks to the author(s) for their detailed discussion on the potential noisy manifolds and I will maintain my score to recommend the paper for publication.

---

### Note · Authors · 2025-08-13

Through a constructive rebuttal phase, all reviewers ultimately either maintained positive scores or increased their initial scores.

Reviewer KUrq acknowledged our rebuttal and recognized that the changes we proposed based on it would improve the manuscript, yet indicated an intention to maintain their initial score. As stated in our response, we are committed to incorporating these changes into the revised manuscript, and we hope that the final decision will take into account the substance of our rebuttal.

Reviewer BzDS provided the most detailed comments. We believe that most of the technical points raised were addressed during the rebuttal phase. Furthermore, as this reviewer suggested, the discussion helped bring to light several aspects of novelty that should be emphasized more clearly. We believe that incorporating this clarification into the Introduction section will make the manuscript more valuable to readers.

We sincerely thank all reviewers and the Area Chairs for their time, constructive feedback, and thoughtful engagement throughout the review process.

---

### Decision · Program_Chairs · 2025-09-17

**Decision:**

Reject

**Comment:**

This paper investigates the statistical properties of Fréchet regression in CAT(K) spaces, i.e., metric spaces with an upper curvature bound. The main contributions are non-asymptotic convergence results (Section 3.2) for the kernel Fréchet regression estimator and angle stability results (Section 3.3) for the nonparametric (kernel-based) conditional Fréchet mean estimator.

The reviewers raised several concerns. In particular, Reviewer BzDS pointed out that parts of the paper are unclearly written or potentially misleading. More generally, the scope and motivation of the paper remain limited, and the presentation of the results lacks clarity. While the authors provide some discussion of applications and position their work relative to the literature, the significance and novelty are not sufficiently established.

The AC also found several unclear points in the main text and is not convinced by the current presentation. The experiments are also quite toy-level. Due to the high competition this year, the AC recommends rejection but suggests the authors to improve the motivation, clarify the scope and contributions, strengthen the related work discussion, and refine the theorem statements for a future conference or journal submission.